# Flock: A Knowledge Graph Foundation Model via Learning on Random Walks

**Jinwoo Kim**[1*]  **Xingyue Huang**[2*]  **Krzysztof Olejniczak**[2]  **Kyungbin Min**[1]
**Michael Bronstein**[2,4]  **Seunghoon Hong**[1]  **İsmail İlkan Ceylan**[3,4,2]
[1]KAIST   [2]University of Oxford   [3]TU Wien   [4]AITHYRA

## Abstract

We study the problem of zero-shot link prediction on knowledge graphs (KGs), which requires models to generalize to *novel entities* and *novel relations*. Knowledge graph foundation models (KGFMs) address this task by enforcing equivariance over *both* nodes and relations, which enables them to learn structural properties of nodes and relations that transfer to novel KGs with similar structure. However, the conventional notion of deterministic equivariance inherently limits the expressive power of KGFMs, as it prevents them from distinguishing relations that are structurally similar but semantically distinct. To overcome this limitation, we propose to leverage *probabilistic* node-relation equivariance, which preserves equivariance *in distribution* while using structured randomness to break symmetries at inference time. Building on this principle, we present Flock, a KGFM that iteratively samples random walks, encodes them into sequences, embeds them with a sequence model, and aggregates node and relation representations through learned pooling. Flock respects probabilistic node-relation equivariance and, crucially, is a *universal approximator* for isomorphism-invariant link-level functions over KGs. Empirically, Flock perfectly solves our new diagnostic dataset Petals on which current KGFMs fail, and achieves state-of-the-art performance on entity and relation prediction tasks across 54 KGs from diverse domains.

## 1 Introduction

Knowledge graph foundation models (KGFMs) (Galkin et al., 2024; Zhang et al., 2024; Huang et al., 2025) aim to infer missing links over novel knowledge graphs (KGs) that are not part of the training data or domains. This task requires generalization to *both* unseen nodes and unseen relation types. To achieve this, KGFMs learn *node and relation invariants*: structural properties of nodes and relations that are transferable across KGs even when their relational vocabularies differ. This inductive bias is formalized as double equivariance (Gao et al., 2023)—equivariance under permutations of both entities and relations—and used as a core design principle of current KGFMs.

**Problem statement.** In this work, we challenge a fundamental assumption dictated by strict equivariance in existing KGFMs: *structural isomorphism of relations implies semantic equivalence*. Consider, for example, the KG in Figure 1, where the relations like and dislike are structurally isomorphic yet semantically opposite. Any KGFM that computes relation invariants is forced to assign the same representation to both like and dislike—losing the ability to distinguish entities with opposite relationships. This expressiveness limitation is an architectural one and *cannot* be resolved through finetuning, further limiting the downstream use of existing KGFMs. This raises a central question: how can we design KGFMs that are both *expressive* and have the right *inductive bias* for generalization?

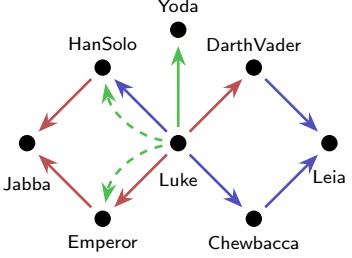

Figure 1: A KG representing characters' relationships in Star Wars movies. Blue arrows indicate like, red – dislike, and green arrows indicate friendWith.

---

*Equal contribution.

**Approach.** We propose a new approach for KGFMs that relies on *probabilistic* node-relation equivariance as inductive bias, instead of enforcing *deterministic* equivariance over nodes and relations. Probabilistic equivariance relaxes the hard constraint that "structurally isomorphic relations *must* have identical representations", and requires only that the representations of structurally isomorphic relations need to be equivalent *in distribution* over a model's stochastic predictive processes (Srinivasan & Ribeiro, 2020; Abboud et al., 2021). This way, the model retains the inductive bias needed for generalizing across different KGs, while the stochasticity of each forward pass ensures that structurally identical but semantically distinct relations are assigned different representations.

Inspired by the success of models that learn probabilistic invariants via random walks (Perozzi et al., 2014; Grover & Leskovec, 2016; Nikolentzos & Vazirgiannis, 2020; Kim et al., 2025), we introduce FLOCK, a KGFM that inherently computes probabilistic node-relation invariants. Given a potentially unseen KG and a query, FLOCK iteratively samples random walks over the KG based on the query, encoding both encountered nodes and relations with a recording protocol. To ensure the model can generalize to unseen entities and relation types, we *anonymize* all nodes and relations, enforcing that FLOCK only learn from their structural roles. These anonymized sequences are then fed into a sequence processor, and the representations for each node and relation are aggregated via a consensus protocol. Finally, we construct per-query features from the aggregated entity and relation embeddings and input them into a binary classifier for link prediction.

**Key findings and contributions.** The design of FLOCK offers several key advantages over existing KGFMs. First, it entirely abandons the conventional two-stage process of encoding relations and node representations via two separate networks, and does not rely on message passing at all, thereby avoiding the well-known expressivity limitations of MPNNs on KGs (Barceló et al., 2022; Huang et al., 2023; 2025). Second, FLOCK is a universal approximator (Proposition 4.1), capable of approximating every link-level function on KGs of any bounded size. Finally, FLOCK's architecture inherently respects probabilistic node-relation equivariance, enabling strong generalization. Our experiments on both entity and relation prediction validate this approach, demonstrating that FLOCK consistently outperforms state-of-the-art KGFMs on existing benchmarks. Our contributions are:

- We highlight a limitation of existing KGFMs: their reliance on deterministic node–relation equivariance prevents them from distinguishing between structurally similar but semantically different relations, limiting their expressivity.

- We propose to leverage probabilistic node-relation equivariance, a property for KGFMs that ensures equivariance only in distribution, as an effective solution balancing the model expressivity and generalization.

- We propose FLOCK, a KGFM that respects probabilistic node-relation equivariance. FLOCK replaces the conventional two-stage, message-passing paradigm with a direct sequence encoding approach based on random walks, and is a universal approximator of link-level functions.

- We validate our approach on entity and relation prediction tasks across 54 diverse KGs, where FLOCK consistently achieves state-of-the-art performance. We further construct a synthetic dataset PETALS to confirm our theoretical results empirically.

All proofs are in §B. The code is available at `https://github.com/jw9730/flock`.

## 2 RELATED WORK

**Link prediction and KGFMs.** Early methods for inferring missing links in KGs relied on learned embeddings (Bordes et al., 2013; Sun et al., 2019; Balazevic et al., 2019; Abboud et al., 2020; Schlichtkrull et al., 2018; Vashishth et al., 2020), operating in the *transductive* setting and incapable of generalizing to unseen entities or relation types. Later MPNN-based methods based on the labeling trick (Teru et al., 2020; Zhang et al., 2021) or conditional message passing (Zhu et al., 2021; 2023; Zhang & Yao, 2022; Zhang et al., 2023b; Huang et al., 2023) unlocked the *node-inductive* scenario, while remaining restricted to a fixed relational vocabulary. KGFMs eliminate this restriction and enable *node-relation inductive* link prediction over both unseen nodes and relations (Geng et al., 2023), typically by first encoding relations and then nodes. Early examples are InGram (Lee et al., 2023) and ULTRA (Galkin et al., 2024), later extended by TRIX (Zhang et al., 2024) into a more expressive framework. KG-ICL (Cui et al., 2024) achieved full inductiveness by combining in-context

learning with node-relation tokenization. ISDEA (Gao et al., 2023) and MTDEA (Zhou et al., 2023) highlighted the benefits of node-relation equivariance, while MOTIF (Huang et al., 2025) proposed a general KGFM framework with a theoretical analysis of their expressive power. Our work advances the field with a stochastic KGFM that achieves invariance in probability to ensure generalization while being provably more expressive than the existing methods. Notably, FLOCK achieves universality without any form of message passing, instead relying on random walks and sequence models to encode both nodes and relations anonymously. This is distinct from prior stochastic KGFMs that rely on random initialization of message passing (Lee et al., 2023; Gao et al., 2023).

**Random walks for graph representations.** Random walks have been widely adopted in graph learning due to their simplicity and ability to gather context from neighborhoods. DeepWalk (Perozzi et al., 2014) and node2vec (Grover & Leskovec, 2016) were among the first approaches, treating walks as analogues of natural language sentences and processing them with skip-gram models. Subsequent work has developed diverse neural architectures based on random walks: Nikolentzos & Vazirgiannis (2020) generates graph-level predictions using joint walks on direct products of graphs and their subgraphs, CRaWL (Tönshoff et al., 2021) represents a graph as a collection of walks and processes them with a convolutional network, WalkLM (Tan et al., 2023) samples walks from graphs with textual features and embeds them using a language model, RWNN (Kim et al., 2025) and RUM (Wang & Cho, 2024) anonymize walks and process them with sequence networks, and NeuralWalker (Chen et al., 2025) integrates walk encodings into message passing.

**Probabilistic invariance.** Neural architectures that enforce invariance to specific transformations often exhibit more stable training and improved performance (Bronstein et al., 2021), but this inductive bias can reduce their expressivity by preventing the model from distinguishing non-equivalent inputs. In graph learning, this trade-off is exemplified by MPNNs, whose power is limited by the 1-WL test (Xu et al., 2019; Morris et al., 2019). Randomization has emerged as a solution, enhancing expressivity through techniques such as noise injection (Abboud et al., 2021), vertex dropping (Papp et al., 2021), subgraph sampling (Bevilacqua et al., 2022; Zhang et al., 2023a), dynamic rewiring (Finkelshtein et al., 2024), and random walks (Kim et al., 2025; Wang & Cho, 2024). Despite their stochasticity, such methods can remain probabilistically invariant, ensuring that equivalent inputs yield identical expected outputs, or even identical output distributions. In line with a prior work (Gao et al., 2023), we extend the notion of probabilistic invariance to KGs and prove that FLOCK satisfies invariance in distribution, and further clarify its usefulness in KG learning.

## 3  PRELIMINARY

**Knowledge graphs.** A *knowledge graph* (KG) is a tuple $G = (V, E, R)$, where $V$ denotes the set of entities (nodes), $R$ the set of relation types, and $E \subseteq V \times R \times V$ the set of labeled edges (*facts*). A fact is written as $(h, r, t)$ (or $h \xrightarrow{r} t$ interchangeably) with $r \in R$ and $h, t \in V$. A (potential) *link* in $G$ is any triple $(h, r, t)$ in $V \times R \times V$, regardless of whether it is present in $E$. We denote by $R^{-1}$ the set of inverses of relations $R$, defined as $\{r^{-1} \mid r \in R\}$, and mean $r$ when writing $(r^{-1})^{-1}$. Further, let $\mathbb{K}_{n,m}$ be the space of knowledge graphs with $n$ vertices and $m$ relation types.

**Isomorphism.** An *isomorphism* between two knowledge graphs $G = (V, E, R)$ and $G' = (V', E', R')$ is a pair of bijections $\mu = (\pi, \phi)$, where $\pi : V \to V'$ and $\phi : R \to R'$, such that a fact $(h, r, t)$ belongs to $E$ if and only if the fact $\mu((h, r, t)) = (\pi(h), \phi(r), \pi(t))$ belongs to $E'$. Two KGs are *isomorphic* if such a mapping exists, in which case we write $G \simeq G'$.

**Link invariance.** In this work, we focus on link-invariant functions. Let $\omega$ be a function assigning to each KG $G = (V, E, R) \in \mathbb{K}_{n,m}$ a map $\omega(G) : V \times R \times V \to \mathbb{R}^d$. We say that $\omega$ is *link invariant* if for every pair of isomorphic KGs $G, G' \in \mathbb{K}_{n,m}$, every isomorphism $(\pi, \phi)$ from $G$ to $G'$, and every link $(h, r, t)$ in $G$, we have $\omega(G)((h, r, t)) = \omega(G')((\pi(h), \phi(r), \pi(t)))$.

**Probabilistic invariance.** A stochastic KG model $\varphi$ can be viewed as a function that takes a KG and returns a random variable $\varphi(G)$. Following Kim et al. (2025), we call $\varphi$ *invariant in probability* if

$$\forall G, G' \in \mathbb{K}_{n,m} : \qquad G \simeq G' \implies \varphi(G) \stackrel{d}{=} \varphi(G'),$$

i.e., the distributions of $\varphi(G)$ and $\varphi(G')$ are equal. In particular, this implies $\mathbb{E}[\varphi(G)] = \mathbb{E}[\varphi(G')]$.

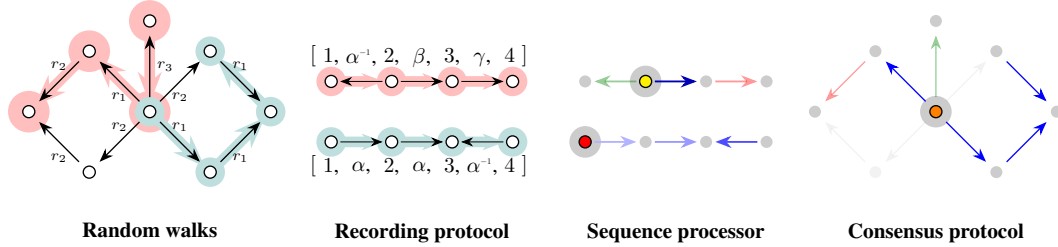

Random walks        Recording protocol        Sequence processor        Consensus protocol

Figure 2: **An overview.** In each updating step, FLOCK **(1)** samples random walks on the KG (two walks indicated by red and teal, respectively), **(2)** anonymizes the encountered nodes and relations via a recording protocol (for each walk, nodes are anonymized as $1, 2, ...$ and relations as $\alpha, \beta, ...$), and **(3)** feeds the sequences in a sequence processor to compute node and relation representations. **(4)** A consensus protocol then pools them back to the original KG's nodes and relations.

## 4   METHODOLOGY

We present FLOCK, a KGFM respecting probabilistic node-relation invariance. FLOCK is a randomized function $X_\theta(\cdot)$ which takes as input a KG $G = (V, E, R)$ and a link prediction query $q$. We consider two types of queries: *entity prediction* $q = (h, r, ?)$ and *relation prediction* $q = (h, ?, t)$. FLOCK outputs a random variable $\hat{\mathbf{y}} \sim X_\theta(G, q)$ which is suited for the task at hand. For entity prediction, it outputs $\hat{\mathbf{y}} : V \to [0, 1]$ such that a potential link $(h, r, t)$ can be evaluated by $\hat{\mathbf{y}}(t) \in [0, 1]$. For relation prediction, it outputs $\hat{\mathbf{y}} : R \to [0, 1]$ such that a link $(h, r, t)$ can be evaluated by $\hat{\mathbf{y}}(r)$.

At test time, we average multiple ($P$) independent stochastic predictions to produce the final output. This improves performance and reduces variance through an ensembling effect.

We describe the architecture of FLOCK in §4.1 focused on four main components, and then analyze its theoretical properties in §4.2, showing universality and probabilistic equivariance. An expansion on the model details can be found in §A.

### 4.1   FLOCK

Internally, FLOCK has two lookup tables of hidden states, $\mathbf{v} : V \to \mathbb{R}^d$ for entities and $\mathbf{r} : R \to \mathbb{R}^d$ for relations, respectively. At each forward pass, it starts from trained initializations of these states $\mathbf{v}^{(0)}(\cdot) := \mathbf{v}_0$ and $\mathbf{r}^{(0)}(\cdot) := \mathbf{r}_0$, and updates them iteratively $\mathbf{v}^{(i)}, \mathbf{r}^{(i)}$ for $i \leq I$. Each update is done residually using a randomized function $U_{\theta_i}$:

$$\mathbf{v}^{(i+1)} := \mathbf{v}^{(i)} + \Delta\mathbf{v}, \qquad \mathbf{r}^{(i+1)} := \mathbf{r}^{(i)} + \Delta\mathbf{r}, \qquad (\Delta\mathbf{v}, \Delta\mathbf{r}) \sim \text{update}_{\theta_i}(\mathbf{v}^{(i)}, \mathbf{r}^{(i)}).$$

The final hidden states $\mathbf{v}^{(I)} : V \to \mathbb{R}^d$ and $\mathbf{r}^{(I)} : R \to \mathbb{R}^d$ are then processed by a binary classifier $\text{head} : \mathbb{R}^d \to [0, 1]$ to produce the output $\hat{\mathbf{y}}$ which is $V \to [0, 1]$ or $R \to [0, 1]$ depending on task.

We now describe the randomized $\text{update}_\theta$. We drop $i$ for brevity. It consists of four components:

1. **Random walk algorithm** produces $n$ random walks $\eta_1, ..., \eta_n$ of length $\ell$ on the input KG.

2. **Recording protocol** $w : \eta_j \mapsto \mathbf{z}_j$ transforms each walk into a graph-agnostic sequence.

3. **Sequence processor** $f_\theta : \mathbf{z}_j \mapsto \mathbf{h}_j$ processes each sequence independently, outputting features.

4. **Consensus protocol** $c : (\mathbf{h}_{1:N}, \eta_{1:N}) \mapsto (\Delta\mathbf{v}, \Delta\mathbf{r})$ collects features of all walks and decides state updates for each entity and relation type.

An overview is presented in Figure 2. We note that $w$, $f_\theta$, and $c$ are all deterministic, and the random walk is the only source of stochasticity. We now discuss the design choice for each. For the ease of exposition, we explain for entity prediction tasks $q = (h, r, ?)$, but relation prediction is similar.

**Random walks.**    In FLOCK, random walks are central in two ways: they rewrite the connectivity of nodes and relations as sequences, and support generalization via probabilistic equivariance.

Formally, the random walk algorithm produces $n$ random walks $\eta_1, ..., \eta_n$ of length $\ell$ on KG $G$. Each random walk $\eta$ is a chain of random variables, written as:n as:

$$\eta = v_0 \xrightarrow{r_1} v_1 \xrightarrow{r_2} \cdots \xrightarrow{r_\ell} v_\ell, \qquad v_s \in V, r_s \in R, (v_{s-1}, r_s, v_s) \in E,$$

where the underlying transition mechanism and $\ell$ are hyperparameters.

To support probabilistic equivariance, we ask the walk algorithm to be invariant in probability. We say $\eta$ is invariant in probability if for any $G \simeq H$ in $\mathbb{K}_{n,m}$ with isomorphism $(\pi, \phi)$ from $G$ to $H$:

$$\pi(v_0) \xrightarrow{\phi(r_1)} \pi(v_1) \xrightarrow{\phi(r_2)} \cdots \xrightarrow{\phi(r_\ell)} \pi(v_\ell) \stackrel{d}{=} u_0 \xrightarrow{s_1} u_1 \xrightarrow{s_2} \cdots \xrightarrow{s_\ell} u_\ell,$$

where $v_0 \xrightarrow{r_1} \cdots \xrightarrow{r_\ell} v_\ell$ and $u_0 \xrightarrow{s_1} \cdots \xrightarrow{s_\ell} u_\ell$ follow the distributions of $\eta(G, \ell)$ and $\eta(H, \ell)$, respectively. In such case, the isomorphism $(\pi, \phi)$ yields a natural translation from walks in $G$ to $H$.

In FLOCK, we use a simple random walk algorithm which we show to be invariant in probability. Specifically, we use uniform walks with non-backtracking, with minor modifications to handle directed multi-edges of KGs. Despite the simplicity, we find that this choice works well in practice, consistent with findings of prior works (Tönshoff et al., 2021; Kim et al., 2025).

Under this choice, we diversify the starting locations of walks such that local context around the query $q$ and broad regions of the nodes and relations in a KG are both well-captured. Our *diversification strategy* is as follows: given a base walk count $n$, for entity prediction queries $(h, r, ?)$, we use $3n$ walks with three types of start locations. The first $n$ walks start at query node $h$, capturing local context around the query; the second $n$ walks start by traversing a random edge $(v, s, u)$ where $s$ is a uniformly chosen relation, broadly capturing the relations of the KG including $r$; the last $n$ walks start at random nodes, broadly capturing various regions of the KG. For relation prediction queries $(h, ?, t)$, we additionally start $n$ walks at the tail node $t$, sampling a total of $4n$ walks.

We lastly discuss how to choose the base walk count $n$. While this is a fixed hyperparameter $n_{\text{train}}$ at pretraining, we find that scaling it adaptively to input KG at test-time benefits size generalization. We thus propose *test-time adaptation of walk counts*, and use:

$$n = n_{\text{train}} \times \text{harmonic mean} \left( \frac{|V|}{|V|_{\text{train}}}, \frac{|E|}{|E|_{\text{train}}} \right), \tag{1}$$

where $|V|_{\text{train}}, |E|_{\text{train}}$ are average numbers of nodes and edges in pretraining KGs, respectively. Intuitively, this scales $n$ proportionally to the sizes of test KGs relative to pretraining. In practice, we clamp $n$ to the nearest power of 2 and limit its value in an interval to avoid out-of-memory errors.

**Recording protocol.** While random walks provide a basis for invariant sequence representations of KGs, two issues remain: (1) They reveal nodes $v_s$ and relations $r_s$ specific to each KG which obstructs transferability to unseen KGs. (2) They do not offer a way to condition on current states of entities $\mathbf{v}$, relations $\mathbf{r}$, and the query $q = (h, r, ?)$ as often done in KGFMs via the labeling trick.

The recording protocol $w : \eta_j \mapsto \mathbf{z}_j$ resolves this by transforming each walk into a *graph-agnostic* sequence that only leaves structural information. Motivated by prior works on node anonymization for invariance (Kim et al., 2025; Wang & Cho, 2024), we propose an extension called node-relation anonymization: reserve separate namespaces for nodes and relations, respectively, and assign unique names in the order of discovery. For example, with $1, 2, 3, ...$ for nodes and $\alpha, \beta, ...$ for relations:

$$\eta = v_0 \xrightarrow{r_1} v_1 \xrightarrow{r_2} v_2 \xrightarrow{r_1^{-1}} v_0 \qquad \mapsto \qquad 1 \xrightarrow{\alpha} 2 \xrightarrow{\beta} 3 \xrightarrow{\alpha^{-1}} 1,$$

where $(\cdot)^{-1}$ marks direction of a relation. The protocol additionally employs a simple conditioning on current states $(\mathbf{v}, \mathbf{r})$ and query $q = (h, r, ?)$, completing the record $\mathbf{z}$ as follows:

$$w : \eta \mapsto \mathbf{z} = (1, \mathbf{v}(v_0), \mathbf{1}_h(v_0)) \xrightarrow{\alpha, \mathbf{r}(r_1), \mathbf{1}_r(r_1)} (2, \mathbf{v}(v_1), \mathbf{1}_h(v_1)) \xrightarrow{\beta, \mathbf{r}(r_2), \mathbf{1}_r(r_2)} \cdots, \tag{2}$$

with indicator functions $\mathbf{1}_h(\cdot), \mathbf{1}_r(\cdot)$ at $h$ and $r$, respectively. As we will show, the recording protocol keeps node-relation invariance by hiding nodes and relations while leaving their structural roles.

**Sequence processor.** Now that the recordings $\mathbf{z}$ only encode structural information of KG, we can safely process them with an arbitrary neural network $f_\theta : \mathbf{z} \mapsto \mathbf{h}$ without the risk of losing invariance. Since $\mathbf{z}$ are sequences, we choose sequence networks to leverage their inductive bias. Specifically, we use bidirectional GRU (Cho et al., 2014) equipped with RMSNorm (Zhang & Sennrich, 2019) and SwiGLU feedforward network (Shazeer, 2020), which provided robust results. To convert anonymizations into input feature vectors to the GRU, we use trainable embedding tables.

Given that $f_\theta$ is a sequence network, it is convenient to interpret its output $\mathbf{h}$ as positionally aligned with each step of the walk $\eta$ or record $\mathbf{z}$. Specifically, for the example in Equation 2, we obtain:

$$f_\theta : \mathbf{z} \mapsto \mathbf{h} = (\Delta\mathbf{v}_0, a_0) \xrightarrow{\Delta\mathbf{r}_1, b_1} (\Delta\mathbf{v}_1, a_1) \xrightarrow{\Delta\mathbf{r}_2, b_2} \cdots .$$

where $\Delta\mathbf{v}_s, \Delta\mathbf{r}_s \in \mathbb{R}^{h \times d_h}$ and $a_s, b_s \in \mathbb{R}^h$ are the decoded outputs at each position using linear projections. Intuitively, $\Delta\mathbf{v}_s, \Delta\mathbf{r}_s$ encode proposals of state updates for entities and relations by $f_\theta$, and $a_s, b_s$ encode respective confidences of $f_\theta$ for the proposed updates. This separation is useful due to the localized, pure-structure nature of the recordings $\mathbf{z}$. If a random walk $\eta$ densely visited a cycle-like region and then terminated in a dangling manner, it is natural to assign more confidence to the cycle-like region of the structural encodings $\mathbf{h}$, and less confidence to the dangling region.

**Consensus protocol.** After sequence processing, we are left with a handful of state update proposals $\mathbf{h}_{1:N}$ from $f_\theta$, that are positionally aligned with random walks $\eta_{1:N}$ on KG. The consensus protocol $c$ uses the information to decide final state updates $\Delta\mathbf{v} : V \to \mathbb{R}^d$ and $\Delta\mathbf{r} : R \to \mathbb{R}^d$.

Since $c$ can access how each $\Delta\mathbf{v}_s$ within $\mathbf{h}_j$ is associated to a node $v_s \in V$ (and how each $\Delta\mathbf{r}_s$ is associated to a relation $r_s \in R$) through the random walk $\eta_j$, a simple way to form a consensus is by finding all proposals $\{\Delta\mathbf{v}_s\}$ associated to each node $v$, and all $\{\Delta\mathbf{r}_s\}$ associated to each relation $r$, and take averages of these proposals. The drawback is that uninformative proposals from e.g., dangling regions of walks are not directly suppressed, and can affect the state updates.

We can leverage the confidences $a_s, b_s$ from $f_\theta$ to alleviate this issue. For each node $v \in V$ or relation $r \in R$, we first find all respective associated pairs $\{(\Delta\mathbf{v}_s, a_s)\}$ or $\{(\Delta\mathbf{r}_s, b_s)\}$ of proposals and confidences, and compute a multi-head softmax-normalized weighted average:

$$\Delta\mathbf{v}(v) \coloneqq \left[\sum \exp(a_s) \odot \Delta\mathbf{v}_s\right] \oslash \sum \exp(a_s) \quad \Delta\mathbf{r}(r) \coloneqq \left[\sum \exp(b_s) \odot \Delta\mathbf{r}_s\right] \oslash \sum \exp(b_s),$$

where $\odot$ and $\oslash$ are row-wise multiplication and division, respectively. Intuitively, this normalization induces competition between state update proposals, naturally leading to uninformative proposals being suppressed. Similar ideas are presented by Locatello et al. (2020).

Again, we can show that the consensus protocol does not operate in a way specific to particular KGs, and hence retains node-relation equivariance.

## 4.2 Theoretical analysis

**Expressivity.** Following the notion of probabilistic expressivity introduced by Abboud et al. (2021), we say that a FLOCK model $X_\theta$ is a universal approximator of link invariant functions over $\mathbb{K}_{n,m}$ if for any link invariant $\varphi : \mathbb{K}_{n,m} \to (V \times R \times V \to [0,1])$ and any $\epsilon, \delta > 0$, there exists a choice of the network parameters $\theta$ and the length of the sampled random walks $\ell$, such that:

$$\mathbb{P}(|\varphi(G)((h,r,t)) - X_\theta(G, (h,r,?))(t)| < \epsilon) > 1 - \delta$$

for all graphs $G = (V, E, R) \in \mathbb{K}_{n,m}$ and all links $(h, r, t) \in V \times R \times V$.

**Proposition 4.1.** *With a powerful enough sequence processor $f_\theta$, the* FLOCK *framework described above is a universal approximator of link invariant functions over $\mathbb{K}_{n,m}$ for all pairs $(n, m)$.*

All proofs are in §B. To offer an intuition behind the result, we provide a proof sketch.

*Proof sketch.* A sufficiently long random walk will cover all edges of the graph with high probability. Then, from its anonymized version, assigning unique positional identifiers to every node and relation, one can reconstruct the input graph, up to isomorphism. Thus, with a sufficiently expressive sequence processor, FLOCK can approximate any link-invariant function.

**Invariance.** Despite the stochastic nature of our framework, beyond randomized node embeddings (Abboud et al., 2021), FLOCK can be provably guaranteed to satisfy probabilistic invariance:

**Proposition 4.2.** *Suppose that the walk sampling protocol $\eta$ is invariant in probability and both the recording protocol $w$ and the consensus protocol $c$ are invariant. Then, regardless of the choice of the deterministic sequence processor $f_\theta$, the corresponding* FLOCK *model is invariant in probability.*

*Proof sketch.* Since each of these components is invariant (in probability), and invariance of individual component is preserved under composition, we have that FLOCK is invariant.

Moreover, the designs of FLOCK's components provided earlier in this section satisfy the conditions of Proposition 4.2. Therefore, the suggested pipeline is indeed invariant in probability:

**Proposition 4.3.** *Any* FLOCK *model with components as outlined in this section, and detailed in §A is invariant in probability.*

## 5 EXPERIMENTS

We test FLOCK over a wide range of KGs for both entity and relation predictions, aiming to answer:

**Q1.** Can FLOCK approximate functions that existing KGFMs cannot?

**Q2.** How does FLOCK generalize to unseen entities and relations compared to existing KGFMs?

**Q3.** How does performance scale with the sizes of pretraining graph mix and test-time ensemble?

**Q4.** What is the impact of choices of each component on the behavior and performance of FLOCK?

We further provide detailed scalability analysis in §D and §E, and comparisons against current KGFMs augmented with noise injection in §F. Experiment details and hyperparameters are in §I.

### 5.1 SYNTHETIC DATASET

**Setup.** To validate the limitations of KGFMs with node-relation equivariance (**Q1**), we construct a synthetic benchmark PETALS. It contains 220 instances, each including: **(1)** a KG $G = (V, E, R)$ consisting of a 'central' node $s$, a 'stem' $T \subset V$ with query relation $r_0$, and multiple cyclic 'petals', each 'colored' with a different pair of relations in $R \setminus \{r_0\}$, **(2)** an entity prediction query $(h, r_0, ?)$ with $h \in \{s\} \cup T$, and **(3)** two candidate targets $t_1$ and $t_2$ from the same 'petal', located at the same distance from $s$. An example is in Figure 3. See §C for more details.

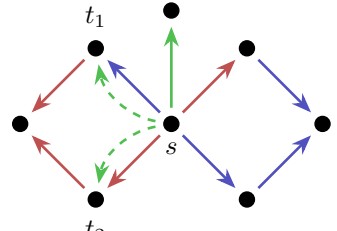

Figure 3: A KG in PETALS. KGFMs with relational invariants must equate $r_1$ and $r_2$, thus predicting the same scores for dashed queries with $r_0$.

PETALS is designed such that each instance always admits non-trivial automorphisms, meaning that swapping relations occurring in the same 'petal' results in an isomorphic KG. Consequently, any model computing relation invariants will not be able to distinguish between potential links $(s, r_0, t_1)$ and $(s, r_0, t_2)$. However, the samples are constructed so that these links are not isomorphic from the graph perspective, making them distinguishable for general link-invariant functions. We say a model *solves* an instance if it can classify $(s, r_0, t_1)$ as TRUE and $(s, r_0, t_2)$ as FALSE. For empirical validation, we train ULTRA (Galkin et al., 2024), MOTIF($\mathcal{F}_{\text{Path}}^3$) (Huang et al., 2025), TRIX (Zhang et al., 2024), and FLOCK from scratch and validate them on the training instances.

**Results.** The results are provided in Table 1. As expected, all existing KGFMs relying on learning deterministic relational invariants fail to distinguish between the candidate target triplets completely, achieving 50% accuracy due to random guesses. In contrast, FLOCK succeeds on *all* considered instances, displaying that, while remaining invariant in probability, it can differentiate between non-isomorphic links, even with isomorphic relations.

Table 1: PETALS accuracies.

| Model | PETALS |
|---|---|
| ULTRA | 50% |
| MOTIF($\mathcal{F}_{\text{Path}}^3$) | 50% |
| TRIX | 50% |
| **FLOCK** | **100%** |

Table 2: Average entity prediction MRR and Hits@10 over 54 KGs from distinct domains.

| Model | Inductive $e, r$ (23 graphs) | | Inductive $e$ (18 graphs) | | Transductive (13 graphs) | | Total Avg (54 graphs) | | Pretrained (3 graphs) | |
|---|---|---|---|---|---|---|---|---|---|---|
| | MRR | H@10 | MRR | H@10 | MRR | H@10 | MRR | H@10 | MRR | H@10 |
| ULTRA (zero-shot) | 0.345 | 0.513 | 0.431 | 0.566 | 0.312 | 0.458 | 0.366 | 0.518 | - | - |
| TRIX (zero-shot) | 0.368 | 0.540 | 0.455 | 0.592 | 0.339 | 0.500 | 0.390 | 0.548 | - | - |
| FLOCK (zero-shot) | **0.369** | **0.554** | **0.456** | **0.604** | **0.340** | **0.509** | **0.391** | **0.560** | - | - |
| ULTRA (finetuned) | 0.397 | 0.556 | 0.440 | 0.582 | 0.379 | 0.543 | 0.408 | 0.562 | 0.407 | **0.568** |
| TRIX (finetuned) | 0.401 | 0.556 | 0.459 | 0.595 | **0.390** | **0.558** | 0.418 | 0.569 | **0.415** | 0.564 |
| FLOCK (finetuned) | **0.417** | **0.576** | **0.473** | **0.619** | 0.383 | 0.544 | **0.427** | **0.582** | 0.415 | 0.561 |

Table 3: Average relation prediction MRR and Hits@1 over 54 KGs from distinct domains.

| Model | Inductive $e, r$ (23 graphs) | | Inductive $e$ (18 graphs) | | Transductive (13 graphs) | | Total Avg (54 graphs) | | Pretrained (3 graphs) | |
|---|---|---|---|---|---|---|---|---|---|---|
| | MRR | H@1 | MRR | H@1 | MRR | H@1 | MRR | H@1 | MRR | H@1 |
| ULTRA (zero-shot) | 0.785 | 0.691 | 0.714 | 0.590 | 0.629 | 0.507 | 0.724 | 0.613 | - | - |
| TRIX (zero-shot) | 0.842 | 0.770 | 0.756 | 0.611 | 0.752 | 0.647 | 0.792 | 0.687 | - | - |
| FLOCK (zero-shot) | **0.898** | **0.846** | **0.864** | **0.782** | **0.873** | **0.813** | **0.881** | **0.817** | - | - |
| ULTRA (finetuned) | 0.823 | 0.741 | 0.716 | 0.591 | 0.707 | 0.608 | 0.759 | 0.659 | 0.876 | 0.817 |
| TRIX (finetuned) | 0.850 | 0.785 | 0.759 | 0.615 | 0.785 | 0.693 | 0.804 | 0.706 | 0.879 | 0.797 |
| FLOCK (finetuned) | **0.929** | **0.889** | **0.887** | **0.808** | **0.897** | **0.844** | **0.907** | **0.851** | **0.977** | **0.959** |

## 5.2 ENTITY AND RELATION PREDICTION OVER KNOWLEDGE GRAPHS

**Setup.** To answer **Q2**, we follow the protocol of Galkin et al. (2024); Zhang et al. (2024) and pretrain FLOCK on FB15k-237 (Toutanova & Chen, 2015), WN18RR (Dettmers et al., 2018), and CoDEx Medium (Safavi & Koutra, 2020). We then evaluate its zero-shot and finetuned inference performance with the test set of 54 KGs (see §I for details). These KGs are extracted from diverse domains across three settings: inductive on nodes and relations (**Inductive** $e, r$), inductive on nodes (**Inductive** $e$), and **transductive**. Note that these settings differ only during finetuning setup; in zero-shot setup, all entities and relations are unseen. We choose state-of-the-art KGFMs ULTRA (Galkin et al., 2024) and TRIX (Zhang et al., 2024) as baselines, as they are pretrained on the same KGs, to ensure a fair comparison. For evaluation, we use the filtered ranking protocol (Bordes et al., 2013), reporting mean reciprocal rank (MRR) and Hits@10 for entity prediction, and Hits@1 for relation prediction, as some KGs have fewer than 10 relations. Per-dataset results are in §I.

**Entity prediction.** Table 2 shows the entity prediction results. In the zero-shot setting, FLOCK consistently outperforms ULTRA and TRIX, demonstrating strong generalization across diverse domains. Notably, on *Metafam* (Zhou et al., 2023), a dataset designed to challenge models with conflicting and compositional relational patterns, FLOCK roughly doubles MRR over ULTRA and achieves $\approx 40\%$ gain in MRR over TRIX in the zero-shot setting. We find that FLOCK distinguishes structurally similar but semantically conflicting relations while ULTRA fails (§G), explaining the gain. These findings align with our hypothesis that probabilistic equivariance improves expressivity without sacrificing generalization. In the finetuning setting, we observe a similar pattern: FLOCK maintains a consistent improvement over all datasets except transductive ones, where KGs are generally larger. We hypothesize that the gap stems from walk coverages. Unlike ULTRA and TRIX, whose message passing guarantees a full neighborhood coverage around the queried node, FLOCK relies on sampled walks that may not fully cover target nodes of interest. We find that FLOCK favors sparse KGs (§H), consistent with this hypothesis as random walks cover sparse graphs faster.

**Relation prediction.** Table 3 presents the relation prediction results. FLOCK substantially outperforms all existing KGFMs across all categories in the zero-shot setting, achieving an $11.2\%$ relative improvement in MRR compared to the best baseline TRIX. FLOCK shows a further performance boost of $12.8\%$ in the finetuned setting. We attribute this huge gain to FLOCK's joint encoding of entities and relations during the updating step via the sequence encoder, while existing KGFMs, ULTRA and TRIX, have separate update steps for entities and relations. This joint update mechanism yields more holistic representations of both entities and relations with minimal information loss.

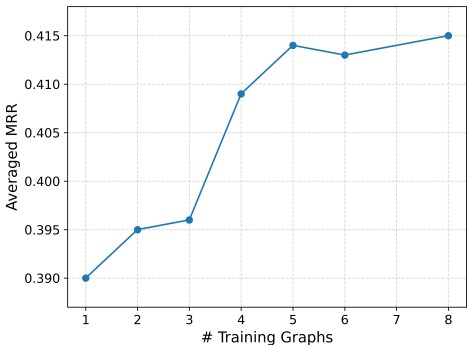
(a) Zero-shot MRR vs. #pretraining graphs.

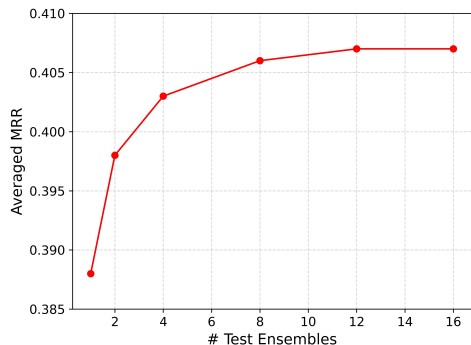
(b) Zero-shot MRR vs. #ensembled predictions.

Figure 4: Pretraining and test-time scaling of FLOCK on 41 inductive KG datasets.

Table 4: Ablation study of adaptive test-time walks with zero-shot entity prediction task. We show the average number of entities $|V|$, triples $|E|$, base walks $n$, MRR, and Hits@10.

| Dataset split | Statistics | | FLOCK | | | FLOCK w/o Adap. (1) | | |
| --- | --- | --- | --- | --- | --- | --- | --- | --- |
| | $|V|$ | $|E|$ | $n$ | MRR | Hits@10 | $n$ | MRR | Hits@10 |
| **Inductive** $e, r$ | 5,303 | 10,838 | 28.40 | **0.369** | **0.554** | 128 | 0.357 | 0.551 |
| **Inductive** $e$ | 7,578 | 29,090 | 18.08 | **0.456** | **0.604** | 128 | 0.441 | 0.596 |
| **Transductive** | 47,810 | 387,491 | 214.15 | **0.340** | **0.509** | 128 | 0.334 | 0.493 |

## 5.3 SCALING ANALYSIS

**Size of pretraining graph mix.** To assess whether FLOCK benefits from more pretraining graph and data (**Q3**), we follow the setup of Galkin et al. (2024), and pretrain FLOCK on an increasing number of KGs. We then evaluate them on all 41 inductive benchmarks for a fair comparison. We present the detailed pretraining graph mix in Table 16. As shown in Figure 4a, FLOCK's generalization improves consistently as the number of pretraining KGs increases, exhibiting clear scaling behavior, which is a core characteristic of being a foundation model.

**Number of ensembled predictions.** To assess how test-time ensemble size $P$ affects performance (**Q3**), we take the pretrained FLOCK and run zero-shot entity prediction on 41 inductive KGs by increasing the number of ensembled passes. As shown in Figure 4b, performance improves from 1 to 8 passes and then begins to plateau beyond 12. This indicates a clear scaling behavior: larger ensembles provide a more accurate estimate of the underlying node and relation distributions.

## 5.4 ABLATION STUDIES

**Setup.** To understand the impact of each design choice on the performance and behavior of FLOCK (**Q4**), we conduct a series of ablation studies spanning random walks, sequence processor and the consensus protocol, in the entity prediction task in the zero-shot setting.

**Adaptive test-time walks.** Recall that we employ *test-time adaptation of walk counts*, which adaptively selects the base walk count $n$ based on the graph size, computed via the harmonic-mean rule shown in (1) during inference. Table 4 compares this adaptive setting with a fixed setting that uses 128 base walks per sample for all datasets, matching the pretraining setup ($n_{\text{train}} = 128$). As expected, the average selected base count $n$ is smaller on both inductive splits and larger on the transductive split, yet the adaptive mechanism improves performance across all settings. This is consistent with the intuition that adaptive $n$ scales up walks on larger KGs to improve coverage while allocating fewer walks on smaller KGs to reduce redundant visits; FLOCK maintains comparable *visiting rates* and *coverage* to those seen during pretraining, thereby producing representations closer to the pretraining distribution and resulting in consistent performance gains.

Table 5: Detailed ablation study with zero-shot entity prediction task. For the transductive split, considering resource limits, we test NELL995, NELL23k, WDsinger, ConceptNet100k, and YAGO310.

| Model | Inductive $e, r$ (23 graphs) | | Inductive $e$ (18 graphs) | | Transductive (5 graphs) | | Total Avg (46 graphs) | |
|---|---|---|---|---|---|---|---|---|
| | MRR | Hits@10 | MRR | Hits@10 | MRR | Hits@10 | MRR | Hits@10 |
| FLOCK ($\ell = 128$) | 0.369 | 0.554 | 0.456 | 0.604 | 0.360 | 0.542 | **0.395** | **0.567** |
| w/o non-backtracking | 0.370 | 0.549 | 0.456 | 0.605 | 0.334 | 0.499 | 0.386 | 0.551 |
| $\ell = 64$ | 0.372 | 0.556 | 0.459 | 0.606 | 0.351 | 0.534 | 0.394 | 0.565 |
| $\ell = 256$ | 0.360 | 0.548 | 0.458 | 0.605 | 0.338 | 0.508 | 0.385 | 0.553 |
| w/o diverse starts | 0.360 | 0.539 | 0.448 | 0.596 | 0.319 | 0.488 | 0.385 | 0.553 |
| transformer $f_\theta$ | 0.356 | 0.542 | 0.410 | 0.591 | 0.312 | 0.477 | 0.359 | 0.537 |
| w/o weighted consensus | 0.351 | 0.526 | 0.448 | 0.593 | 0.361 | 0.515 | 0.387 | 0.545 |

**Non-backtracking walks.** FLOCK employs *non-backtracking* uniform random walk, which has an effect of faster exploration and coverage of distant regions (Alon et al., 2007). In Table 5, we compare this with uniform walk that may backtrack and hence is slower in global exploration. While non-backtracking does not alter results much on inductive splits, it significantly improves performance on the transductive split. This is consistent with the idea that improving coverage especially benefits performances on large KGs, which FLOCK achieves via non-backtracking.

**Walk lengths.** FLOCK uses random walks of length $\ell = 128$, a choice made by finding the lowest $\ell$ reliably visiting target node and relation on various KGs. Table 5 compares this with shorter and longer walks by a factor of two. As expected from coverage, shorter walks show degraded results on the transductive split with large KGs. Longer walks are overall worse, which is explained by higher learning complexity of the sequence processor that has a small hidden dimension (64) for scalability. FLOCK finds a balance of coverage and learnability, achieving robust results on diverse KGs.

**Diverse starting locations of walks.** We recall that FLOCK uses a *diversification* strategy of starting locations of walks, with $n$ walks from the query node, $n$ walks from random relation, and $n$ walks from random node, adding up to $3n$ walks capturing both local context near query and global information of KG. Table 5 compares this against all $3n$ walks starting at the query node. As expected, this causes degradations on all splits, showing the benefit of using both local and global information.

**Sequence processor.** FLOCK uses a sequence processor $f_\theta$ with bidirectional GRU. In Table 5, we compare this against a transformer $f_\theta$ with a similar SwiGLU-RMSNorm architecture (Dubey et al., 2024) and parameter count. This alternative does not deliver good results, which is explained by the restrictions on model scales that are enforced to scale to large KGs. FLOCK benefits from reasoning efficiency of GRU in limited parameter regime, gaining good performance and scalability together.

**Consensus protocol.** FLOCK uses softmax-weighted averaging to pool sequence processor outputs into state updates for nodes and relations, under the intuition that this can suppress uninformative proposals from the sequence processor better than simple, unweighted averaging. Table 5 provides a comparison, showing that weighted consensus outperforms the unweighted counterpart. This verifies our intuition on how the design of the consensus protocol strengthens FLOCK.

## 6 CONCLUSIONS

We introduced FLOCK, a knowledge graph foundation model that respects probabilistic node-relation equivariance. FLOCK iteratively samples query-conditioned random walks, records encountered nodes and relations via a recording protocol, and relies on a sequence processor and consensus protocol to obtain node and relation representations. We evaluate FLOCK on 54 KGs across different domains for both entity and relation prediction, demonstrating superior zero-shot and finetuned performance. We further construct a synthetic dataset PETALS to validate our theoretical findings. One limitation is scalability (§E): ensuring coverage of the sampled random walk in large KGs requires many long walks, which can quickly become computationally infeasible. A future direction is to develop approximation strategies (Chamberlain et al., 2023; Łącki et al., 2020) that reduce the cost of random walk sampling while retaining FLOCK's downstream performance. Another avenue for future work is studying the families of approximable functions when the walk lengths are restricted, for example based on connections to subgraph-based reconstructions (Cotta et al., 2021).

## ACKNOWLEDGMENT

This work was in part supported by the National Research Foundation of Korea (RS-2024-00351212 and RS-2024-00436165) and the Institute of Information & Communications Technology Planning & Evaluation (IITP) (RS-2024-00509279, RS-2022-II220926, RS-2022-II220959, and RS-2019-II190075) funded by the Korean government (MSIT). Computational resources were in part provided by the "HPC support" project funded by MSIT and NIPA. MB is supported by EPSRC Turing AI World-Leading Research Fellowship No. EP/X040062/1 and EPSRC AI Hub on Mathematical Foundations of Intelligence: An "Erlangen Programme" for AI No. EP/Y028872/1.

## ETHICS STATEMENT

This work introduces a probabilistic framework for knowledge graph foundation models, aiming to improve the generalization in zero-shot link prediction. Our contributions are methodological and theoretical, with evaluations performed on publicly available benchmarks and a synthetic dataset designed to validate our theoretical results. We do not anticipate any direct ethical risks associated with this approach. We acknowledge and adhere to the ICLR Code of Ethics.

## REPRODUCIBILITY STATEMENT

We make every effort to ensure the reproducibility of the experiments in our paper. We release a codebase at `https://github.com/jw9730/flock` with training and evaluation scripts for FLOCK, including pretraining scripts and checkpoints on FB15k-237, WN18RR, and CoDEx Medium, evaluation over 54 KGs, and the synthetic dataset PETALS generator. All architectural details needed to reimplement the method, including the random-walk sampler, recording protocol, sequence processor, and consensus protocol, are specified in §A, and our theoretical claims are supported with complete proofs in §B. We additionally include further experimental details in §I.

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

## A  METHODOLOGY - DETAILS

In this section, we expand on the descriptions of individual components of FLOCK summarized in §4: the random walk algorithm, the recording protocol, the sequence processor, and the consensus protocol.

### A.1  UNIFORM RANDOM WALK

Let $G = (V, E, R)$ be a knowledge graph, and let $\ell$ be the length of random walks. For each node $v \in V$, we will denote by $\mathcal{N}(v)$ the set of neighbors of $v$:

$$\mathcal{N}(v) = \{w \in V : \exists r \in R.(v, r, w) \in E \vee (w, r, v) \in E\}$$

and by $E(v, w)$, the set of relational edges from $v$ to $w$ (allowing for the inverse direction):

$$E(v, w) = \{(v, r, w) \in R \times \{v\} \times \{w\} : (v, r, w) \in E\}$$
$$\cup \{(v, r^{-1}, w) \in R^{-1} \times \{v\} \times \{w\} : (w, r, v) \in E\}$$

where $R^{-1}$ is the set symbolizing the inverses of relation types in $R$. The uniform random walk with no backtracking $\eta(G, \ell)$ of length $\ell$ over $G$, represented as:

$$V_0 \xrightarrow{R_1} V_1 \xrightarrow{R_2} \cdots \xrightarrow{R_\ell} V_\ell$$

is a second-order Markov process that follows the rules:

$$\mathbb{P}(V_{i+2} = v \mid V_{i+1} = w, V_i = u) = \begin{cases} 0 & \text{if } v = u \text{ and } |\mathcal{N}_w| > 1 \\ 1 & \text{if } v = u \text{ and } \mathcal{N}_w = \{u\} \\ \frac{1}{|\mathcal{N}_w| - 1} & \text{if } v \neq u \text{ and } v \in \mathcal{N}_w \\ 0 & \text{if } v \notin \mathcal{N}_w \end{cases} \tag{3}$$

$$\mathbb{P}(R_{j+1} = r \mid V_{j+1} = w, V_j = u) = \begin{cases} \frac{1}{|E_{(w,u)}|} & \text{if } r(w, u) \in E_{(w,u)} \\ 0 & \text{otherwise} \end{cases}$$

for all $i \geq 0, j \geq 1$. Intuitively, at each step of the walk, we first select a neighbor (except for the vertex chosen one step ago) of the current node uniformly at random (disregarding multi-edges and edge directions), and then sample an edge between these two nodes uniformly at random. If the current node has only one neighbor, we are forced to return to it.

The initial conditions depend on the selected scenario. Given a query $q = (h, r, ?)$ over $G$, we can describe the process of selecting the first step $V_0 \xrightarrow{R_1} V_1$ as setting either (each with probability $\frac{1}{3}$):

- $V_0 = h$ and selecting the first step uniformly at random as described above, meaning:

$$\mathbb{P}(V_1 = v \mid V_0 = h) = \begin{cases} \frac{1}{|\mathcal{N}_h|} & \text{if } v \in \mathcal{N}_h \\ 0 & \text{if } v \notin \mathcal{N}_h \end{cases}$$

$$\mathbb{P}(R_1 = r \mid V_1 = w) = \begin{cases} \frac{1}{|E_{(w,h)}|} & \text{if } r(w, h) \in E_{(w,h)} \\ 0 & \text{otherwise} \end{cases}$$

- setting $R_1 = r$ and selecting $V_0 \xrightarrow{R_1} V_1$ uniformly at random from edges with type $r$.
- choosing $V_0$ uniformly at random, and then sampling the first step at random as well:

$$\mathbb{P}(V_0 = w) = \frac{1}{|V|}$$

$$\mathbb{P}(V_1 = w \mid V_0 = v) = \begin{cases} \frac{1}{|\mathcal{N}_w|} & \text{if } v \in \mathcal{N}_w \\ 0 & \text{if } v \notin \mathcal{N}_w \end{cases}$$

$$\mathbb{P}(R_1 = r \mid V_1 = v, V_0 = w) = \begin{cases} \frac{1}{|E_{(w,v)}|} & \text{if } r(w, v) \in E_{(w,v)} \\ 0 & \text{otherwise} \end{cases}$$

For the relation prediction objective, we add one more scenario, similar to the first one described above, but substituting $V_0 = t$ instead. For that problem, each scenario is chosen with probability $\frac{1}{4}$.

## A.2 RECORDING FUNCTION

Given a KG $G = (V, E, R)$, a query $q = (h_q, r_q, ?)$, a walk $\bar{\eta} = v_0 \xrightarrow{r_1} v_1 \xrightarrow{r_2} \ldots \xrightarrow{r_\ell} v_\ell$ of length $\ell$ over $G$, and a set of embeddings $\mathbf{v}$ of nodes $V$ and $\mathbf{r}$ of relations $R$, our recording function $w$ first splits the walk into a sequence of $\ell + 1$ steps:

$$(r_0, v_0), (r_1, v_1), \ldots (r_\ell, v_\ell)$$

with $r_0 = r_\varnothing$ being a special marker for no relation. Each step $(r_i, v_i)$ is transformed into a 7-tuple:

$$S_i = \big(\mathrm{id}_V(v_i; \bar{\eta}), \mathrm{id}_R(r_i; \bar{\eta}), \mathrm{dir}_i, \delta_{v_i = h_q}, \delta_{r_i = r_q}, \mathbf{v}(v_i), \mathbf{r}(r_i)\big)$$

where:

- $\mathrm{id}_V(v_i; \bar{\eta})$ and $\mathrm{id}_R(r_i; \bar{\eta})$ are the anonymized id's of the node $v_i$ and relation $r_i$, evaluated as:

$$\mathrm{id}_V(v_i; \bar{\eta}) = \arg\min_t [v_t = v_i]$$

$$\mathrm{id}_R(r_i; \bar{\eta}) = \arg\min_t \big[r_t = r_i \vee r_t = r_i^{-1}\big]$$

- $\mathrm{dir}_i$ denotes the direction in which we follow the edge. We set $\mathrm{dir}_i = 0$ if $r_i \in R$ (the edge is traversed from head to tail) and $\mathrm{dir}_i = 1$ if $r_i \in R^{-1}$ (the edge is taken in the reverse direction).
- $\delta_{v_i = h_q}$ and $\delta_{r_i \sim r_q}$ are binary flags representing whether the current node $v_i$ is the query head $v_q$ and if the relation $r_i$ is either the queried relation $r_q$ or its inverse $r_q^{-1}$.
- $\mathbf{v}(v_i), \mathbf{r}(r_i)$ are the embeddings of $v_i$ and $r_i$, respectively.

The output of $w$ for $\bar{\eta}$ given $G, q, \mathbf{v}, \mathbf{r}$ is then:

$$w(\bar{\eta}; G, q, \mathbf{v}, \mathbf{r}) = (S_0, S_1, \ldots, S_\ell)$$

## A.3 SEQUENCE PROCESSOR

Once the sampled walks are anonymized by the recording protocol $w$, the output for each walk $\bar{\eta}_i$:

$$w(\bar{\eta}; G, q, \mathbf{v}, \mathbf{r}) = (S_0, S_1, \ldots, S_\ell)$$

is passed through the sequence processor $f_\theta$, parametrized by the following modules:

- $\mathbf{A}_v, \mathbf{A}_r \in \mathbb{R}^{(\ell+1) \times d}$: embedding tables for anonymized vertices and relations, respectively,
- $\mathbf{D} \in \mathbb{R}^{2 \times d}$: look-up table for the direction embedding,
- $\mathbf{Q}_h, \mathbf{Q}_r \in \mathbb{R}^{2 \times d}$: embedding tables for the binary query labels,
- $\mathbf{V}, \mathbf{R} : \mathbb{R}^d \to \mathbb{R}^d$: linear maps applied to the passed embeddings of vertices and relations,
- $\boldsymbol{\Omega}$: a bi-directional GRU (Cho et al., 2014) cell equipped with RMSNorm (Zhang & Sennrich, 2019) and SwiGLU (Shazeer, 2020) activation function.

For each step, encoding $S_i$ of the form:

$$S_i = \big(\mathrm{id}_V(v_i; \bar{\eta}_i), \mathrm{id}_R(r_i; \bar{\eta}_i), \mathrm{dir}_i, \delta_{v_i = h_q}, \delta_{r_i = r_q}, \mathbf{v}(v_i), \mathbf{r}(r_i)\big)$$

we evaluate the processed embedding $\mathbf{c_i}$ of $S_i$ as a sum of the corresponding encoded components:

$$\begin{aligned} \mathbf{c}_i = &\mathbf{A}_v(\mathrm{id}_V(v_i; \bar{\eta}_i)) + \mathbf{A}_r(\mathrm{id}_R(r_i; \bar{\eta}_i)) + \mathbf{D}(\mathrm{dir}_i) \\ &+ \mathbf{Q_h}(\delta_{v_i = h_q}) + \mathbf{Q}_r(\delta_{r_i = r_q}) + \mathbf{V}(\mathbf{v}(v_i)) + \mathbf{R}(\mathbf{r}(r_i)) \end{aligned}$$

These are then passed to the GRU cell $\boldsymbol{\Omega}$, which fuses the features across the whole walk and produces multi-head embeddings of vertices and relations, as well as the associated weights:

$$\left(\mathbf{s}_i^{(V)}, \mathbf{s}_i^{(R)}, \mathbf{a}_i^{(V)}, \mathbf{a}_i^{(R)}\right) = \boldsymbol{\Omega}([\mathbf{c}_0, \mathbf{c}_1, \ldots, \mathbf{c}_\ell])$$

where $\mathbf{s}_i^{(V)}, \mathbf{s}_i^{(R)} \in \mathbb{R}^{(\ell+1) \times h \times d_h}$ and $\mathbf{a}_i^{(V)}, \mathbf{a}_i^{(R)} \in \mathbb{R}^{(\ell+1) \times h}$. Stacking all $N$ of them gives us the final output of the sequence processor.

### A.4 CONSENSUS PROTOCOL

Given walks $\bar{\eta}_{1:N}$ over $G = (V, E, R)$ and the outputs $\mathbf{s}^{(V)}, \mathbf{s}^{(R)}, \mathbf{a}^{(V)}, \mathbf{a}^{(R)}$ of the sequence processor, the consensus protocol $c$ aggregates the signal for each node by evaluating a weighted sum over the appearances of this node across the walks. More precisely, for each node $v \in V$, we find all pairs of indices $(i, j)$, such that the $j^{\text{th}}$ node visited in $\bar{\eta}_i$ was $v$, and concatenate the weighted sums of embeddings produced by each head, with weights exponentially proportional to the scores $\mathbf{a}^{(V)}$:

$$\Delta\mathbf{v}(v) = \bigoplus_{k=1}^{h} \frac{\displaystyle\sum_{\substack{i,j \\ \bar{\eta}_i(v_j)=v}} \exp\left(\mathbf{a}_{i,j,k}^{(V)}\right) \cdot \mathbf{s}_{i,j,k}^{(V)}}{\displaystyle\sum_{\substack{i,j \\ \bar{\eta}_i(v_j)=v}} \exp\left(\mathbf{a}_{i,j,k}^{(V)}\right)}$$

Similarly, we aggregate the encodings for relations, considering their occurrences in both directions:

$$\Delta\mathbf{r}(r) = \bigoplus_{k=1}^{h} \frac{\displaystyle\sum_{\substack{i,j \\ \bar{\eta}_i(r_j)\in\{r,r^{-1}\}}} \exp\left(\mathbf{a}_{i,j,k}^{(R)}\right) \cdot \mathbf{s}_{i,j,k}^{(R)}}{\displaystyle\sum_{\substack{i,j \\ \bar{\eta}_i(r_j)\in\{r,r^{-1}\}}} \exp\left(\mathbf{a}_{i,j,k}^{(R)}\right)}$$

In both formulas above, $\bigoplus$ denotes concatenation.

Additionally, we say that a consensus protocol $c$ is *invariant* if for any pair of isomorphic KGs $G = (V, E, R)$ and $H = (V', E', R')$, any isomorphism $\mu = (\pi, \phi)$ from $G$ to $H$, any list of embeddings $\mathbf{h}_{1:N}$ with $\mathbf{h}_i \in \mathbb{R}^d$, and any sequence of sampled walks $\bar{\eta}_{1:N}$ over $G$, the outputs

$$(\Delta\mathbf{v}, \Delta\mathbf{r}) = c(\mathbf{h}_{1:N}, \bar{\eta}_{1:N})$$
$$(\Delta\mathbf{v}', \Delta\mathbf{r}') = c(\mathbf{h}_{1:N}, \mu(\bar{\eta}_{1:N}))$$

satisfy:

$$\Delta\mathbf{v}(v) = \Delta\mathbf{v}'(\pi(v)) \qquad \forall v \in V$$
$$\Delta\mathbf{r}(r) = \Delta\mathbf{r}'(\phi(r)) \qquad \forall r \in R$$

# B PROOFS

## B.1 EXPRESSIVITY

The main proposition of this section formalizes the fact that FLOCK can approximate any link-invariant function over fixed-size knowledge graphs in probability. Intuitively, when the length of the sampled walks $\ell$ becomes higher, the probability of a single walk witnessing all the edges grows to 1. Once a walk visits all the edges, a sufficiently powerful sequence processor can derive the whole graph structure from its anonymized representation, recreating the graph in its entirety, up to isomorphism. Then, the processor can return the value of the approximated function for that graph.

We start by showing that the edge cover time $C_E(\cdot)$ of graphs in $K_{n,m}$ is bounded:

**Lemma B.1.** *Let $G \in \mathbb{K}_{n,m}$ for some $n, m$. The edge cover time $C_E(G)$ of $G$, using the algorithm from §A.1, is finite.*

*Proof.* Let $G = (V, E, R) \in \mathbb{K}_{n,m}$ be a graph. For any edge $e \in E$ and any vertex $v \in V$, let $H_v(e)$ denote the expected number of steps of the random walk algorithm $\eta$ described in §A.1. Then, the edge cover time $C_E(G)$ of $G$ with $\eta$, i.e. the expected number of steps that $\eta$ needs to take before visiting every edge in $G$, is bounded above by:

$$C_E(G) \leq \sum_{e \in E} \max_{v \in V} H_v(e) \leq m \cdot \max_{\substack{e \in E \\ v \in V}} H_v(e)$$

Indeed, consider the event of visiting all these edges in order $e_1, \ldots, e_m$:

$$
\begin{aligned}
C_E(G) &= \mathbb{E}[\text{\#steps to visit all } e_1, \ldots, e_m] \\
&\leq \mathbb{E}[\text{\#steps to visit } e_1, \text{ then } e_2, \ldots, \text{ then } e_m] \\
&\leq \mathbb{E}[\text{\#steps to visit } e_1] + \sum_{i=1}^{m-1} \mathbb{E}[\text{\#steps to visit } e_{i+1} \text{ starting from } h_i \text{ or } t_i] \\
&\leq \max_{v \in V} H_v(e_1) + \sum_{i=1}^{m-1} \max(H_{h_i}(e_{i+1}), H_{t_i}(e_{i+1})) \\
&\leq \max_{v \in V} H_v(e_1) + \sum_{i=1}^{m-1} \max_{v \in V} H_v(e_{i+1}) \\
&= \sum_{i=1}^{m} \max_{v \in V} H_v(e_i)
\end{aligned}
$$

where $h_i$ and $t_i$ are the head and tail of the edge $e_i$, respectively. Therefore, to show that $C_E(G)$ is finite, it suffices to prove that $H_v(e)$ is bounded for all $v \in V, e \in E$.

Fix $v \in V$ and $e \in E$. Consider an infinite random walk generated with $\eta$ over $G$, starting at $v$:

$$v = v_0 \xrightarrow{r_1} v_1 \xrightarrow{r_2} v_2 \xrightarrow{r_3} \ldots$$

We want to bound the expected first index $t$, such that $e$ is the edge traversed in step $v_{t-1} \xrightarrow{r_t} v_t$. Denote by $\Delta$ a maximum degree of a vertex in $G$ (counted as the number of connected vertices $\mathcal{N}(v)$), by $\rho$ the maximum number of edges between any single pair of nodes and by $d$ – the diameter of the graph, i.e. the length of the longest shortest path between two vertices (in the undirected version of $G$). Consider the series of events $A_0, A_1, \ldots$ where $A_i$ is characterized as:

$$A_i := \text{the event that starting from } v_{i(d+2)} \text{ the walk will follow a shortest path}$$
$$\text{to one of the endpoints of } e \text{ and then go through } e$$

Let $e = (h_e, r_e, t_e)$. For all values of $i$, by definition, the length of the shortest path from $v_{i(d+2)}$ to $h_e$ or $t_e$ is at most $d$. Therefore, the whole part of the walk described in $A_i$ is at most $d + 1$ steps long. By the definition of the used random walk algorithm, which only looks at the previously taken edge, we can deduce that the events $A_i$ are all mutually independent.

Moreover, let $v_{i(d+2)} = u_0 \xrightarrow{s_1} u_1 \xrightarrow{s_2} \ldots \xrightarrow{s_\ell} u_\ell \in \{h_e, t_e\}$ be a shortest path from $v_{i(d+2)}$ to one of $h_e, t_e$. Note that by minimality, there cannot be any backtracking while following this path. Therefore, the probability of the next visited node is dependent only on the value of the previous one, and we can bound the probability $P(A_i)$ of $A_i$ from below by:

$$\mathbb{P}(A_i) \geq \mathbb{P}(\text{pass through } e \text{ after reaching } h_e \text{ or } t_e) \cdot \prod_{j=0}^{\ell-1} \mathbb{P}(v_{i(d+2)+j+1} = u_{j+1} \mid v_{i(d+2)+j} = u_j)$$

The first term on the right hand side is the probability of selecting $e$ while being at $h_e$ or $t_e$, which is the probability of first selecting the other endpoint (out of at most $\Delta$ neighbors) and then picking $e$ over other edges between $h_e$ and $t_e$ (of which there is at most $\rho$). Hence:

$$\mathbb{P}(\text{pass through } e \text{ after reaching } h_e \text{ or } t_e) \geq \frac{1}{\Delta} \cdot \frac{1}{\rho} = \frac{1}{\Delta \cdot \rho}$$

As we never reach a backtracking situation by minimality of the shortest path, we can also write:

$$\mathbb{P}(v_{i(d+2)+j+1} = u_{j+1} \mid v_{i(d+2)+j} = u_j) = \frac{1}{|\mathcal{N}(v_{i(d+2)+j})|} \geq \frac{1}{\Delta}$$

Combining these observations, we can derive a bound for $\mathbb{P}(A_i)$ in terms of $\Delta, \rho$ and $d$:

$$\mathbb{P}(A_i) \geq \mathbb{P}(\text{pass through } e \text{ after reaching } h_e \text{ or } t_e) \cdot \prod_{j=0}^{\ell-1} \mathbb{P}(v_{i(d+2)+j+1} = u_{j+1} \mid v_{i(d+2)+j} = u_j)$$

$$\geq \frac{1}{\Delta \cdot \rho} \cdot \prod_{j=0}^{\ell-1} \frac{1}{\Delta} \quad \geq \quad \frac{1}{\Delta \cdot \rho} \left(\frac{1}{\Delta}\right)^\ell \quad \geq \quad \frac{1}{\Delta \cdot \rho} \left(\frac{1}{\Delta}\right)^d \quad = \quad \frac{1}{\rho \Delta^{d+1}}$$

Finally, note that if $A_i$ is true, then the first index $t$ such that $v_{t-1} \xrightarrow{r_t} v_t$ traverses $e$ is at most $(i+1)(d+2)$. We can therefore bound the expectation of such $t$, being $H_v(e) = H_{v_0}(e)$ by:

$$\begin{aligned}
H_v(e) &\leq (d+2) \cdot \mathbb{P}(A_0) + 2(d+2) \cdot \mathbb{P}(\neg A_0 \wedge A_1) + 3(d+2) \cdot \mathbb{P}(\neg A_0 \wedge \neg A_1 \wedge A_2) + \ldots \\
&= (d+2) \cdot \mathbb{P}(A_0) + 2(d+2) \cdot \mathbb{P}(\neg A_0) \cdot P(A_1) + 3(d+2) \cdot \mathbb{P}(\neg A_0) \cdot \mathbb{P}(\neg A_1) \cdot \mathbb{P}(\wedge A_2) + \ldots \\
&= (d+2) + \mathbb{P}(\neg A_0) \cdot (d+2 + \mathbb{P}(\neg A_1) \cdot (d+2 + \mathbb{P}(\neg A_2) \cdot (\ldots))) \\
&\leq (d+2) + \left(1 - \frac{1}{\rho \Delta^{d+1}}\right) \cdot \left(d+2 + \left(1 - \frac{1}{\rho \Delta^{d+1}}\right) \cdot \left(d+2 + \left(1 - \frac{1}{\rho \Delta^{d+1}}\right) \cdot (\ldots)\right)\right) \\
&= \rho(d+2)\Delta^{d+1}
\end{aligned}$$

Since $\rho \leq m, d+2 \leq n$ and $\Delta \leq n$, we have $H_v(e) \leq m(n+2)n^n$, which completes the proof. $\square$

**Remark B.2.** *The bound obtained in the proof of Lemma B.1 is very crude. In fact, we could transform the given knowledge graph into a simple graph (undirected, with no multi-edges) by substituting each edge $u \xrightarrow{r} v$ with two undirected edges $u \leftrightarrow v_{(u,r,v)} \leftrightarrow v$. The augmented graph will then have $n + m$ vertices, and our random walk algorithm naturally translates to a weighted random walk on the transformed graph. This hints at an assumption that in practice, the edge cover time of the used random walk algorithm is of the magnitude $O((n+m)^3) = O(n^3 + m^3)$.*

Let us now prove a fact about the number of distinct, up to isomorphism, graphs in $\mathbb{K}_{n,m}$.

**Lemma B.3.** *For any $n, m$, the number of isomorphism classes in $\mathbb{K}_{n,m}$ is finite.*

*Proof.* Since the number of distinct relation types that a graph in $\mathbb{K}_{n,m}$ is at most $m$, it suffices to show that the number of isomorphism classes of graphs in $\mathbb{K}_{n,m}$ with exactly $k$ relation types is bounded, for all $k \in \{1, 2, \ldots, m\}$.

Fix the number $k \in \{1, 2, \ldots, m\}$ and consider $G = (V, E, R) \in \mathbb{K}_{n,m}$ with $|R| = k$. We will show that, up to isomorphism, there are finitely many such choices of $G$. Firstly, as renaming does not change the graph structure, without loss of generality, we can assume that:

$$V = \{v_1, v_2, \ldots, v_n\} \quad \text{and} \quad R = \{r_1, r_2, \ldots, r_k\}$$

Then, there are exactly $n^2 k$ possible relational edges $e \in (V \times R \times V)$, and $E \subseteq V \times R \times V$ is a set of $m$ elements. Hence, there are $\binom{n^2 k}{m}$ possible choices of $E$, and hence, at most $\binom{n^2 k}{m}$ non-isomorphic choices of $G$. Since $k$ was chosen arbitrarily, this completes the proof. $\square$

**Lemma B.4.** *For each pair $(n, m)$, there exists a number $C_{n,m}$ such that the edge cover time, using the algorithm from §A.1, of any knowledge graph in $\mathbb{K}_{n,m}$ is at most $C_{n,m}$.*

*Proof.* The result follows from Lemmas B.1 and B.3. As two isomorphic graphs have identical cover time, we can set $C_{n,m}$ to be the maximum of cover times of representatives of all isomorphic classes, which, by finiteness of both, is well-defined. □

**Lemma B.5.** *Let $G \in \mathbb{K}_{n,m}$ be a graph, $q = (h_q, r_q, ?)$ be a link query over $G$, and $\bar{\eta}$ be a walk over $G$. If $\bar{\eta}$ traverses all edges of $G$, then using only the output $w(\bar{\eta}; G, q, \cdot, \cdot)$ of the recording function $w$ detailed in §A.2, we can construct a graph-query pair $(H, q')$ isomorphic to $(G, q)$.*

*Proof.* Suppose that $\bar{\eta} = v_0 \xrightarrow{r_1} v_1 \xrightarrow{r_2} \ldots \xrightarrow{r_\ell} v_\ell$ visits all edges of $G = (V, E, R)$ and let $\ell$ be its length. Recall the anonymization functions $\mathrm{id}_V(\cdot; \bar{\eta})$ and $\mathrm{id}_R(\cdot; \bar{\eta})$ as defined in §A.2. The output $w(\bar{\eta}; G, q, \cdot, \cdot)$ (the embedding functions provided as the last two arguments are irrelevant) is a sequence of tuples $S_0, S_1, \ldots, S_\ell$ with each $S_i$ equal to:

$$S_i = \left( \mathrm{id}_V(v_i; \bar{\eta}), \mathrm{id}_R(r_i; \bar{\eta}), \mathrm{dir}_i, \delta_{v_i = h_q}, \delta_{r_i = r_q}, \cdot, \cdot \right)$$

Consider a graph $H = (V', E', R')$ constructed as follows:

- the vertices $V'$ correspond to the anonymized node ids $\mathrm{id}_V(v_i; \bar{\eta})$:

$$V' = \{ \mathrm{id}_V(v; \bar{\eta}) \mid v \in V \}$$

  Since each vertex must have been visited by $\bar{\eta}$, this is well-defined.

- the relation types $R'$ are the anonymized relation ids $\mathrm{id}_R(r_i; \bar{\eta})$:

$$R' = \{ \mathrm{id}_V(r; \bar{\eta}) \mid r \in R \}$$

  Again, this is well-defined, as each relation must have been noticed by $\bar{\eta}$.

- the edges $E'$ are reconstructed from the consecutive step encodings using the anonymized vertex and relation indices and the direction $\mathrm{dir}_i$:

$$E' = \{ (\mathrm{id}_V(v_{i-1}), \mathrm{id}_R(r_i), \mathrm{id}_V(v_i)) \mid \mathrm{dir}_i = 0, 1 \le i \le l \}$$
$$\cup \{ (\mathrm{id}_V(v_i), \mathrm{id}_R(r_i), \mathrm{id}_V(v_{i-1})) \mid \mathrm{dir}_i = 1, 1 \le i \le l \}$$

and a query $q' = (\mathrm{id}_V(v_i; \bar{\eta}), \mathrm{id}_R(r_j; \bar{\eta}), ?)$ for $i, j$ such that $\delta_{v_i = h_q} = 1$ and $\delta_{r_j = r_q} = 1$.

Then by the definition of $w$ (§A.2), it is straightforward to check that the pair $(\mathrm{id}_V(\cdot; \bar{\eta}), \mathrm{id}_R(\cdot; \bar{\eta}))$ defines an isomorphism from $(G, q)$ to $(H, q')$. Indeed, both these functions are injective by construction, and as $\bar{\eta}$ witnesses all nodes and relations, they are well-defined bijections. For each unique edge traversed by $\bar{\eta}$, there exists a unique edge in $E'$ translated to the anonymized space, which implies an isomorphism between $E$ and $E'$. Finally, by utilizing the flags $\delta_{v_i = h_q}$ and $\delta_{r_j = r_q}$, we can identify the query head node and relation in the new graph. All things considered, we can reconstruct the pair $(G, q)$, up to isomorphism, from the output of $w(\bar{\eta}; G, q, \cdot, \cdot)$. □

We are now ready to prove the main result regarding the universality of FLOCK as an approximation of link invariant functions. The outline of the proof is as follows: 1) Using the upper-bound on the edge cover time of graphs in $\mathbb{K}_{n,m}$ derived in Lemma B.4, we can bound the probability of sampling a walk that visits all edges, 2) Once such a walk is sampled, we can recover the graph and query, up to isomorphism, from its anonymized form (Lemma B.5), 3) Lastly, we can return the value of the approximated function for the derived isomorphic instance. Since the approximated function is link invariant, if the reconstructed graph matches the original one, we return precisely the correct value.

**Proposition 4.1.** *With a powerful enough sequence processor $f_\theta$, the FLOCK framework described in §4 is a universal approximator of link invariant functions over $\mathbb{K}_{n,m}$ for all pairs $(n, m)$.*

*Proof.* Let $\varphi : \mathbb{K}_{n,m} \to (V \times R \times V \to [0, 1])$ be a link invariant function over $\mathbb{K}_{n,m}$ returning values from the interval $[0, 1]$. Let $G = (V, E, R) \in \mathbb{K}_{n,m}$, $q = (h, r, ?)$ be a link prediction query over $G$ and $t \in V$ be a target node. Pick some $\epsilon, \delta > 0$. Our goal is to show that:

$$\mathbb{P}(|\varphi(G)((h, r, t)) - X_\theta(G, (h, r, ?))(t)| < \epsilon) > 1 - \delta \tag{4}$$

For simplicity, let us consider a situation where only a single walk $\bar{\eta}$ of length $\ell$ is sampled by the model (otherwise, omit additional walks). We will also restrict the argument to a single refinement case – the result can be extended to multiple refinement steps by returning $\Delta\mathbf{v}, \Delta\mathbf{r} = 0$ during all additional iterations. Consider a sequence processor $f_\theta$ that given the output $w(\bar{\eta}; G, q, \cdot, \cdot)$ of the recording protocol, creates a graph-query pair $(H, q')$ with $q' = (h_{q'}, r_{q'}, ?)$ using the strategy described in the proof of Lemma B.5, and returns a vector $\mathbf{h} \in \mathbb{R}^{l+1}$ whose $i^{\text{th}}$ entry is equal $\mathbf{h}_i = \varphi(H)((h_{q'}, r_{q'}, \text{id}_V(v_i; \bar{\eta}))$ where $v_i$ is the $i^{\text{th}}$ node visited by $\bar{\eta}$. The consensus protocol $c$, provided $t$ was visited by $\bar{\eta}$, can then identify $t$ as one of the $v_j$ and pull the corresponding embedding $\mathbf{h}_j = \varphi(H)((h_{q'}, r_{q'}, \text{id}_V(t; \bar{\eta}))$, returning it as the output $\mathbf{v}(t) = \mathbf{h}_j$ (note that no matter which specific value of $j$ is chosen, this value will be the same). Finally, the classification head can work as an identity operation, returning $X_\theta(G, q)(t) = \mathbf{v}(t) = \varphi(H)((h_{q'}, r_{q'}, \text{id}_V(t; \bar{\eta}))$.

We claim that if the sampled walk $\bar{\eta}$ traverses all edges of $G$, then the output of the FLOCK model described above satisfies:

$$\varphi(G)((h, r, t)) = X_\theta(G, (h, r, ?))(t)$$

By Lemma B.5, in such case, the reconstructed pair $(H, q')$ is isomorphic to $(G, q)$ by the isomorphism $id = (\text{id}_V(\cdot; \bar{\eta}), \text{id}_R(\cdot; \bar{\eta}))$. Since $\varphi$ is link invariant, we can write:

$$\begin{aligned}
\varphi(G)((h, r, t)) &= \varphi(\text{id}(G))((\text{id}_V(h; \bar{\eta}), \text{id}_R(r; \bar{\eta}), \text{id}_V(t; \bar{\eta}))) \\
&= \varphi(H)((h_{q'}, r_{q'}, \text{id}_V(t; \bar{\eta}))) \\
&= X_\theta(G, (h, r, ?))(t)
\end{aligned}$$

Therefore, whenever the walk $\bar{\eta}$ witnesses all edges of $G$, the output of the FLOCK model satisfies:

$$\varphi(G)((h, r, t)) = X_\theta(G, (h, r, ?))(t)$$

Hence, to show (4), it suffices to prove that we can uniformly choose the length $\ell$ of the random walk so that the probability of $\bar{\eta}$ covering all the edges is greater than $1 - \delta$. By Markov's inequality:

$$\begin{aligned}
\mathbb{P}(\bar{\eta} \text{ does not cover all edges}) &= \mathbb{P}(\text{it takes } > \ell \text{ steps for } \eta \text{ to cover edges of } G) \\
&\leq \frac{\mathbb{E}[\#\text{steps such that } \eta \text{ covers all edges of } G]}{\ell} \\
&= \frac{C_E(G)}{\ell}
\end{aligned}$$

But by Lemma B.4, $C_E(G) \leq C_{n,m}$ for some constant $C_{n,m}$. Hence, taking $\ell > \frac{C_{n,m}}{\delta}$, we get:

$$\mathbb{P}(\bar{\eta} \text{ does not cover all edges}) \leq \frac{C_E(G)}{\ell} \leq \frac{C_{n,m}}{\ell} < \delta$$

This means that for such a choice of $\ell$:

$$\mathbb{P}(\bar{\eta} \text{ witnesses all edges of } G) > 1 - \delta$$

which leads to the conclusion that for $\ell > \frac{C_{n,m}}{\delta}$, the proposed FLOCK framework satisfies:

$$\mathbb{P}(|\varphi(G)((h, r, t)) - X_\theta(G, (h, r, ?))(t)| < \epsilon) > 1 - \delta$$

for any choice of $G = (V, E, R) \in \mathbb{K}_{n,m}$ and $(h, r, t) \in V \times R \times V$. $\qquad\square$

## B.2 INVARIANCE

First, let us recall the definition of invariance for the context of knowledge graphs and the associated notion of invariance in probability. We say that a function $\varphi$ taking KGs as input is invariant if for any pair of isomorphic KGs $G \simeq H$ it produces the same input, i.e. $G \simeq H \implies \varphi(G) = \varphi(H)$.

We extend the notion of invariance for further types of inputs, not limited to full knowledge graphs, particularly to random walks and link prediction queries. Let $G = (V, E, R) \in \mathbb{K}_{n,m}$ and let $H = (V', E', R') \simeq G$ be a KG isomorphic to $G$ via the isomorphism $\mu = (\pi, \phi)$. For any $h \in V$ and $r \in R$, we identify the link prediction query $q = (h, r, ?)$ in $H$ using the isomorphism $\mu$ as:

$$\mu(q) = \mu((h, r, ?)) = (\pi(h), \phi(r), ?)$$

Similarly, let $\eta = v_0 \xrightarrow{r_1} \ldots \xrightarrow{r_\ell} v_\ell$ be a walk of length $\ell$ in $G$. The view of $\eta$ with $\mu$ is defined as:

$$\mu\left(v_0 \xrightarrow{r_1} v_1 \xrightarrow{r_2} \ldots \xrightarrow{r_\ell} v_\ell\right) = \pi(v_0) \xrightarrow{\phi(r_1)} \pi(v_1) \xrightarrow{\phi(r_2)} \ldots \xrightarrow{\phi(r_\ell)} \pi(v_\ell)$$

Let $f$ be a function taking inputs drawn from KGs. We call $f$ invariant if for any pair of isomorphic graphs $G \overset{\mu}{\simeq} H$ and an associated isomorphism $\mu = (\pi, \phi)$, $f$ satisfies

$$f(x) = f(\mu(x))$$

where $x$ can be, e.g., a walk or link prediction query. In words, invariance means that the function preserves output under the re-identifications of the input graph and the induced transformations of queries and walks.

This notion extends to functions with multiple inputs, where we enforce the transformation on each graph-related input. For example, a function $\varphi$ taking a KG, query and a $d$-dimensional vector is invariant if it satisfies:

$$\forall G \overset{\mu}{\simeq} H, q, \mathbf{v} \in \mathbb{R}^d : \quad \varphi(G, q, \mathbf{v}) = \varphi(\mu(G), \mu(q), \mathbf{v})$$

Following the definition of *invariance in probability*, provided in §3, we extend all the definitions above to the stochastic case, replacing equality ($=$) with equality in distribution ($\overset{d}{=}$).

We can now prove the main propositions stated in §4.2. Let's begin with the more general:

**Proposition 4.2.** *Suppose that the walk sampling protocol $\eta$ is invariant in probability and both the recording protocol $w$ and the consensus protocol $c$ are invariant. Then, regardless of the choice of the deterministic sequence processor $f_\theta$, the corresponding* FLOCK *model is invariant in probability.*

*Proof.* Let $(V, E, R) = G \simeq H = (V', E', R')$ be isomorphic knowledge graphs with isomorphism $\mu = (\pi, \phi)$ transforming $G$ into $H$. Our goal is to show that when the statement conditions are met for a FLOCK model $X_\theta$ with $I$ refinement steps, then for any link prediction query $q = (h, r, ?)$ and any target node $t \in V$, the prediction of FLOCK for $t$ over $(G, q)$ is an identical random variable to the prediction for $\pi(t)$ over $(H, \mu(q))$, i.e.

$$X_\theta(G, q)(t) \overset{d}{=} X_\theta(H, \mu(q))(\pi(t))$$

where $\mu(q) = (\pi(h), \phi(r), ?)$. Recall that these predictions are defined as:

$$X_\theta(G, q)(t) := \text{head}(\mathbf{v}^{(I)}(t) + \mathbf{r}^{(I)}(r))$$

$$X_\theta(H, \mu(q))(\pi(t)) := \text{head}(\mathbf{v}'^{(I)}(\pi(t)) + \mathbf{r}'^{(I)}(\phi(r)))$$

As head is a deterministic map, it suffices to show that the final embeddings $\mathbf{v}^{(I)}, \mathbf{r}^{(I)}$ for $(G, q)$ and $\mathbf{v}'^{(I)}, \mathbf{r}'^{(I)}$ for $(H, \mu(q))$ satisfy:

$$\mathbf{v}^{(I)}(v) \overset{d}{=} \mathbf{v}'^{(I)}(\pi(v)) \quad \text{and} \quad \mathbf{r}^{(I)}(r) \overset{d}{=} \mathbf{r}'^{(I)}(\phi(r)) \qquad \forall v \in V, r \in R$$

We will prove this result by induction on the number of layers $i$. The base case $i = 0$ is trivial, as we initialize the embeddings of all nodes with a pretrained vector $\mathbf{v}_0$, and all relations with $\mathbf{r}_0$.

For the induction step, suppose the claim holds for $i$. We drop the superscript $(i)$ for readability. The result for $i + 1$ becomes apparent by unfolding the definitions of invariance of the considered components. Since $\eta$ is invariant in probability, we have

$$\mu(\eta(G)) \overset{d}{=} \eta(H) \tag{5}$$

Let $\eta_1, \ldots, \eta_n$ be the random walks over $G$ using $\eta$ and $\eta'_1, \ldots, \eta'_n$ be random walks over $H$. Now, $\eta_1, \ldots, \eta_n$ are independent and identically distributed random variables, each following the distribution $\eta_j \sim \eta(G)$. Similarly, using (5):

$$\eta'_j \sim \eta(H) \overset{d}{=} \mu(\eta(G)) \implies \eta'_j \overset{d}{=} \mu(\eta_j) \tag{6}$$

As the recording protocol $w$ is invariant, $w(\eta_j) = w(\mu(\eta_j))$ for all $j$, which with (6) yields:

$$\mathbf{z}_j := w(\eta_j) = w(\mu(\eta_j)) \overset{d}{=} w(\eta'_j) := \mathbf{z'_j} \tag{7}$$

Then, $f_\theta$ is a deterministic map, so (7) implies:

$$\mathbf{h}_j := f_\theta(\mathbf{z}_j) \stackrel{d}{=} f_\theta(\mathbf{z}'_j) := \mathbf{h}'_j$$

Let $(\Delta\mathbf{v}, \Delta\mathbf{r}) = c(\mathbf{h}_{1:N}, \eta_{1:N})$, $(\Delta\mathbf{v}', \Delta\mathbf{r}') = c(\mathbf{h}'_{1:N}, \eta'_{1:N})$ be the outputs of the consensus protocol. We will denote by $c_\mathbf{v}$ and $c_\mathbf{r}$, the restrictions to the first and second output, e.g. $\Delta\mathbf{v} = c_\mathbf{v}(\mathbf{h}_{1:N}, \eta_{1:N})$. Let $\mathbf{x} \in \mathbb{R}^d$ be a vector, and denote by $\mathcal{W}(G)$ the space of walks over $G$. For any vertex $v \in V$, the probability that $\Delta\mathbf{v}(v) = \mathbf{x}$ equals:

$$
\begin{aligned}
\mathbb{P}(\Delta\mathbf{v}(v) = \mathbf{x}) &= \sum_{\bar{\eta} \in \mathcal{W}(G)^n} \mathbb{P}(\Delta\mathbf{v}(v) = \mathbf{x} | \eta_{1:N} = \bar{\eta}) \cdot \mathbb{P}(\eta_{1:N} = \bar{\eta}) \\
&= \sum_{\bar{\eta} \in \mathcal{W}(G)^n} \mathbb{P}(c_\mathbf{v}(\mathbf{h}_{1:N}, \eta_{1:N}) = \mathbf{x} | \eta_{1:N} = \bar{\eta}) \cdot \mathbb{P}(\eta_{1:N} = \bar{\eta}) \\
&= \sum_{\bar{\eta} \in \mathcal{W}(G)^n} \mathbb{P}(c_\mathbf{v}(f_\theta(w(\eta_{1:N})), \eta_{1:N}) = \mathbf{x} | \eta_{1:N} = \bar{\eta}) \cdot \mathbb{P}(\eta_{1:N} = \bar{\eta}) \\
&= \sum_{\bar{\eta} \in \mathcal{W}(G)^n} \mathbb{P}(c_\mathbf{v}(f_\theta(w(\bar{\eta})), \bar{\eta}) = \mathbf{x} | \eta_{1:N} = \bar{\eta}) \cdot \mathbb{P}(\eta_{1:N} = \bar{\eta}) \\
&= \sum_{\substack{\bar{\eta} \in \mathcal{W}(G)^n \\ c_\mathbf{v}(f_\theta(w(\bar{\eta})), \bar{\eta})(v) = \mathbf{x}}} \mathbb{P}(\eta_{1:N} = \bar{\eta})
\end{aligned}
$$

Similarly, we can derive:

$$\mathbb{P}(\Delta\mathbf{v}'(\pi(v)) = \mathbf{x}) = \sum_{\substack{\bar{\eta}' \in \mathcal{W}(H)^n \\ c_\mathbf{v}(f_\theta(w(\bar{\eta}')), \bar{\eta}')(\pi(v)) = \mathbf{x}}} \mathbb{P}(\eta'_{1:N} = \bar{\eta}')$$

Using the invariance of the consensus protocol and the invariance of $f_\theta \circ w$, we can write:

$$
\begin{aligned}
c_\mathbf{v}(f_\theta(w(\bar{\eta}')), \bar{\eta}')(\pi(v)) &= c_\mathbf{v}(f_\theta(w(\mu(\bar{\eta}))), \mu(\bar{\eta}))(\pi(v)) \\
&= c_\mathbf{v}(f_\theta(w(\bar{\eta})), \mu(\bar{\eta}))(\pi(v)) \\
&= c_\mathbf{v}(f_\theta(w(\bar{\eta})), \bar{\eta})(v)
\end{aligned}
$$

The graph isomorphism $\mu$ defines a bijection between walks $\mathcal{W}(G)$ in $G$ and walks $\mathcal{W}(H)$ in $H$, so we can use this correspondence to deduce:

$$
\begin{aligned}
\mathbb{P}(\Delta\mathbf{v}'(\pi(v)) = \mathbf{x}) &= \sum_{\substack{\bar{\eta}' \in \mathcal{W}(H)^n \\ c_\mathbf{v}(f_\theta(w(\bar{\eta}')), \bar{\eta}')(\pi(v)) = \mathbf{x}}} \mathbb{P}(\eta'_{1:N} = \bar{\eta}') \\
&= \sum_{\substack{\mu(\bar{\eta}) \in \mathcal{W}(H)^n \\ c_\mathbf{v}(f_\theta(w(\mu(\bar{\eta}))), \mu(\bar{\eta}))(\pi(v)) = \mathbf{x}}} \mathbb{P}(\eta'_{1:N} = \mu(\bar{\eta})) \\
&= \sum_{\substack{\bar{\eta} \in \mathcal{W}(G)^n \\ c_\mathbf{v}(f_\theta(w(\bar{\eta})), \bar{\eta})(v) = \mathbf{x}}} \mathbb{P}(\eta'_{1:N} = \mu(\bar{\eta}))
\end{aligned}
\tag{8}
$$

Since $\eta$ is invariant in probability, $\mathbb{P}(\eta_{1:N} = \bar{\eta}) = \mathbb{P}(\eta'_{1:N} = \mu(\bar{\eta}))$. Applying this to (8) yields:

$$
\begin{aligned}
\mathbb{P}(\Delta\mathbf{v}'(\pi(v)) = \mathbf{x}) &= \sum_{\substack{\bar{\eta} \in \mathcal{W}(G)^n \\ c_\mathbf{v}(f_\theta(w(\bar{\eta})), \bar{\eta})(v) = \mathbf{x}}} \mathbb{P}(\eta'_{1:N} = \mu(\bar{\eta})) \\
&= \sum_{\substack{\bar{\eta} \in \mathcal{W}(G)^n \\ c_\mathbf{v}(f_\theta(w(\bar{\eta})), \bar{\eta})(v) = \mathbf{x}}} \mathbb{P}(\eta_{1:N} = \bar{\eta}) = \mathbb{P}(\Delta\mathbf{v}(v) = \mathbf{x})
\end{aligned}
$$

As $\mathbf{x}$ was chosen arbitrarily, we can conclude that $\Delta\mathbf{v}(v) \stackrel{d}{=} \Delta\mathbf{v}'(\pi(v))$. The proof for relations follows analogously, considering $c_{\mathbf{r}}$ instead of $c_{\mathbf{v}}$. This allows us to write:

$$
\begin{aligned}
\Delta\mathbf{v}(v) &\stackrel{d}{=} \Delta\mathbf{v}'(\pi(v)) && \forall v \in V \\
\Delta\mathbf{r}(r) &\stackrel{d}{=} \Delta\mathbf{r}'(\phi(r)) && \forall r \in R
\end{aligned}
\tag{9}
$$

By the induction hypothesis, $\mathbf{v}^{(i)}(v) \stackrel{d}{=} \mathbf{v}'^{(i)}(\pi(v))$ for all $v \in V$ and $\mathbf{r}^{(i)}(r) \stackrel{d}{=} \mathbf{r}'^{(i)}(r)$ for all $r \in R$. Therefore, by (9), combined with properties of sums of random variables:

$$
\mathbf{v}^{(i+1)}(v) := \mathbf{v}^{(i)}(v) + \Delta\mathbf{v}(v) \stackrel{d}{=} \mathbf{v}'^{(i)}(\pi(v)) + \Delta\mathbf{v}'(\pi(v)) := \mathbf{v}'^{(i+1)}(\pi(v)) \qquad \forall v \in V
$$

$$
\mathbf{r}^{(i+1)}(r) := \mathbf{r}^{(i)}(r) + \Delta\mathbf{r}(r) \stackrel{d}{=} \mathbf{r}'^{(i)}(\phi(r)) + \Delta\mathbf{r}'(\phi(r)) := \mathbf{r}'^{(i+1)}(\phi(r)) \qquad \forall r \in R
$$

which completes the induction step, and hence the proof. $\qquad\square$

We can use the conclusion from Proposition 4.2 to prove the probabilistic invariance of the architecture proposed in §4. To be able to apply it, we first need to verify the invariance of all used components, which we formalize in the following lemmas.

**Lemma B.6.** *The choice of the first step $v_0 \xrightarrow{r_1} v_1$ of the uniform random walk algorithm described in §A.1 is invariant.*

*Proof.* Let $G = (V, E, R)$ be a graph and let $H \simeq G$ be an isomorphic graph, with the isomorphism $\mu = (\pi, \phi)$ taking $G$ to $H$. Consider a link prediction query $q = (h, r, ?)$ over $G$, and its identification $q' = \mu(q) = (\pi(h), \phi(r), ?)$. The goal is to show that when using $\eta$ described in §A.1 for $(G, q)$ and $(H, q')$, the first steps:

$$
V_0 \xrightarrow{R_1} V_1 \qquad \text{and} \qquad U_0 \xrightarrow{S_1} U_1
$$

of the execution of $\eta$ over $G$ and $H$, respectively, satisfy the following property:

$$
\pi(V_0) \xrightarrow{\phi(R_1)} \pi(V_1) \stackrel{d}{=} U_0 \xrightarrow{S_1} U_1
$$

By definition of $\eta$, there are three scenarios of choosing the first step, each with probability $\frac{1}{3}$. Hence, it suffices to show that within each scenario, the selection process is invariant in probability:

- **Scenario 1:** selecting the query head as the first node, then proceeding by random. First, $\pi$ takes the head node of $q$ to the head node of $q'$. Secondly, as isomorphisms preserve the number of neighboring nodes and number of edges between a pair of nodes, we have:

$$
\mathbb{P}(V_1 = v \mid V_0 = h) = \begin{cases} \frac{1}{|\mathcal{N}_h|} & \text{if } v \in \mathcal{N}_h \\ 0 & \text{if } v \notin \mathcal{N}_h \end{cases}
$$

$$
= \begin{cases} \frac{1}{|\mathcal{N}_{\pi(h)}|} & \text{if } \pi(v) \in \mathcal{N}_{\pi(h)} \\ 0 & \text{if } \pi(v) \notin \mathcal{N}_{\pi(h)} \end{cases} = \mathbb{P}(U_1 = \pi(v) \mid U_0 = \pi(h))
$$

  and

$$
\mathbb{P}(R_1 = r \mid V_1 = w) = \begin{cases} \frac{1}{|E_{(w,h)}|} & \text{if } r(w, h) \in E_{(w,h)} \\ 0 & \text{otherwise} \end{cases}
$$

$$
= \begin{cases} \frac{1}{|E_{(\pi(w),\pi(h))}|} & \text{if } \phi(r)(\pi(w), \pi(h)) \in E_{(\pi(w),\pi(h))} \\ 0 & \text{otherwise} \end{cases}
$$

$$
= \mathbb{P}(S_1 = \phi(r) \mid U_1 = \pi(w))
$$

- **Scenario 2:** selecting an edge with query relation type at random. Here, we use the fact that isomorphisms preserve the number of edges of a given type. Hence, $\mu$ defines a bijection between the sets of edges with type $r$ in $G$ and type $\phi(r)$ in $H$, which allows us to conclude that this scenario is also invariant in probability.

- **Scenario 3:** selecting the first step completely at random. This case is similar to Scenario 1 – using the invariance of the number of neighboring nodes under isomorphism, we can repeat similar calculations in a straightforward manner to show probabilistic invariance.

Either way, we find that the selection process of the first step of $\eta$ over $G$ translates naturally via $\mu$ to the choice of the first step over $H$, proving the desired statement. $\square$

**Lemma B.7.** *Suppose that the first step $v_0 \xrightarrow{r_1} v_1$ is chosen in an invariant manner. Then, the uniform random walk with no backtracking algorithm $\eta$ is invariant in probability.*

*Proof.* Let $G = (V, E, R)$ be a knowledge graph, and let $\ell$ be the length of random walks. Let $H$ be a KG isomorphic to $G$ via the isomorphism $\mu = (\pi, \phi)$. We aim to show that:

$$\mu(\eta(G, \ell)) = \pi(V_0) \xrightarrow{\phi(R_1)} \pi(V_1) \xrightarrow{\phi(R_2)} \dots \xrightarrow{\phi(R_\ell)} \pi(V_\ell) \overset{d}{=} U_0 \xrightarrow{S_1} U_1 \xrightarrow{S_2} \dots \xrightarrow{S_\ell} U_\ell = \eta(H, \ell)$$

Let $\bar{\eta} = v_0 \xrightarrow{r_1} v_1 \xrightarrow{r_2} \dots \xrightarrow{r_\ell} v_\ell \in \mathcal{W}(G)$ be a walk of length $\ell$ over $G$. It suffices to show that the probability of sampling $\bar{\eta}$ from $G$ is identical to the probability of sampling $\mu(\bar{\eta})$ from $H$:

$$\mathbb{P}(\eta(G, \ell) = \bar{\eta}) = \mathbb{P}(\eta(H, \ell) = \mu(\bar{\eta}))$$

To see this, let us expand the definitions of $\mathbb{P}(\eta(G, \ell) = \bar{\eta})$:

$$
\begin{aligned}
\mathbb{P}(\eta(G, \ell) = \bar{\eta}) = &\mathbb{P}(V_0 = v_0) \\
&\cdot \mathbb{P}(V_1 = v_1 \mid V_0 = v_0) \\
&\cdot \prod_{i=0}^{\ell-2} \mathbb{P}(V_{i+2} = v_{i+2} \mid V_{i+1} = v_{i+1}, V_i = v_i) \\
&\cdot \prod_{i=0}^{\ell-1} \mathbb{P}(R_{i+1} = r_{i+1} \mid V_{i+1} = v_{i+1}, V_i = v_i)
\end{aligned}
\tag{10}
$$

and $P(\eta(H, \ell) = \mu(\bar{\eta}))$:

$$
\begin{aligned}
\mathbb{P}(\eta(H, \ell) = \mu(\bar{\eta})) = &\mathbb{P}(U_0 = \pi(v_0)) \\
&\cdot \mathbb{P}(U_1 = \pi(v_1) \mid U_0 = \pi(v_0)) \\
&\cdot \prod_{i=0}^{\ell-2} \mathbb{P}(U_{i+2} = \pi(v_{i+2}) \mid U_{i+1} = \pi(v_{i+1}), U_i = \pi(v_i)) \\
&\cdot \prod_{i=0}^{\ell-1} \mathbb{P}(S_{i+1} = \phi(r_{i+1}) \mid U_{i+1} = \pi(v_{i+1}), U_i = \pi(v_i))
\end{aligned}
\tag{11}
$$

Given that the graph isomorphism preserves the number of neighbors for each node and is a bijection, we can easily verify using the definitions from (3) that the following indeed hold:

$$
\begin{aligned}
\mathbb{P}(V_{i+2} = v_{i+2} \mid V_{i+1} = v_{i+1}, V_i = v_i) &= \mathbb{P}(U_{i+2} = \pi(v_{i+2}) \mid U_{i+1} = \pi(v_{i+1}), U_i = \pi(v_i)) \\
\mathbb{P}(R_{j+1} = r_{j+1} \mid V_{j+1} = v_{j+1}, V_j = v_j) &= \mathbb{P}(S_{j+1} = \phi(r_{j+1}) \mid U_{j+1} = \pi(v_{j+1}), U_j = \pi(v_j))
\end{aligned}
\tag{12}
$$

for all $i \in \{0, 1, \dots, \ell - 2\}, j \in \{1, \dots, \ell - 1\}$. Moreover, by the assumption that the first step $V_0 \xrightarrow{R_1} V_1$ is invariant, we have:

$$\mathbb{P}((V_0, R_1, V_1) = (v_0, r_1, v_1)) = \mathbb{P}((U_0, S_1, U_1) = (\pi(v_0), \phi(r_1), \pi(v_1))) \tag{13}$$

But by the laws of conditional probability:

$$
\begin{aligned}
\mathbb{P}((V_0, R_1, V_1) = (v_0, r_1, v_1)) &= \mathbb{P}(R_1 = r_1 \mid V_0 = v_0, V_1 = v_1) \cdot \mathbb{P}(V_0 = v_0, V_1 = v_1) \\
&= \mathbb{P}(R_1 = r_1 \mid V_0 = v_0, V_1 = v_1) \cdot \mathbb{P}(V_1 = v_0 \mid V_0 = v_0) \cdot \mathbb{P}(V_0 = v_0)
\end{aligned}
$$

and analogously:

$$
\begin{aligned}
\mathbb{P}((U_0, S_1, U_1) &= (\pi(v_0), \phi(r_1), \pi(v_1))) \\
&= \mathbb{P}(S_1 = \phi(r_1) \mid U_0 = \pi(v_0), U_1 = \pi(v_1)) \cdot \mathbb{P}(U_1 = \pi(v_0) \mid U_0 = \pi(v_0)) \cdot \mathbb{P}(U_0 = \pi(v_0))
\end{aligned}
$$

Substituting these equalities into (13) and multiplying both sides by the equalities from (12) for all choices of $i \in \{0, 1, \dots, \ell - 2\}, j \in \{1, \dots, \ell - 1\}$, we get precisely the equality of the right sides of (10) and (11). Hence,

$$\mathbb{P}(\eta(G, \ell) = \bar{\eta}) = \mathbb{P}(\eta(H, \ell) = \mu(\bar{\eta}))$$

and we can conclude that $\mu(\eta(G, \ell)) \overset{d}{=} \eta(H, \ell)$, and the algorithm $\eta$ is invariant in probability. $\square$

**Corollary B.8.** *The random walk algorithm presented in §A.1 is invariant in probability.*

**Lemma B.9.** *The recording protocol $w$, as described in §A.2, is invariant, provided that the embedding functions $\mathbf{v}$ and $\mathbf{r}$ are invariant.*

*Proof.* Let $G = (V, E, R)$ and $H = (V', E', R')$ be isomorphic knowledge graphs with the isomorphism $\mu = (\pi, \phi)$ taking $G$ to $H$. Let $q = (h_q, r_q, ?)$ be a link prediction query over $G$, and $\mu(q) = (\pi(h_q), \phi(r_q), ?)$ be its identification in $H$. Let $\bar{\eta} = v_0 \xrightarrow{r_1} v_1 \xrightarrow{r_2} \ldots \xrightarrow{r_\ell} v_\ell \in \mathcal{W}(G)$ be a walk over $G$, and $\bar{\eta}' = \mu(\bar{\eta}) = \pi(v_0) \xrightarrow{\phi(r_1)} \pi(v_1) \xrightarrow{\phi(r_2)} \ldots \xrightarrow{\phi(r_\ell)} \pi(v_\ell)$ be the analogous walk over $H$. To prove that the recording protocol $w$ outlined in §A.2 is invariant, it suffices to show that the encoding of each step:

$$S_i = \big(\mathrm{id}_V(v_i; \bar{\eta}), \mathrm{id}_R(r_i; \bar{\eta}), \mathrm{dir}_i, \delta_{v_i = h_q}, \delta_{r_i = r_q}, \mathbf{v}(v_i), \mathbf{r}(r_i)\big)$$

is identical for $\bar{\eta}$ and $\bar{\eta}'$. We will show this for each component:

- since $\pi$ defines a bijection between nodes in $G$ and $H$, for any $i$, we have:

$$\mathrm{id}_V(v_i; \bar{\eta}) = \arg\min_t [v_t = v_i] = \arg\min_t [\pi(v_t) = \pi(v_i)] = \mathrm{id}_V(\pi(v_i); \bar{\eta}')$$

- similarly to the point above, $\phi$ is a bijection between relations of $G$ and $H$, so we can write:

$$\begin{aligned}
\mathrm{id}_R(r_i; \bar{\eta}) &= \arg\min_t \big[r_t = r_i \vee r_t = r_i^{-1}\big] \\
&= \arg\min_t \big[\phi(r_t) = \phi(r_i) \vee \phi(r_t) = \phi(r_i)^{-1}\big] \\
&= \mathrm{id}_R(\phi(r_i); \bar{\eta}')
\end{aligned}$$

- $\mathrm{dir}_i$ is clearly preserved, as the isomorphism $\mu$ preserves the directions of edges,

- as $\pi, \phi$ are bijections the masks $\delta_{v_i = h_q}, \delta_{r_i = r_q}$, representing whether the $i$'th node and relation match the types in the query, satisfy:

$$\begin{aligned}
v_i = h_q &\iff \pi(v_i) = \pi(h_q) &\implies& \quad \delta_{v_i = h_q} = \delta_{\pi(v_i) = \pi(h_q)} \\
r_i = r_q &\iff \phi(r_i) = \phi(r_q) &\implies& \quad \delta_{r_i = r_q} = \delta_{\phi(r_i) = \phi(r_q)}
\end{aligned}$$

- $\mathbf{v}$ and $\mathbf{r}$ are invariant by assumption, so:

$$\mathbf{v}(v_i) = \mathbf{v}(\pi(v_i)) \qquad \text{and} \qquad \mathbf{r}(r_i) = \mathbf{r}(\phi(r_i))$$

Combining all these observations, we can conclude that $w(\bar{\eta}; G, q, \mathbf{v}, \mathbf{r}) = w(\mu(\bar{\eta}); H, \mu(q), \mathbf{v}, \mathbf{r})$ and $w$ is indeed invariant. $\qquad\square$

**Lemma B.10.** *The consensus protocol $c$, as described in §A.4, is invariant.*

*Proof.* Let $G = (V, E, R)$ be a knowledge graph and $H$ be isomorphic to $G$ via an isomorphism $\mu = (\pi, \phi)$. Let $\bar{\eta}_{1:N} \in \mathcal{W}(G)$ be a sequence of walks in $G$. To show that the output of the consensus protocol is invariant, we need to prove that for each $v \in V$ and $r \in R$, the following holds:

$$\Delta\mathbf{v}(v) = \mathbf{v}'(\pi(v)) \qquad \text{and} \qquad \Delta\mathbf{r}(r) = \Delta\mathbf{r}'(\phi(r)) \tag{14}$$

where $(\Delta\mathbf{v}, \Delta\mathbf{r}) = c(\mathbf{h}, \bar{\eta}_{1:N})$ and $(\Delta\mathbf{v}', \Delta\mathbf{r}') = c(\mathbf{h}, \mu(\bar{\eta}_{1:N}))$ for $\mathbf{h} = (\mathbf{s}^{(V)}, \mathbf{s}^{(R)}, \mathbf{a}^{(V)}, \mathbf{a}^{(R)})$.

The result follows from the fact that $\pi$ and $\phi$ are bijections – whenever $v$ is the $j^{\text{th}}$ vertex visited in the walk $\bar{\eta}_i$, the $j^{\text{th}}$ node of $\mu(\bar{\eta}_i)$ must be $\pi(v)$ (and vice versa). An analogous result holds for the relations. Hence, the aggregation performed by $c$ for $v$ (resp. $r$) over $\bar{\eta}_{1:N}$ is equivalent to the aggregation for $\pi(v)$ (resp. $\phi(r)$) over $\mu(\bar{\eta}_{1:N})$, and (14) is indeed satisfied. $\qquad\square$

**Proposition 4.3.** FLOCK *with components as described in §4 is invariant in probability.*

*Proof.* The result follows naturally from aggregating the results of Corollary B.8 and Lemmas B.9 and B.10, followed by applying Proposition 4.2. $\qquad\square$

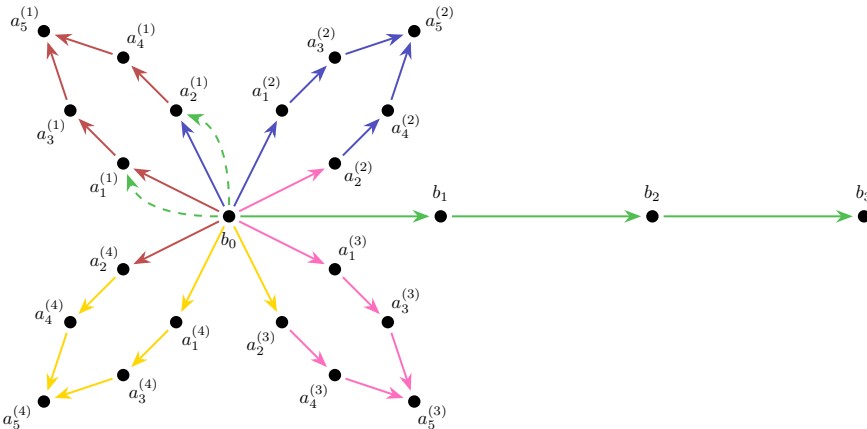

Figure 5: An example of a graph from PETALS with $c = 4$, $l = 2$ and $t = 3$, and the associated link prediction instances (dashed). The relation types 'red', 'blue', 'pink' and 'yellow' are structurally isomorphic, hence become equated in the eyes of the existing KGFMs.

## C    DETAILS OF THE PETALS BENCHMARK

State-of-the-art knowledge graph foundation models (KGFMs) typically impose relational invariance. Formally, given two knowledge graphs $G = (V, E, R)$ and $H = (V', E', R')$, if there exists an isomorphism $(\pi, \phi)$ from $G$ to $H$, then for any $r \in R$, the model enforces identical representations for $r$ and its image $\phi(r) \in R'$. This design promotes generalization across different graphs, as it aligns analogous relations, but reduces expressivity within a single graph ($G = H$), where relations related by automorphisms are forced to be indistinguishable. Concretely, if an automorphism $(\pi, \phi)$ of $G$ maps $r_1$ to $r_2$, then the model must treat $r_1$ and $r_2$ as identical during inference. While some approaches mitigate this limitation via the labeling trick, assigning distinct embeddings to query-specific nodes and relations, this only isolates the queried relation type and does not resolve the underlying issue in general.

Motivated by this limitation, we introduce the PETALS benchmark. PETALS comprises 220 graphs, each paired with a link prediction query $(h, r, ?)$ and a target set $\{t_1, t_2\}$. While $t_1$ and $t_2$ are non-isomorphic, KGFMs enforcing relational invariance are unable to distinguish them, producing identical predictions. We empirically validate this property by evaluating the classification accuracy of marking $t_1$ as TRUE and $t_2$ as FALSE, reported in Table 1.

### C.1    STRUCTURE OF THE STUDIED KGS

Knowledge graphs in PETALS follow a flower-like structure, parametrized by the number $c$ of 'petals', their length $l$ and the length $t$ of the 'stem' (see Figure 5 for visualization).

**Vertices.** Each 'petal' is a set $A^{(i)}$ of $2l + 1$ vertices $A^{(i)} = \left\{ a_1^{(i)}, a_2^{(i)}, \ldots, a_{2l+1}^{(i)} \right\}$, while the stem $B$ consists of $t + 1$ nodes $B = \{b_0, b_1, \ldots, b_t\}$. The full set of entities is then:

$$V = B \cup \bigcup_{i=1}^{c} A(i) = \{b_0, b_1, \ldots, b_t\} \cup \left\{ a_j^{(i)} \mid 1 \le i \le c, 1 \le j \le 2l + 1 \right\}$$

We call $b_0$ the 'central' node, as it is connected to every petal, as described below.

**Edges.** The nodes of the stem are connected in a consecutive manner by the same relation type $r_0$. Precisely, for each $i \in 1, \cdots, t$, there exists an edge $(b_{i-1}, r_0, b_i)$. Each petal $A^{(i)}$ is associated with two edge types $r_1^{(i)}, r_2^{(i)}$, and is connected to the central node $b_0$ with links $\left( b_0, r_1^{(i)}, a_1^{(i)} \right)$ and $\left( b_0, r_2^{(i)}, a_2^{(i)} \right)$. The rest of the petal is connected with edges of type $r_1^{(i)}$ only, going from $a_{2j-1}^{(i)}$ to $a_{2j+1}^{(i)}$, and from $a_{2j}^{(i)}$ to $a_{2j+2}^{(i)}$. Finally, there are also edges linking $a_{2\ell-1}^{(i)}$ and $a_{2l}^{(i)}$ to $a_{2l+1}^{(i)}$.

Therefore, the full set of edges can be characterized as:

$$E = (\{(b_{i-1}, r_0, b_i) \mid 1 \le i \le t\}) \cup \left( \bigcup_{i=1}^{c} \left\{ (b_0, r_1^{(i)}, a_1^{(i)}), (b_0, r_2^{(i)}, a_2^{(i)}) \right\} \right)$$

$$\cup \left( \bigcup_{i=1}^{c} \bigcup_{j=1}^{\ell-1} \left\{ \left( a_{2j-1}^{(i)}, r_1^{(i)}, a_{2j+1}^{(i)} \right), \left( a_{2j}^{(i)}, r_1^{(i)}, a_{2j+2}^{(i)} \right) \right\} \right)$$

$$\cup \left( \bigcup_{i=1}^{c} \left\{ \left( a_{2\ell-1}^{(i)}, r_1^{(i)}, a_{2l+1}^{(i)} \right), \left( a_{2l}^{(i)}, r_1^{(i)}, a_{2l+1}^{(i)} \right) \right\} \right)$$

We select each of the types $r_1^{(i)}$ and $r_2^{(i)}$ from the set of considered relations $R = \{r_1, \ldots, r_{|R|}\}$ so that any relation-invariant model will equate all petals (i.e. so that for each pair of petals, there is an automorphism taking one to another). For instance, Figure 5 displays a cyclic pattern, in which $r_2^{(i)} = r_1^{(i+1)}$. Such symmetry causes all petals to be isomorphic, and leads to the inability of KGFMs to distinguish between the relations inside them.

**Link prediction instances.** Although the petals are isomorphic to each other, given the asymmetry of edge types from $b_0$ to $a_1^{(i)}$ and $a_2^{(i)}$, the nodes within a single petal generally can be distinguished. Therefore, for each graph with the structure as described above, we randomly sample one of the stem nodes $b_s$, and ask the link prediction query $(b_s, r_0, ?)$. For the target nodes, we randomly select petal index $i$ and distance $j$ from the central node $b_0$, and consider the predictions for $a_{2j-1}^{(i)}$ and $a_{2j}^{(i)}$. For example, Figure 5 shows the case when $b_s = b_0$, $i = 1$ and $j = 1$, where the query is $(b_0, r_0, ?)$ and we are interested in the scores for $a_1^{(1)}$ and $a_2^{(1)}$.

### C.2 PARAMETERS AND GENERATION

We construct PETALS by manually designing 11 relation-assignment schemes that guarantee isomorphism across all petals. For each such selection, which already determines the number $c$ of petals, we generate 20 graphs corresponding to all combinations of $t \in \{1, 2, 3, 4\}$ and $l \in \{1, 2, 3, 4, 5\}$. Each graph is paired with a link prediction query and two target nodes, sampled as described above. This yields $11 \cdot 20 = 220$ instances that constitute the PETALS benchmark.

Table 6: Training scalability analysis on a single NVIDIA RTX A6000 (48 GB) with batch size = 8. FLOCK using 16 number of base walks and 1 ensemble.

| Model | Parameters | Time / batch (s) | GPU memory (GB) |
|---|---|---|---|
| ULTRA | 168,705 | 0.117 | 2.110 |
| TRIX | 87,138 | 0.690 | 3.442 |
| FLOCK | 801,969 | 1.30 | 27.89 |

Table 7: Inference scalability on a single NVIDIA RTX A6000 (48 GB) with batch size $= 8$. Left columns specify base walks $n$ and ensembled passes $P$. Dashes indicate not applicable.

| Model | # Base Walks $n$ | Ensemble $P$ | Time /batch (s) | GPU memory (GB) |
|---|---|---|---|---|
| ULTRA | — | 1 | 0.073 | 0.848 |
| TRIX | — | 1 | 0.500 | 1.382 |
| FLOCK | 16 | 1 | 1.26 | 2.868 |
| | 16 | 2 | 1.99 | 2.864 |
| | 16 | 4 | 3.24 | 3.938 |
| | 16 | 8 | 5.45 | 5.172 |
| | 16 | 16 | 9.45 | 8.892 |
| | 128 | 1 | 1.77 | 5.000 |
| | 128 | 2 | 2.80 | 7.880 |
| | 128 | 4 | 5.00 | 14.42 |
| | 128 | 8 | 10.05 | 43.68 |

## D COMPUTATIONAL COMPLEXITY

Recall that $I$ is the iterations in each forward pass of FLOCK; $n$ is the base walk count; $\ell$ is the walk length; $L$ is the number of linear sequence-model layers (such as GRU); and $d$ is the hidden dimension for the sequence processor. Note that in practice, we perform $P$ forward passes and ensemble their outputs to reduce variance. For a single pass ($P=1$), walk sampling and recording cost $O(n\ell)$, while the sequence processor with $L$ layers of hidden dimension $d$ costs $O(n\ell L d^2)$. The consensus protocol costs $O(n\ell d)$. In total, the time complexity is $O(PIn\ell L d^2)$, which scales linearly with the number of (base) walks $n$, the length of walks $\ell$, and the number of ensembled predictions $P$. We empirically verified this in §E.

Compared with message-passing KGFMs like ULTRA and TRIX, FLOCK's complexity is *independent* of the graph size and average degree; empirically, however, using more walks (increasing $n$) and longer walks (increasing $\ell$) improves coverage and yields more fine-grained representation.

The space complexity of FLOCK per forward pass is $O(n\ell d)$ plus model parameters $O(Ld^2)$. Note that running ensembles sequentially keeps peak memory near this bound, while parallel ensembling increases it by a factor of $P$.

## E SCALABILITY ANALYSIS

To investigate the scalability of the proposed method FLOCK, we report the training and inference time per batch and peak GPU memory for ULTRA, TRIX, and variants of FLOCK on a single RTX A6000 (48 GB) in Tables 6 and 7.

**Training.** During training, we fix FLOCK to $n = 16$ base walks and with an ensemble size of $P = 1$, which yields higher cost than ULTRA/TRIX but remains feasible on a single GPU. In addition, unlike ULTRA/TRIX, FLOCK does not rely on GNN message passing where highly optimized fused sparse kernels (e.g., RSPMM kernel developed in Zhu et al. (2021)) accelerate computation; instead, FLOCK's runtime is dominated by walk sampling and sequence encoding, making time per batch the main bottleneck. As a result, pretraining typically takes about three days. One avenue for future work is to develop similarly highly optimized kernels for random-walk sampling.

Table 8: Noise injection over the best performing KGFM baseline TRIX.

(a) Zero-shot entity prediction.

|  | MRR | Hits@10 |
|---|---|---|
| TRIX | 0.366 | 0.518 |
| + noise | 0.385 | 0.545 |
| FLOCK | **0.391** | **0.560** |

(b) Zero-shot relation prediction.

|  | MRR | Hits@1 |
|---|---|---|
| TRIX | 0.792 | 0.687 |
| + noise | 0.739 | 0.643 |
| FLOCK | **0.881** | **0.817** |

(c) Accuracy on PETALS.

|  | Accuracy |
|---|---|
| TRIX | 50% |
| + noise | 52% |
| FLOCK | **100%** |

**Inference.** Additionally, we report the inference results in Table 7, where we vary the number of walks $n$ and ensembled passes $P$. We observe near-linear growth of latency and VRAM with $n$, reflecting the dominant costs of walk sampling and sequence processing. Note that during inference, ensembled predictions are parallelizable, meaning that with sufficient GPU memory, these $P$ stochastic passes can be executed concurrently, so the latency grows sublinearly in $P$, while peak VRAM scales roughly linearly with $P$. In practice, reducing $n$ (walks) or $P$ (ensembled passes) lowers both memory and latency, while larger $n/P$ settings trade extra cost for better coverage and stability on harder KGs.

## F  NOISE INJECTION OVER EXISTING KGFMS

**Setup.** Since noise injection is a possible way to build a probabilistic equivariant KGFM in a different way from our approach (Gao et al., 2023), it is natural to ask how such KGFMs would perform compared to FLOCK. To answer this question, we apply noise injection over the best performing KGFMs baselines TRIX. Specifically, in each forward pass, we add element-wise noise sampled from a uniform distribution $\epsilon \sim \mathcal{U}[-0.5, 0]$ to all relation and entity embeddings after the initialization stage. Note that the addition of noise technically breaks deterministic node-relation equivariance, but the resulting model (TRIX + noise) still respects probabilistic node-relation equivariance. We then pretrain TRIX using the same experimental setup shown in §5.2, and compare with TRIX without noise injection and FLOCK. To minimize the variance induced by injected noise and to ensure a fair comparison, we report ensembled prediction results with 16 samples for both TRIX + noise and FLOCK. This is a strong baseline implementing the ideas of prior work on noise injection and test-time ensembling for message passing networks on KGs (Lee et al., 2023; Gao et al., 2023).

**Results.** We report the average zero-shot performance for entity prediction and relation prediction over 54 KGs in Tables 8a and 8b, respectively, as well as trained performance for PETALS in Table 8c. Across all tasks, TRIX with naive noise injection fails to close the gap between FLOCK. In particular, TRIX + noise degrades compared with vanilla TRIX without noise injection in relation prediction, while boosting the performance in the entity prediction task. We hypothesize that such a difference lies in the added randomization breaks symmetry among entity embeddings more than among relation embeddings, and entity prediction depends more on having distinguishable entity representations than relation prediction does. Additionally, we attribute this performance gap between FLOCK and TRIX + noise to the source of randomization. FLOCK introduces stochasticity through random walks, which induces *structure-informed* perturbations that respect the underlying topology. In contrast, TRIX with naive noise injection attempts to break deterministic node-relation equivariance by introducing structure-agnostic noise, which might, in turn, hurt the model's generalization. Together, these findings suggest that simply adding structure-agnostic noise is insufficient; performance gains only arise when stochasticity is topology-aware and is induced from the graph structure in a principled way.

## G  CASE STUDY: RELATION EMBEDDING ON METAFAM

**Setup.** To further showcase why expressivity matters for zero-shot generalization, we present a case study on the METAFAM dataset (Zhou et al., 2023). METAFAM is built from a fixed family-relations ontology: during training, models observe edges with relations `motherOf`, `fatherOf`, `daughterOf`, `sonOf`, while the test queries only involve `motherOf` and `fatherOf`. Up to gender symmetries, this reduces to two effective predictive patterns (*parent_of* vs. *child_of*), and the test set focuses on a single one (*parent_of*).

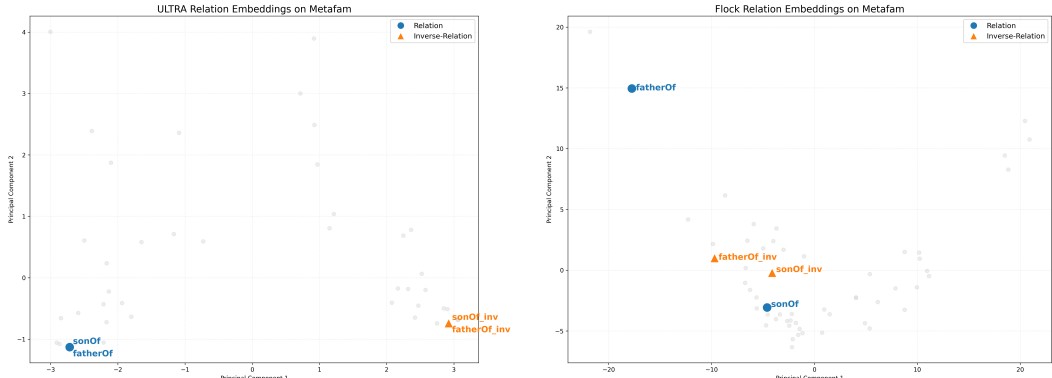

Figure 6: PCA of relation embeddings on METAFAM. ULTRA (Left) maps several inverse pairs (e.g., `fatherOf` vs. `sonOf`) to almost similar embeddings, where FLOCK (Right) yields clearly separated embeddings, indicating that its probabilistic equivariance allow FLOCK to distinguish between these semantically different relations, explaining its strong zero-shot performances.

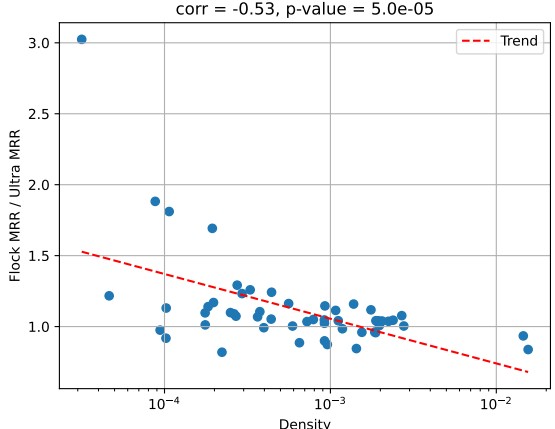

Figure 7: The zero-shot entity prediction performance of FLOCK relative to ULTRA, plotted against the densities of the 53 KGs. Performance of FLOCK and log-density of KGs have a Pearson correlation coefficient of -0.53 with p-value 5.0e-5, showing a statistically significant negative correlation.

In the zero-shot setting, KGFMs cannot adapt to this ontology and must rely on their pretrained relation representations. Notice that here, METAFAM is challenging: many relations are structurally similar (e.g., `fatherOf`, `sonOf`, `sisterOf`, `nieceOf`) yet encode opposite predictive patterns.

**Result.** Figure 6 shows that ULTRA's relation embeddings largely collapse these families, placing `fatherOf` and `sonOf` in almost identical positions in the PCA plane. This collapse makes it difficult to distinguish who is the parent and who is the child, leading to poor zero-shot performance. FLOCK, in contrast, can distinguish between these relations even if they are structurally similar, thanks to its random-walk sampling which introduce probabilistic equivariance on nodes and relations. As a results, FLOCK can produce distinct embeddings to `fatherOf` and `sonOf` and achieves much stronger zero-shot performance on METAFAM.

## H  ANALYSIS OF KG SPARSITY AND PERFORMANCE

**Setup.** While §G explains FLOCK's high performance for METAFAM, understanding its performances for other general KGs would be beneficial. Thus, we present an additional analysis on the 53 remaining KGs by identifying a structural property that is indicative of the performance of

FLOCK. For the performance measure, we use the relative gain of FLOCK's zero-shot entity prediction MRR compared to ULTRA. For the structural property, we focus on density of KGs defined by $\frac{|E|}{|V|(|V|-1)}$ which affects the speed of random walks traversing all edges of a KG (intuitively because more edges means more time needed to traverse all of them), and hence is relevant in the context of our theory in §4.2. We make the argument more grounded below.

**Result.** Figure 7 shows that FLOCK tends to perform well on sparse KGs, while less so on dense KGs, and the tendency is statistically significant. Interestingly, this agrees with our theoretical analysis in §4.2, in which a necessary condition for universality is that the random walk covers all edges of a KG with a high probability (Proposition 4.1). The time taken until covering all edges is called the edge cover time, and it is known to be e.g., $O(|V||E|)$ for uniform walks (Zuckerman, 1991), which is proportional to the density of a graph. This suggests the performance of FLOCK is associated with the easiness to visit as many edges as possible rapidly, which is more challenging for dense KGs. This analysis is consistent with the observations of recent works on graph learning based on random walks, e.g., Wang & Cho (2024, Section 6) and Kim et al. (2025).

## I FURTHER EXPERIMENTAL DETAILS

**Datasets.** This section provides the full details for all experiments described in the main text. For pretraining, we fit the FLOCK model on three standard transductive knowledge graph completion benchmarks, following Galkin et al. (2024): FB15k-237 (Toutanova & Chen, 2015), WN18RR (Dettmers et al., 2018), and CoDEx Medium (Safavi & Koutra, 2020). Then, we evaluate zero-shot transfer of both entity prediction and relation prediction, as well as the finetuning performance on multiple datasets grouped as follows:

- **Inductive** $e, r$**.** Link prediction tasks involving previously unseen nodes and relation types. This includes the 13 datasets from INGRAM (Lee et al., 2023): FB-25, FB-50, FB-75, FB-100, WK-25, WK-50, WK-75, WK-100, NL-0, NL-25, NL-50, NL-75, NL-100, as well as 10 datasets from MTDEA (Zhou et al., 2023): MT1 tax, MT1 health, MT2 org, MT2 sci, MT3 art, MT3 infra, MT4 sci, MT4 health, Metafam, and FBNELL.

- **Inductive** $e$**.** Link prediction tasks involving novel nodes but fixed relation types. This category comprises 12 GraIL datasets (Teru et al., 2020) (WN-v1 through WN-v4, FB-v1 through FB-v4, NL-v1 through NL-v4), 4 INDIGO benchmarks (Liu et al., 2021) (HM 1k, HM 3k, HM 5k, HM Indigo), and 2 NodePiece datasets (Galkin et al., 2022): ILPC Small and ILPC Large.

- **Transductive.** Link prediction tasks where both entities and relations are observed during training. These include CoDEx Small, CoDEx Large (Safavi & Koutra, 2020), NELL-995 (Xiong et al., 2017), YAGO 310 (Mahdisoltani et al., 2015), WDsinger, NELL23k, FB15k-237(10), FB15k-237(20), FB15k-237(50) (Lv et al., 2020), AristoV4 (Chen et al., 2021), DBpedia100k (Ding et al., 2018), ConceptNet100k (Malaviya et al., 2020), and Hetionet (Himmelstein et al., 2017).

**Full results of §5.2.** Full tables of zero-shot entity prediction results are presented in Table 9, and full tables of finetuned performance are given in Table 10. We further provide the complete zero-shot and finetuned relation prediction results in Table 11 and Table 12. The dataset statistics are in Table 13, Table 14 and Table 15. Table 16 presents the pretraining graph mix shown in §5.3. Finally, detailed hyperparameter settings can be found in Table 17, Table 18 and Table 19.

**Training.** Following conventions in the literature (Zhu et al., 2021; Huang et al., 2023), for each triple $(h, r, t)$, we add the corresponding inverse triple $(h, r^{-1}, t)$, where $r^{-1}$ is a fresh relation symbol. All FLOCK instances and its variants are optimized to minimize the negative log-likelihood over positive and negative facts under the *partial completeness assumption* (Galárraga et al., 2013), where negatives are generated by randomly corrupting either the head or the tail entity (for entity prediction) or by corrupting the relation (for relation prediction). To reduce overfitting, we remove edges that directly connect the queried endpoints. The best checkpoint is selected by validation performance. For entity prediction, we take the embedding for potential target $t$ and relations $r$, and obtain the score $p(h, r, t)$ by passing into a 2-layer MLP. For relation prediction, we concatenate the embedding for source $h$, target $t$, and potential relation $r$ to obtain the score $p(h, r, t)$.

Let $(h, r, t)$ be a positive triple and let $k$ denote the number of negatives sampled per positive, where $(h_i, r, t_i)$ is the $i$-th negative samples for entity prediction, and $h, r_i, t_i$ is the $i$-th negative samples for relation prediction. Following Sun et al. (2019), we also consider a self-adversarial variant where negatives are reweighted according to their current difficulty. With adversarial temperature $\alpha > 0$, the weights for entity and relation prediction, respectively, are

$$w_{i,\alpha}^{\text{ent}} = \text{Softmax}\left(\frac{\log\big(1 - p(h_i', r, t_i')\big)}{\alpha}\right), \qquad w_{i,\alpha}^{\text{rel}} = \text{Softmax}\left(\frac{\log\big(1 - p(h, r_i', t)\big)}{\alpha}\right).$$

The corresponding losses become

$$\mathcal{L}_{\text{ent}}^{\text{adv}} = -\log p(h, r, t) \;-\; \sum_{i=1}^{k} w_{i,\alpha}^{\text{ent}} \log\big(1 - p(h_i', r, t_i')\big),$$

$$\mathcal{L}_{\text{rel}}^{\text{adv}} = -\log p(h, r, t) \;-\; \sum_{i=1}^{k} w_{i,\alpha}^{\text{rel}} \log\big(1 - p(h, r_i', t)\big).$$

Table 9: Zero-shot entity prediction results. Bold indicates the best score per row.

| Dataset | ULTRA | | TRIX | | FLOCK | |
|---|---|---|---|---|---|---|
| | MRR | Hits@10 | MRR | Hits@10 | MRR | Hits@10 |
| **Inductive $e, r$** | | | | | | |
| FB-25 | 0.388 | 0.640 | 0.393 | 0.650 | **0.404** | **0.664** |
| FB-50 | 0.338 | 0.543 | 0.334 | 0.547 | **0.352** | **0.566** |
| FB-75 | 0.403 | 0.604 | 0.401 | 0.611 | **0.418** | **0.622** |
| FB-100 | 0.449 | 0.642 | 0.436 | 0.635 | **0.452** | **0.663** |
| WK-25 | **0.316** | **0.532** | 0.305 | 0.496 | 0.280 | 0.491 |
| WK-50 | **0.166** | **0.324** | **0.166** | 0.313 | 0.136 | 0.278 |
| WK-75 | 0.365 | 0.537 | 0.368 | 0.513 | 0.382 | **0.538** |
| WK-100 | 0.164 | 0.286 | **0.188** | 0.299 | 0.187 | **0.304** |
| NL-0 | 0.342 | 0.523 | **0.385** | 0.549 | 0.381 | **0.606** |
| NL-25 | **0.395** | 0.569 | 0.377 | 0.589 | 0.345 | **0.590** |
| NL-50 | **0.407** | **0.570** | 0.404 | 0.548 | 0.366 | 0.565 |
| NL-75 | **0.368** | **0.547** | 0.351 | 0.525 | 0.311 | 0.524 |
| NL-100 | 0.471 | 0.651 | **0.486** | 0.676 | 0.452 | **0.692** |
| MT1 tax | 0.224 | 0.305 | **0.358** | 0.452 | 0.282 | **0.383** |
| MT1 health | 0.298 | 0.374 | 0.376 | 0.457 | **0.385** | **0.481** |
| MT2 org | 0.095 | 0.159 | 0.091 | 0.156 | **0.100** | **0.163** |
| MT2 sci | 0.258 | 0.354 | **0.323** | **0.465** | 0.318 | 0.458 |
| MT3 art | 0.259 | 0.402 | 0.284 | 0.441 | **0.301** | **0.466** |
| MT3 infra | 0.619 | 0.755 | 0.655 | 0.797 | **0.684** | **0.821** |
| MT4 sci | 0.274 | 0.449 | 0.290 | 0.460 | **0.301** | **0.463** |
| MT4 health | 0.624 | 0.737 | 0.677 | 0.775 | **0.680** | **0.780** |
| Metafam | 0.238 | 0.644 | 0.341 | 0.815 | **0.476** | **0.935** |
| FBNELL | 0.485 | 0.652 | 0.473 | 0.660 | **0.502** | **0.700** |
| **Inductive $e$** | | | | | | |
| FB-v1 | 0.498 | 0.656 | **0.515** | 0.682 | 0.500 | **0.697** |
| FB-v2 | 0.512 | 0.700 | 0.525 | 0.730 | **0.535** | **0.737** |
| FB-v3 | 0.491 | 0.654 | 0.501 | 0.669 | **0.511** | **0.685** |
| FB-v4 | 0.486 | 0.677 | 0.493 | 0.687 | **0.505** | **0.702** |
| WN-v1 | 0.648 | 0.768 | **0.699** | 0.791 | 0.698 | **0.803** |
| WN-v2 | 0.663 | 0.765 | 0.678 | 0.781 | **0.696** | **0.790** |
| WN-v3 | 0.376 | 0.476 | 0.418 | 0.541 | **0.467** | **0.608** |
| WN-v4 | 0.611 | 0.705 | 0.648 | 0.723 | **0.653** | **0.729** |
| NL-v1 | 0.785 | **0.913** | **0.806** | 0.898 | 0.658 | 0.863 |
| NL-v2 | 0.526 | 0.707 | 0.569 | 0.768 | **0.588** | **0.797** |
| NL-v3 | 0.515 | 0.702 | 0.558 | 0.743 | **0.590** | **0.783** |
| NL-v4 | 0.479 | 0.712 | 0.538 | 0.765 | **0.555** | **0.786** |
| HM 1k | 0.059 | 0.092 | **0.072** | **0.128** | 0.069 | 0.119 |
| HM 3k | 0.037 | 0.077 | **0.069** | **0.119** | 0.067 | 0.118 |
| HM 5k | 0.034 | 0.071 | 0.062 | 0.110 | **0.064** | **0.116** |
| HM Indigo | **0.440** | **0.648** | 0.436 | 0.645 | 0.423 | 0.638 |
| ILPC Small | 0.302 | 0.443 | 0.303 | 0.455 | **0.309** | **0.459** |
| ILPC Large | 0.290 | 0.424 | 0.307 | 0.428 | **0.318** | **0.438** |
| **Transductive** | | | | | | |
| NELL995 | 0.406 | 0.543 | 0.472 | 0.629 | **0.494** | **0.655** |
| NELL23k | 0.239 | 0.408 | **0.290** | **0.497** | 0.233 | 0.398 |
| WDsinger | 0.382 | 0.498 | **0.511** | **0.609** | 0.410 | 0.528 |
| ConceptNet100k | 0.082 | 0.162 | 0.193 | 0.345 | **0.248** | **0.453** |
| CoDEx Small | **0.472** | 0.667 | **0.472** | **0.670** | 0.441 | 0.644 |
| CoDEx Large | 0.338 | **0.469** | 0.335 | **0.469** | **0.342** | 0.464 |
| YAGO310 | **0.451** | 0.615 | 0.409 | 0.627 | 0.414 | **0.674** |
| AristoV4 | 0.182 | 0.282 | 0.181 | 0.286 | **0.308** | **0.443** |
| DBpedia100k | 0.398 | 0.576 | 0.426 | 0.603 | **0.450** | **0.627** |
| Hetionet | 0.257 | 0.379 | **0.279** | **0.420** | 0.246 | 0.371 |
| FB15k-237(10) | **0.248** | 0.398 | 0.246 | 0.393 | 0.246 | **0.402** |
| FB15k-237(20) | 0.272 | 0.436 | 0.269 | 0.430 | **0.273** | **0.444** |
| FB15k-237(50) | **0.324** | **0.526** | 0.321 | 0.521 | 0.319 | 0.518 |

Table 10: Finetuned entity prediction results. Bold indicates the best score per row.

| Dataset | ULTRA | | TRIX | | FLOCK | |
|---|---|---|---|---|---|---|
| | MRR | Hits@10 | MRR | Hits@10 | MRR | Hits@10 |
| **Inductive** $e, r$ | | | | | | |
| FB-25 | 0.383 | 0.635 | 0.393 | 0.650 | **0.405** | **0.666** |
| FB-50 | 0.334 | 0.538 | 0.334 | 0.547 | **0.357** | **0.570** |
| FB-75 | 0.400 | 0.598 | 0.401 | 0.611 | **0.425** | **0.630** |
| FB-100 | 0.444 | 0.643 | 0.436 | 0.633 | **0.460** | **0.668** |
| WK-25 | **0.321** | **0.535** | 0.300 | 0.493 | 0.298 | 0.506 |
| WK-50 | 0.140 | 0.280 | **0.166** | **0.313** | 0.127 | 0.260 |
| WK-75 | 0.380 | 0.530 | 0.368 | 0.513 | **0.405** | **0.556** |
| WK-100 | 0.168 | 0.286 | **0.188** | 0.299 | 0.187 | **0.306** |
| NL-0 | 0.329 | 0.551 | 0.385 | 0.549 | **0.418** | **0.619** |
| NL-25 | **0.407** | 0.596 | 0.377 | 0.589 | 0.405 | **0.626** |
| NL-50 | **0.418** | **0.595** | 0.405 | 0.555 | 0.391 | 0.562 |
| NL-75 | **0.374** | **0.570** | 0.351 | 0.525 | 0.344 | 0.544 |
| NL-100 | 0.458 | 0.684 | 0.482 | 0.691 | **0.486** | **0.714** |
| MT1 tax | 0.330 | 0.459 | 0.397 | **0.508** | **0.413** | 0.497 |
| MT1 health | 0.380 | 0.467 | 0.376 | 0.457 | **0.394** | **0.493** |
| MT2 org | 0.104 | 0.170 | 0.098 | 0.162 | **0.107** | **0.174** |
| MT2 sci | 0.311 | 0.451 | 0.331 | **0.526** | **0.366** | 0.525 |
| MT3 art | 0.306 | 0.473 | 0.289 | 0.441 | **0.330** | **0.483** |
| MT3 infra | 0.657 | 0.807 | 0.672 | 0.810 | **0.709** | **0.838** |
| MT4 sci | 0.303 | 0.478 | 0.305 | 0.482 | **0.324** | **0.509** |
| MT4 health | 0.704 | 0.785 | 0.702 | 0.785 | **0.711** | **0.790** |
| Metafam | **0.997** | **1.000** | **0.997** | **1.000** | 0.992 | **1.000** |
| FBNELL | 0.481 | 0.661 | 0.478 | 0.655 | **0.531** | **0.714** |
| **Inductive** $e$ | | | | | | |
| FB-v1 | 0.509 | 0.670 | 0.515 | 0.682 | **0.549** | **0.721** |
| FB-v2 | 0.524 | 0.710 | 0.525 | 0.730 | **0.553** | **0.754** |
| FB-v3 | 0.504 | 0.663 | 0.501 | 0.669 | **0.528** | **0.696** |
| FB-v4 | 0.496 | 0.684 | 0.493 | 0.687 | **0.510** | **0.702** |
| WN-v1 | 0.685 | 0.793 | 0.705 | 0.798 | **0.715** | **0.811** |
| WN-v2 | 0.679 | 0.779 | 0.682 | 0.780 | **0.702** | **0.795** |
| WN-v3 | 0.411 | 0.546 | 0.425 | 0.543 | **0.494** | **0.627** |
| WN-v4 | 0.614 | 0.720 | 0.650 | 0.722 | **0.665** | **0.741** |
| NL-v1 | 0.757 | 0.878 | **0.804** | 0.899 | 0.762 | **0.928** |
| NL-v2 | 0.575 | 0.761 | 0.571 | 0.764 | **0.612** | **0.806** |
| NL-v3 | 0.563 | 0.755 | 0.571 | 0.759 | **0.606** | **0.803** |
| NL-v4 | 0.469 | 0.733 | 0.551 | 0.772 | **0.572** | **0.801** |
| HM 1k | 0.042 | 0.100 | **0.072** | 0.128 | 0.071 | **0.153** |
| HM 3k | 0.030 | 0.090 | **0.069** | 0.119 | 0.067 | **0.153** |
| HM 5k | 0.025 | 0.068 | **0.074** | 0.118 | 0.061 | **0.130** |
| HM Indigo | 0.432 | 0.639 | **0.436** | **0.645** | 0.418 | 0.633 |
| ILPC Small | 0.303 | 0.453 | 0.303 | **0.455** | **0.305** | 0.454 |
| ILPC Large | 0.308 | **0.431** | 0.310 | **0.431** | **0.320** | 0.441 |
| **Transductive** | | | | | | |
| NELL995 | 0.509 | 0.660 | 0.506 | 0.648 | **0.531** | **0.665** |
| NELL23k | 0.268 | 0.450 | **0.306** | **0.536** | 0.280 | 0.465 |
| WDsinger | 0.417 | 0.526 | **0.502** | **0.620** | 0.435 | 0.543 |
| ConceptNet100k | 0.310 | 0.529 | 0.340 | 0.564 | **0.352** | **0.580** |
| CoDEx Small | **0.490** | **0.686** | 0.484 | 0.676 | 0.463 | 0.648 |
| CoDEx Large | 0.343 | 0.478 | **0.348** | **0.481** | 0.342 | 0.467 |
| YAGO310 | **0.557** | **0.710** | 0.541 | 0.702 | 0.552 | 0.700 |
| AristoV4 | 0.343 | 0.496 | 0.345 | 0.499 | **0.383** | **0.523** |
| DBpedia100k | 0.436 | 0.603 | 0.457 | 0.619 | **0.470** | **0.623** |
| Hetionet | **0.399** | **0.538** | 0.394 | 0.534 | 0.314 | 0.465 |
| FB15k-237(10) | 0.254 | 0.411 | 0.253 | 0.408 | **0.260** | **0.420** |
| FB15k-237(20) | 0.274 | 0.445 | 0.273 | 0.441 | **0.284** | **0.459** |
| FB15k-237(50) | **0.325** | **0.528** | 0.322 | 0.522 | 0.317 | 0.517 |
| **Pretrained** | | | | | | |
| FB15k-237 | **0.368** | **0.564** | 0.366 | 0.559 | 0.343 | 0.532 |
| WN18RR | 0.480 | 0.614 | 0.514 | 0.611 | **0.550** | **0.656** |
| CoDEx Medium | **0.372** | **0.525** | 0.365 | 0.521 | 0.351 | 0.496 |

Table 11: Zero-shot relation prediction results. Bold indicates the best score per row.

| Dataset | ULTRA | | TRIX | | FLOCK | |
|---|---|---|---|---|---|---|
| | MRR | Hits@1 | MRR | Hits@1 | MRR | Hits@1 |
| **Inductive $e, r$** | | | | | | |
| FB-25 | 0.687 | 0.565 | 0.805 | 0.724 | **0.895** | **0.839** |
| FB-50 | 0.696 | 0.575 | 0.780 | 0.699 | **0.880** | **0.820** |
| FB-75 | 0.698 | 0.555 | 0.822 | 0.747 | **0.903** | **0.844** |
| FB-100 | 0.830 | 0.728 | 0.921 | 0.880 | **0.962** | **0.938** |
| WK-25 | 0.857 | 0.760 | 0.881 | 0.823 | **0.952** | **0.929** |
| WK-50 | 0.865 | 0.793 | 0.868 | 0.818 | **0.921** | **0.882** |
| WK-75 | 0.911 | 0.875 | 0.916 | 0.883 | **0.962** | **0.944** |
| WK-100 | 0.887 | 0.812 | 0.907 | 0.869 | **0.963** | **0.937** |
| NL-0 | 0.632 | 0.502 | 0.658 | 0.519 | **0.714** | **0.574** |
| NL-25 | 0.688 | 0.562 | **0.742** | 0.614 | 0.729 | **0.632** |
| NL-50 | 0.680 | 0.569 | 0.755 | 0.636 | **0.813** | **0.728** |
| NL-75 | 0.795 | 0.692 | 0.788 | 0.699 | **0.833** | **0.756** |
| NL-100 | 0.743 | 0.564 | 0.884 | 0.796 | **0.939** | **0.889** |
| MT1 tax | 0.985 | 0.976 | 0.975 | 0.958 | **0.998** | **0.997** |
| MT1 health | 0.721 | 0.561 | 0.973 | 0.949 | **0.991** | **0.983** |
| MT2 org | 0.974 | 0.951 | 0.986 | 0.973 | **0.991** | **0.984** |
| MT2 sci | 0.976 | 0.961 | 0.964 | 0.941 | **0.995** | **0.992** |
| MT3 art | 0.881 | 0.798 | 0.885 | 0.825 | **0.944** | **0.907** |
| MT3 infra | 0.962 | 0.935 | 0.940 | 0.905 | **0.989** | **0.980** |
| MT4 sci | 0.933 | 0.891 | 0.966 | 0.944 | **0.974** | **0.957** |
| MT4 health | 0.826 | 0.719 | 0.937 | 0.898 | **0.990** | **0.983** |
| Metafam | 0.124 | 0.000 | 0.291 | 0.011 | **0.490** | **0.223** |
| FBNELL | 0.700 | 0.564 | 0.726 | 0.605 | **0.833** | **0.737** |
| **Inductive $e$** | | | | | | |
| FB-v1 | 0.646 | 0.523 | 0.705 | 0.599 | **0.814** | **0.723** |
| FB-v2 | 0.695 | 0.570 | 0.713 | 0.590 | **0.847** | **0.761** |
| FB-v3 | 0.679 | 0.553 | 0.742 | 0.644 | **0.860** | **0.780** |
| FB-v4 | 0.638 | 0.488 | 0.766 | 0.665 | **0.873** | **0.799** |
| WN-v1 | 0.836 | 0.740 | 0.792 | 0.613 | **0.924** | **0.858** |
| WN-v2 | 0.853 | 0.790 | 0.764 | 0.572 | **0.924** | **0.863** |
| WN-v3 | 0.707 | 0.577 | 0.741 | 0.568 | **0.937** | **0.888** |
| WN-v4 | 0.860 | 0.803 | 0.764 | 0.570 | **0.937** | **0.886** |
| NL-v1 | 0.636 | 0.358 | 0.657 | 0.453 | **0.862** | **0.731** |
| NL-v2 | 0.742 | 0.652 | 0.780 | 0.696 | **0.893** | **0.855** |
| NL-v3 | 0.669 | 0.544 | 0.725 | 0.612 | **0.815** | **0.731** |
| NL-v4 | 0.606 | 0.489 | 0.794 | 0.691 | **0.868** | **0.807** |
| ILPC Small | 0.905 | 0.843 | 0.919 | 0.872 | **0.955** | **0.921** |
| ILPC Large | 0.875 | 0.799 | 0.894 | 0.829 | **0.948** | **0.908** |
| HM 1k | 0.626 | 0.447 | 0.663 | 0.414 | **0.687** | **0.500** |
| HM 3k | 0.592 | 0.439 | 0.664 | 0.418 | **0.714** | **0.549** |
| HM 5k | 0.605 | 0.452 | 0.672 | 0.428 | **0.746** | **0.593** |
| HM Indigo | 0.681 | 0.559 | 0.852 | 0.765 | **0.956** | **0.921** |
| **Transductive** | | | | | | |
| NELL995 | 0.583 | 0.437 | 0.578 | 0.457 | **0.684** | **0.555** |
| NELL23k | 0.669 | 0.548 | 0.756 | 0.657 | **0.831** | **0.762** |
| WDsinger | 0.668 | 0.546 | 0.720 | 0.621 | **0.823** | **0.738** |
| ConceptNet100k | 0.181 | 0.083 | 0.650 | 0.469 | **0.795** | **0.658** |
| CoDExSmall | 0.900 | 0.820 | 0.961 | 0.935 | **0.982** | **0.970** |
| CoDExLarge | 0.892 | 0.824 | 0.902 | 0.837 | **0.973** | **0.950** |
| YAGO310 | 0.646 | 0.403 | 0.783 | 0.598 | **0.971** | **0.943** |
| AristoV4 | 0.254 | 0.201 | 0.389 | 0.265 | **0.597** | **0.496** |
| DBpedia100k | 0.650 | 0.509 | 0.717 | 0.582 | **0.919** | **0.861** |
| Hetionet | 0.634 | 0.524 | 0.809 | 0.707 | **0.940** | **0.890** |
| FB15k-237(10) | 0.688 | 0.550 | 0.795 | 0.711 | **0.918** | **0.876** |
| FB15k-237(20) | 0.695 | 0.558 | 0.834 | 0.758 | **0.952** | **0.923** |
| FB15k-237(50) | 0.717 | 0.591 | 0.876 | 0.812 | **0.968** | **0.946** |

Table 12: Finetuned relation prediction results. Bold indicates the best score per row.

| Dataset | ULTRA | | TRIX | | FLOCK | |
|---|---|---|---|---|---|---|
| | MRR | Hits@1 | MRR | Hits@1 | MRR | Hits@1 |
| **Inductive** $e, r$ | | | | | | |
| FB-25 | 0.684 | 0.563 | 0.805 | 0.724 | **0.909** | **0.857** |
| FB-50 | 0.696 | 0.575 | 0.780 | 0.699 | **0.881** | **0.820** |
| FB-75 | 0.754 | 0.638 | 0.822 | 0.699 | **0.911** | **0.854** |
| FB-100 | 0.851 | 0.769 | 0.921 | 0.880 | **0.965** | **0.939** |
| WK-25 | 0.897 | 0.834 | 0.905 | 0.860 | **0.968** | **0.954** |
| WK-50 | 0.865 | 0.793 | 0.881 | 0.840 | **0.925** | **0.876** |
| WK-75 | 0.911 | 0.875 | 0.937 | 0.910 | **0.965** | **0.948** |
| WK-100 | 0.924 | 0.879 | 0.916 | 0.885 | **0.970** | **0.946** |
| NL-0 | 0.632 | 0.502 | 0.655 | 0.518 | **0.731** | **0.602** |
| NL-25 | 0.737 | 0.622 | 0.709 | 0.606 | **0.757** | **0.634** |
| NL-50 | 0.808 | 0.704 | 0.774 | 0.683 | **0.814** | **0.721** |
| NL-75 | 0.795 | 0.678 | 0.790 | 0.671 | **0.848** | **0.774** |
| NL-100 | 0.803 | 0.678 | 0.885 | 0.793 | **0.937** | **0.887** |
| MT1 tax | 0.990 | 0.984 | 0.995 | 0.990 | **0.999** | **0.998** |
| MT1 health | 0.929 | 0.867 | 0.973 | 0.949 | **0.994** | **0.988** |
| MT2 org | 0.981 | 0.963 | 0.987 | 0.978 | **0.994** | **0.988** |
| MT2 sci | 0.977 | 0.961 | 0.990 | 0.984 | **0.995** | **0.992** |
| MT3 art | 0.907 | 0.851 | 0.887 | 0.828 | **0.950** | **0.916** |
| MT3 infra | 0.966 | 0.947 | 0.970 | 0.952 | **0.996** | **0.993** |
| MT4 sci | 0.954 | 0.929 | 0.972 | 0.952 | **0.983** | **0.968** |
| MT4 health | 0.951 | 0.919 | 0.986 | 0.979 | **0.995** | **0.991** |
| Metafam | 0.368 | 0.112 | 0.265 | 0.024 | **0.997** | **0.995** |
| FBNELL | 0.720 | 0.576 | 0.766 | 0.639 | **0.879** | **0.801** |
| **Inductive** $e$ | | | | | | |
| FB-v1 | 0.650 | 0.513 | 0.705 | 0.599 | **0.855** | **0.766** |
| FB-v2 | 0.675 | 0.547 | 0.713 | 0.590 | **0.887** | **0.812** |
| FB-v3 | 0.677 | 0.556 | 0.742 | 0.644 | **0.879** | **0.810** |
| FB-v4 | 0.690 | 0.560 | 0.766 | 0.665 | **0.884** | **0.807** |
| WN-v1 | 0.844 | 0.754 | 0.776 | 0.591 | **0.926** | **0.879** |
| WN-v2 | 0.834 | 0.766 | 0.765 | 0.574 | **0.927** | **0.869** |
| WN-v3 | 0.707 | 0.577 | 0.756 | 0.594 | **0.950** | **0.911** |
| WN-v4 | 0.861 | 0.795 | 0.804 | 0.651 | **0.943** | **0.898** |
| NL-v1 | 0.719 | 0.504 | 0.590 | 0.341 | **0.883** | **0.766** |
| NL-v2 | 0.668 | 0.549 | 0.811 | 0.740 | **0.911** | **0.870** |
| NL-v3 | 0.646 | 0.484 | 0.757 | 0.643 | **0.868** | **0.795** |
| NL-v4 | 0.570 | 0.412 | 0.822 | 0.735 | **0.906** | **0.849** |
| ILPC Small | 0.922 | 0.876 | 0.919 | 0.872 | **0.953** | **0.918** |
| ILPC Large | 0.875 | 0.799 | 0.894 | 0.829 | **0.953** | **0.915** |
| HM 1k | 0.626 | 0.447 | 0.663 | 0.414 | **0.756** | **0.561** |
| HM 3k | 0.592 | 0.439 | 0.664 | 0.418 | **0.790** | **0.623** |
| HM 5k | 0.605 | 0.452 | 0.672 | 0.428 | **0.744** | **0.591** |
| HM Indigo | 0.726 | 0.614 | 0.835 | 0.746 | **0.946** | **0.903** |
| **Transductive** | | | | | | |
| NELL995 | 0.630 | 0.513 | 0.578 | 0.457 | **0.713** | **0.584** |
| NELL23k | 0.688 | 0.571 | 0.755 | 0.658 | **0.869** | **0.805** |
| WDsinger | 0.730 | 0.603 | 0.721 | 0.627 | **0.885** | **0.815** |
| ConceptNet100k | 0.612 | 0.488 | 0.712 | 0.551 | **0.885** | **0.813** |
| CoDExSmall | 0.942 | 0.900 | 0.964 | 0.943 | **0.981** | **0.967** |
| CoDExLarge | 0.907 | 0.850 | 0.908 | 0.845 | **0.973** | **0.950** |
| YAGO310 | 0.930 | 0.891 | 0.826 | 0.666 | **0.970** | **0.942** |
| AristoV4 | 0.254 | 0.201 | 0.498 | 0.381 | **0.651** | **0.547** |
| DBpedia100k | 0.650 | 0.509 | 0.780 | 0.665 | **0.923** | **0.869** |
| Hetionet | 0.737 | 0.646 | 0.922 | 0.862 | **0.942** | **0.897** |
| FB15k-237(10) | 0.688 | 0.550 | 0.795 | 0.711 | **0.940** | **0.905** |
| FB15k-237(20) | 0.695 | 0.558 | 0.846 | 0.778 | **0.958** | **0.931** |
| FB15k-237(50) | 0.728 | 0.618 | 0.903 | 0.858 | **0.970** | **0.948** |
| **Pretrained** | | | | | | |
| FB15k-237 | 0.795 | 0.709 | 0.924 | 0.870 | **0.976** | **0.957** |
| WN18RR | 0.914 | 0.871 | 0.783 | 0.634 | **0.982** | **0.968** |
| CoDExMedium | 0.919 | 0.870 | 0.931 | 0.886 | **0.974** | **0.952** |

Table 13: Dataset statistics for **inductive**-$e, r$ link prediction datasets. Triples are the number of edges given at training, validation, or test graphs, respectively, whereas Valid and Test denote triples to be predicted in the validation and test graphs.

| Dataset | Training Graph | | | Validation Graph | | | | Test Graph | | | |
|---|---|---|---|---|---|---|---|---|---|---|---|
| | Entities | Rels | Triples | Entities | Rels | Triples | Valid | Entities | Rels | Triples | Test |
| FB-25 | 5190 | 163 | 91571 | 4097 | 216 | 17147 | 5716 | 4097 | 216 | 17147 | 5716 |
| FB-50 | 5190 | 153 | 85375 | 4445 | 205 | 11636 | 3879 | 4445 | 205 | 11636 | 3879 |
| FB-75 | 4659 | 134 | 62809 | 2792 | 186 | 9316 | 3106 | 2792 | 186 | 9316 | 3106 |
| FB-100 | 4659 | 134 | 62809 | 2624 | 77 | 6987 | 2329 | 2624 | 77 | 6987 | 2329 |
| WK-25 | 12659 | 47 | 41873 | 3228 | 74 | 3391 | 1130 | 3228 | 74 | 3391 | 1131 |
| WK-50 | 12022 | 72 | 82481 | 9328 | 93 | 9672 | 3224 | 9328 | 93 | 9672 | 3225 |
| WK-75 | 6853 | 52 | 28741 | 2722 | 65 | 3430 | 1143 | 2722 | 65 | 3430 | 1144 |
| WK-100 | 9784 | 67 | 49875 | 12136 | 37 | 13487 | 4496 | 12136 | 37 | 13487 | 4496 |
| NL-0 | 1814 | 134 | 7796 | 2026 | 112 | 2287 | 763 | 2026 | 112 | 2287 | 763 |
| NL-25 | 4396 | 106 | 17578 | 2146 | 120 | 2230 | 743 | 2146 | 120 | 2230 | 744 |
| NL-50 | 4396 | 106 | 17578 | 2335 | 119 | 2576 | 859 | 2335 | 119 | 2576 | 859 |
| NL-75 | 2607 | 96 | 11058 | 1578 | 116 | 1818 | 606 | 1578 | 116 | 1818 | 607 |
| NL-100 | 1258 | 55 | 7832 | 1709 | 53 | 2378 | 793 | 1709 | 53 | 2378 | 793 |
| Metafam | 1316 | 28 | 13821 | 1316 | 28 | 13821 | 590 | 656 | 28 | 7257 | 184 |
| FBNELL | 4636 | 100 | 10275 | 4636 | 100 | 10275 | 1055 | 4752 | 183 | 10685 | 597 |
| Wiki MT1 tax | 10000 | 10 | 17178 | 10000 | 10 | 17178 | 1908 | 10000 | 9 | 16526 | 1834 |
| Wiki MT1 health | 10000 | 7 | 14371 | 10000 | 7 | 14371 | 1596 | 10000 | 7 | 14110 | 1566 |
| Wiki MT2 org | 10000 | 10 | 23233 | 10000 | 10 | 23233 | 2581 | 10000 | 11 | 21976 | 2441 |
| Wiki MT2 sci | 10000 | 16 | 16471 | 10000 | 16 | 16471 | 1830 | 10000 | 16 | 14852 | 1650 |
| Wiki MT3 art | 10000 | 45 | 27262 | 10000 | 45 | 27262 | 3026 | 10000 | 45 | 28023 | 3113 |
| Wiki MT3 infra | 10000 | 24 | 21990 | 10000 | 24 | 21990 | 2443 | 10000 | 27 | 21646 | 2405 |
| Wiki MT4 sci | 10000 | 42 | 12576 | 10000 | 42 | 12576 | 1397 | 10000 | 42 | 12516 | 1388 |
| Wiki MT4 health | 10000 | 21 | 15539 | 10000 | 21 | 15539 | 1725 | 10000 | 20 | 15337 | 1703 |

Table 14: Dataset statistics for **inductive**-$e$ link prediction datasets. Triples are the number of edges given at training, validation, or test graphs, respectively, whereas Valid and Test denote triples to be predicted in the validation and test graphs.

| Dataset | Rels | Training Graph | | Validation Graph | | | Test Graph | | |
|---|---|---|---|---|---|---|---|---|---|
| | | Entities | Triples | Entities | Triples | Valid | Entities | Triples | Test |
| FB-v1 | 180 | 1594 | 4245 | 1594 | 4245 | 489 | 1093 | 1993 | 411 |
| FB-v2 | 200 | 2608 | 9739 | 2608 | 9739 | 1166 | 1660 | 4145 | 947 |
| FB-v3 | 215 | 3668 | 17986 | 3668 | 17986 | 2194 | 2501 | 7406 | 1731 |
| FB-v4 | 219 | 4707 | 27203 | 4707 | 27203 | 3352 | 3051 | 11714 | 2840 |
| WN-v1 | 9 | 2746 | 5410 | 2746 | 5410 | 630 | 922 | 1618 | 373 |
| WN-v2 | 10 | 6954 | 15262 | 6954 | 15262 | 1838 | 2757 | 4011 | 852 |
| WN-v3 | 11 | 12078 | 25901 | 12078 | 25901 | 3097 | 5084 | 6327 | 1143 |
| WN-v4 | 9 | 3861 | 7940 | 3861 | 7940 | 934 | 7084 | 12334 | 2823 |
| NL-v1 | 14 | 3103 | 4687 | 3103 | 4687 | 414 | 225 | 833 | 201 |
| NL-v2 | 88 | 2564 | 8219 | 2564 | 8219 | 922 | 2086 | 4586 | 935 |
| NL-v3 | 142 | 4647 | 16393 | 4647 | 16393 | 1851 | 3566 | 8048 | 1620 |
| NL-v4 | 76 | 2092 | 7546 | 2092 | 7546 | 876 | 2795 | 7073 | 1447 |
| ILPC Small | 48 | 10230 | 78616 | 6653 | 20960 | 2908 | 6653 | 20960 | 2902 |
| ILPC Large | 65 | 46626 | 202446 | 29246 | 77044 | 10179 | 29246 | 77044 | 10184 |
| HM 1k | 11 | 36237 | 93364 | 36311 | 93364 | 1771 | 9899 | 18638 | 476 |
| HM 3k | 11 | 32118 | 71097 | 32250 | 71097 | 1201 | 19218 | 38285 | 1349 |
| HM 5k | 11 | 28601 | 57601 | 28744 | 57601 | 900 | 23792 | 48425 | 2124 |
| HM Indigo | 229 | 12721 | 121601 | 12797 | 121601 | 14121 | 14775 | 250195 | 14904 |

Table 15: Dataset statistics for **transductive** link prediction datasets. Entity task denotes the entity-prediction task: $h/t$ is predicting both heads and tails, and $t$ is predicting only tails.

| Dataset | Entities | Rels | Train | Valid | Test | Entity Task |
|---|---|---|---|---|---|---|
| FB15k-237 | 14541 | 237 | 272115 | 17535 | 20466 | $h/t$ |
| WN18RR | 40943 | 11 | 86835 | 3034 | 3134 | $h/t$ |
| CoDEx Small | 2034 | 42 | 32888 | 1827 | 1828 | $h/t$ |
| CoDEx Medium | 17050 | 51 | 185584 | 10310 | 10311 | $h/t$ |
| CoDEx Large | 77951 | 69 | 551193 | 30622 | 30622 | $h/t$ |
| NELL995 | 74536 | 200 | 149678 | 543 | 2818 | $h/t$ |
| YAGO310 | 123182 | 37 | 1079040 | 5000 | 5000 | $h/t$ |
| WDsinger | 10282 | 135 | 16142 | 2163 | 2203 | $h/t$ |
| NELL23k | 22925 | 200 | 25445 | 4961 | 4952 | $h/t$ |
| AristoV4 | 44949 | 1605 | 242567 | 20000 | 20000 | $h/t$ |
| DBpedia100k | 99604 | 470 | 597572 | 50000 | 50000 | $h/t$ |
| ConceptNet100k | 78334 | 34 | 100000 | 1200 | 1200 | $h/t$ |
| FB15k-237(10) | 11512 | 237 | 27211 | 15624 | 18150 | $t$ |
| FB15k-237(20) | 13166 | 237 | 54423 | 16963 | 19776 | $t$ |
| FB15k-237(50) | 14149 | 237 | 136057 | 17449 | 20324 | $t$ |
| Hetionet | 45158 | 24 | 2025177 | 112510 | 112510 | $h/t$ |

Table 16: Different graph pretraining mix shown in §5.3.

| | 1 | 2 | 3 | 4 | 5 | 6 | 8 |
|---|---|---|---|---|---|---|---|
| FB15k-237 | ✓ | ✓ | ✓ | ✓ | ✓ | ✓ | ✓ |
| WN18RR | | ✓ | ✓ | ✓ | ✓ | ✓ | ✓ |
| CoDEx Medium | | | ✓ | ✓ | ✓ | ✓ | ✓ |
| NELL995 | | | | ✓ | ✓ | ✓ | ✓ |
| YAGO 310 | | | | | ✓ | ✓ | ✓ |
| ConceptNet100k | | | | | | ✓ | ✓ |
| DBpedia100k | | | | | | | ✓ |
| AristoV4 | | | | | | | ✓ |

Table 17: Hyperparameter for FLOCK in pretraining and finetuning setups.

| | Hyperparameter | Entity prediction | Relation prediction |
|---|---|---|---|
| Random walk | Walk length $\ell$ | 128 | 128 |
| | # Pretraining base walk $n_{\text{train}}$ | 128 | 128 |
| | # Test-time or finetuning base walk $n$ | 16–512 | 16–512 |
| Sequence processor | # Layers | 1 | 1 |
| | Hidden dimension | 64 | 64 |
| Consensus protocol | # Heads $h$ | 4 | 4 |
| | Head dimension $d_h$ | 16 | 16 |
| Update | # Update step $I$ | 6 | 6 |
| Ensemble | # Maximum ensembled passes $P$ | 16 | 16 |
| Pretraining | Optimizer | AdamW | AdamW |
| | Learning rate | 0.0005 | 0.0005 |
| | Training steps | 400,000 | 40,000 |
| | Adversarial temperature | 1 | 1 |
| | # Negatives | 512 | 512 |
| | Batch size | 8 | 8 |
| | Weight decay | 0.01 | 0.00 |
| Finetuning | Optimizer | AdamW | AdamW |
| | Learning rate | 0.0005 | 0.0005 |
| | Adversarial temperature | 1 | 1 |
| | # Negatives | 256 | 256 |
| | Batch size | 4–32 | 4–8 |

Table 18: Detailed finetuning and inference hyperparameters for FLOCK in entity prediction. For each dataset, we report the finetuning epochs, batches per epoch, batch size, and the inference settings for both zero-shot and finetuned modes: test-time ensemble size $P$, base walk count $n$. For Hetionet finetuning we used $(P, n) = (1, 1024)$, instead of $(2, 512)$ as in zero-shot.

| Dataset | Epoch | # Batch/Epoch | Batch Size | # Ensembled Passes $P$ | # Base Walk $n$ |
|---|---|---|---|---|---|
| FB15k-237 | 1 | full | 8 | 16 | 128 |
| WN18RR | 1 | full | 8 | 16 | 128 |
| CoDEx Small | 1 | full | 32 | 16 | 16 |
| CoDEx Medium | 1 | full | 8 | 16 | 128 |
| CoDEx Large | 1 | 2000 | 4 | 2 | 512 |
| NELL-995 | 1 | full | 8 | 16 | 128 |
| YAGO310 | 1 | 2000 | 4 | 8 | 512 |
| WDsinger | 1 | full | 8 | 16 | 16 |
| NELL23k | 3 | full | 8 | 16 | 32 |
| FB15k-237(10) | 1 | full | 8 | 16 | 32 |
| FB15k-237(20) | 1 | full | 8 | 16 | 64 |
| FB15k-237(50) | 1 | full | 8 | 16 | 64 |
| Hetionet | 1 | 4000 | 8 | 2 | 512 |
| DBpedia100k | 1 | 1000 | 4 | 2 | 512 |
| AristoV4 | 1 | full | 8 | 4 | 256 |
| ConceptNet100k | 1 | full | 8 | 16 | 128 |
| FB v1–v4 | 1 | full | 8 | 16 | 16 |
| WN v1–v4 | 1 | full | 8 | 16 | 16 |
| NL v1–v4 | 3 | full | 8 | 16 | 16 |
| ILPC Small | 1 | full | 8 | 16 | 16 |
| ILPC Large | 1 | full | 8 | 16 | 64 |
| FB 25–100 | 3 | full | 8 | 16 | 16 |
| WK 25–100 | 3 | full | 8 | 16 | 16 |
| NL 0–100 | 3 | full | 8 | 16 | 16 |
| Wiki MT1 tax | 3 | full | 8 | 16 | 16 |
| Wiki MT1 health | 3 | full | 8 | 16 | 16 |
| Wiki MT2 org | 3 | full | 16 | 16 | 32 |
| Wiki MT2 sci | 3 | full | 8 | 16 | 16 |
| Wiki MT3 art | 3 | full | 16 | 16 | 32 |
| Wiki MT3 infra | 3 | full | 16 | 16 | 32 |
| Wiki MT4 sci | 3 | full | 8 | 16 | 16 |
| Wiki MT4 health | 3 | full | 8 | 16 | 16 |
| Metafam | 3 | full | 8 | 16 | 16 |
| FBNELL | 3 | full | 8 | 16 | 16 |
| HM 1k | 1 | full | 8 | 16 | 16 |
| HM 3k | 1 | full | 16 | 16 | 32 |
| HM 5k | 1 | full | 8 | 16 | 64 |
| HM Indigo | 1 | full | 8 | 16 | 128 |

Table 19: Detailed finetuning and inference hyperparameters for FLOCK in relation prediction. For each dataset, we report the finetuning epochs, batches per epoch, batch size, and the inference settings for both zero-shot and finetuned modes: test-time ensemble size $P$ and base walk count $n$.

| Dataset | Epoch | # Batch/Epoch | Batch Size | # Ensembled Passes $P$ | # Base Walk $n$ |
|---|---|---|---|---|---|
| FB15k-237 | 1 | 1000 | 8 | 16 | 128 |
| WN18RR | 1 | 1000 | 8 | 16 | 128 |
| CoDEx Small | 1 | 1000 | 8 | 16 | 16 |
| CoDEx Medium | 1 | 1000 | 8 | 16 | 128 |
| CoDEx Large | 1 | 1000 | 4 | 2 | 512 |
| NELL-995 | 1 | 1000 | 8 | 16 | 128 |
| YAGO310 | 1 | 1000 | 8 | 4 | 512 |
| WDsinger | 1 | 1000 | 8 | 16 | 16 |
| NELL23k | 1 | 1000 | 8 | 16 | 32 |
| FB15k-237(10) | 1 | 1000 | 8 | 16 | 32 |
| FB15k-237(20) | 1 | 1000 | 8 | 16 | 64 |
| FB15k-237(50) | 1 | 1000 | 8 | 16 | 64 |
| Hetionet | 1 | 1000 | 4 | 2 | 512 |
| DBpedia100k | 1 | 1000 | 4 | 2 | 512 |
| AristoV4 | 1 | 1000 | 8 | 4 | 256 |
| ConceptNet100k | 1 | 1000 | 8 | 16 | 128 |
| FB v1–v4 | 1 | 1000 | 8 | 16 | 16 |
| WN v1–v4 | 1 | 1000 | 8 | 16 | 16 |
| NL v1–v4 | 1 | 1000 | 8 | 16 | 16 |
| ILPC Small | 1 | 1000 | 8 | 16 | 16 |
| ILPC Large | 1 | 1000 | 8 | 16 | 64 |
| FB 25–100 | 1 | 1000 | 8 | 16 | 16 |
| WK 25–100 | 1 | 1000 | 8 | 16 | 16 |
| NL 0–100 | 1 | 1000 | 8 | 16 | 16 |
| Wiki MT1 tax | 1 | 1000 | 8 | 16 | 16 |
| Wiki MT1 health | 1 | 1000 | 8 | 16 | 16 |
| Wiki MT2 org | 1 | 1000 | 8 | 16 | 32 |
| Wiki MT2 sci | 1 | 1000 | 8 | 16 | 16 |
| Wiki MT3 art | 1 | 1000 | 8 | 16 | 32 |
| Wiki MT3 infra | 1 | 1000 | 8 | 16 | 32 |
| Wiki MT4 sci | 1 | 1000 | 8 | 16 | 16 |
| Wiki MT4 health | 1 | 1000 | 8 | 16 | 16 |
| Metafam | 1 | 1000 | 8 | 16 | 16 |
| FBNELL | 1 | 1000 | 8 | 16 | 16 |
| HM 1k | 1 | 1000 | 8 | 16 | 16 |
| HM 3k | 1 | 1000 | 8 | 16 | 32 |
| HM 5k | 1 | 1000 | 8 | 16 | 64 |
| HM Indigo | 1 | 1000 | 8 | 16 | 128 |

