# OpenReview forum: "Flock: A Knowledge Graph Foundation Model via Learning on Random Walks"
_ICLR.cc/2026/Conference — ICLR 2026 Poster_

### Official Review · Reviewer_sNAr · 2025-10-31

**Soundness:** 3
**Presentation:** 3
**Contribution:** 3
**Rating:** 4
**Confidence:** 4

**Summary:**

This paper introduces FLOCK, a novel knowledge graph foundation model for zero-shot link prediction. The authors identify that existing knowledge graph foundation models (KGFMs) rely on deterministic node-relation equivariance, which limits their ability to distinguish between structurally similar but semantically different relations. To address this issue, the authors introduce the concept of probabilistic node-relation equivariance and design FLOCK based on this principle. FLOCK iteratively samples random walks from KGs, encodes them into sequences, embeds them with a sequence model, and aggregates node and relation representations through learned pooling.

**Strengths:**

1. The paper's concept of probabilistic node-relation equivariance extends the theoretical framework of knowledge graph foundation models and provides detailed proofs showing FLOCK is a universal approximator for link-level functions.

2. The authors clearly identify limitations of existing KGFMs, particularly in handling structurally similar but semantically different relations, and provide an intuitive Star Wars character relationship example (Figure 1).

3. The paper presents extensive evaluations on 54 KGs across different domains, including both entity prediction and relation prediction tasks, in both zero-shot and fine-tuned settings, with FLOCK outperforming state-of-the-art models in most cases.

**Weaknesses:**

1. While the authors highlight FLOCK's significant advantage over ULTRA and TRIX on the Metafam dataset in lines 288-290, Table 10's fine-tuning results show all three models achieving nearly 1.000 MRR, contrasting with the significant differences in the zero-shot setting. This inconsistency is not explained.

2. While Section 4.2 provides a thorough theoretical analysis, the paper doesn't adequately discuss how these theoretical advantages directly explain specific performance differences observed in experiments, particularly across different types of knowledge graphs.

3. On certain datasets (e.g., WK-50, Table 8 line 1570+), FLOCK performs worse than ULTRA and TRIX, contradicting the overall conclusions, but these anomalies aren't analyzed or explained.

**Questions:**

1. Given the random walk coverage issues, have you explored more targeted random walk strategies rather than uniform random walks? For example, could biased random walks based on graph structure or query information improve model performance?

2. In the transductive setting, FLOCK underperforms baseline models, which seems to contradict the model's theoretical advantages. Could you provide a deeper analysis of this phenomenon and how this issue might be resolved while maintaining the model's expressiveness?

---

> ### Author Response · Authors · 2025-11-23
>
> >“W1. While the authors highlight FLOCK's significant advantage over ULTRA and TRIX on the Metafam dataset in lines 288-290, Table 10's fine-tuning results show all three models achieving nearly 1.000 MRR, contrasting with the significant differences in the zero-shot setting. This inconsistency is not explained.”
>
> Thank you for the comment. As the reviewer mentioned and as we discussed in **Lines 414-418**, Flock’s advantage in Metafam is significant in zero-shot generalization. Meanwhile, in-domain fine-tuning allows all models to reach a high accuracy. We believe this is explained by two reasons. The first explains why fine-tuning allows all models to reach a high accuracy, and the second explains why zero-shot prediction is hard and the expressive power benefits it.
>
>  1. Link prediction in Metafam becomes relatively easy if one can learn from the relation in the training graph because the training-test distribution overlap is high. Specifically, Metafam is built from a fixed family-relations ontology, where one is asked to predict one of {mother_of, father_of, daughter_of, son_of} in the training dataset, and is asked to predict one of {mother_of, father_of} in the test dataset. Up to symmetries of genders, this means only two predictive patterns (parent_of and child_of) are effectively present in the training data, and the test data asks only a single predictive pattern (parent_of) [1], making the task simple when in-domain fine-tuning is allowed.
>
>  2. If one cannot learn from the relation in the training graph (zero-shot generalization setup), link prediction in Metafam becomes harder and benefits from expressive power. This is because the relations exhibit a nontrivially high degree of structural symmetries (Figure 1 of [3]), making it challenging to distinguish and reason about them in a zero-shot manner. For example, it is known that the relations father_of, son_of, sister_of, niece_of, grandson_of, … are not clearly separated in the representation space of ULTRA [3], although separating son_of and father_of is especially critical for correct predictions due to their conflicting predictive patterns [1]. Here, Flock’s high expressive power, offered by probabilistic equivariance, can offer a natural inductive bias to leverage more clearly distinguished relation representations during pretraining.
>
> To further demonstrate this, we have included the visualization of relation embeddings in **Figure 6** and discussed them in **Appendix G**. We observe that thanks to the expressivity of Flock, it is able to distinguish between father_of, and son_of in the Metafam datasets, unlike ULTRA, where they are assigned to almost identical embeddings. This precisely suggests why Metafam is a good example for showcasing the benefit of increased expressivity.
>
> >“W2. While Section 4.2 provides a thorough theoretical analysis, the paper doesn't adequately discuss how these theoretical advantages directly explain specific performance differences observed in experiments, particularly across different types of knowledge graphs.” (...) “W3. On certain datasets, FLOCK performs worse than ULTRA and TRIX, contradicting the overall conclusions, but these anomalies aren't analyzed or explained.”
>
> Thank you for the constructive comment. In the case of Metafam, we believe the theoretical advantages do explain the zero-shot performance gain, as we have discussed in the response to **W1** and supplemented with empirical evidence in **Appendix G**.
>
> For the other general KGs, we conducted an additional analysis and were able to obtain a holistic explanation that spans both success and failure cases, which also agrees well with our theory. The analysis is given in **Appendix H**, where we find that **Flock tends to perform well on sparse KGs, while less so on dense KGs**, and the tendency is statistically significant. Interestingly, this agrees with our theoretical analysis in **Section 4.2**, in which a necessary condition for universality is that the random walk covers all edges of a KG with a high probability. The time taken until covering all edges is called the edge cover time, and it is known to be $O(|V||E|)$ for uniform walks, which is proportional to the density of a graph [4]. This suggests the performance of Flock is associated with the ease of visiting as many edges as possible rapidly, which is more challenging for dense KGs. Furthermore, this analysis is consistent with the observations of recent works on graph learning based on random walks (see **Section 6** of [5], and also [6]).
>
> We believe this result provides a theoretically grounded analysis and explanation of the performances of Flock across different types of KGs. It also suggests potential improvement strategies, for example, based on alternative recording functions that record edges faster [6]. We leave this as a promising direction for future work.

---

> ### Author Response · Authors · 2025-11-23
>
> >“Q1. Given the random walk coverage issues, have you explored more targeted random walk strategies rather than uniform random walks? For example, could biased random walks based on graph structure or query information improve model performance?”
>
> Thank you for the question. We use non-backtracking uniform random walks (**Lines 231-234**), which in fact introduce nontrivial bias compared to uniform walks, linked to the spectral properties of the graph structure. In particular, it is known that non-backtracking walks can have mixing rates up to twice as fast as the uniform walks [2]. Indeed, we can show that this factor is already contributing to the performance of Flock under the hood.
>
> To demonstrate this, we ran an ablation experiment by allowing random walks to backtrack (i.e., uniform walks). The results are in **Table A**, showing that uniform random walks show significantly degraded performances on large, transductive KGs. This indicates that biasing random walks to cover a KG more rapidly can indeed improve performance, and Flock is already leveraging that property through non-backtracking.
>
> ---
>
> **Table A.** Additional ablation study results (MRR/Hits@10).
>
> | **Model**            | **Inductive** $(e,r)$ | **Inductive** $e$ | **Transductive\*** | **Average**        |
> |----------------------|-----------------------|-------------------|--------------------|--------------------|
> |                      | MRR / H@10            | MRR / H@10        | MRR / H@10         | MRR / H@10         |
> | Flock (Ours)         | 0.369 / 0.554         | 0.456 / 0.604     | 0.360 / 0.542      | **0.395 / 0.567**  |
> | w/o non-backtracking | 0.370 / 0.549         | 0.456 / 0.605     | 0.334 / 0.499      | 0.386 / 0.551      |
>
> \* NELL995, NELL23k, WDsinger, ConceptNet100k, YAGO310 tested due to resource limit.
>
>
> ---

---

> ### Author Response · Authors · 2025-11-23
>
> >“Q2. In the transductive setting, FLOCK underperforms baseline models, which seems to contradict the model's theoretical advantages. Could you provide a deeper analysis of this phenomenon and how this issue might be resolved while maintaining the model's expressiveness?”
>
> We would like to point out that the claim *“In the transductive setting, Flock underperforms baseline models”* is technically not correct, as Flock outperforms TRIX/ULTRA in zero-shot entity prediction as well as in relation prediction in the transductive setting (**Tables 1 & 2**), and is only outperformed by TRIX in fine-tuned entity prediction (**Table 2**). This does **not** contradict our theoretical results, which are about invariance, expressivity, and structural distinguishability, rather than guaranteeing dominance in fully in-domain (fine-tuned) transductive training.
>
> For a deeper analysis of the fine-tuned transductive setting, we refer to the KG-wise performances given in **Table 10**. We find that Flock attains the best performance in 6 out of 13 cases, while TRIX attains the best in 3 cases, and ULTRA attains the best in 4 cases. In other words, Flock actually wins the most frequently, but is affected by less frequent failure cases, specifically NELL23k, WDsinger, and Hetionet.
>
> - For Hetionet, we conjecture that one reason for failure could be underfitting. This is because Hetionet is a large KG (45k nodes and 2M edges) that also has a very large training set (2M triples), and we were restricted to 4,000 fine-tuning steps due to resource constraints.
>
> - For NELL23k and WDsinger, the cause could be more interesting, as they have been constructed in [7] using a shared methodology. We conjecture the reason could be related to the fact that the KGs in [7] were initially designed for tail-prediction, but recent benchmarks use them to evaluate both head and tail predictions [8]. This may lead to ill-defined tasks, e.g., for WDsinger, an example triple is (singer, place_of_birth, place); in this case, tail prediction (singer, place_of_birth, ?) is well-defined, but head prediction (?, place_of_birth, place) is not.
>
>   To supplement this hypothesis, we ran an additional evaluation for NELL23k and WDsinger for tail prediction. The results are in **Table B**, showing that Flock closes the gap. We conjecture that broad coverages of message passing in the baseline methods might enable them to solve ill-posed head prediction tasks to some degree, e.g., by enumerating all potential heads and making a guess, although such tasks could be practically less meaningful.
>
> ---
>
> **Table B.** NELL23k and WDsinger tail prediction results (MRR/Hits@10).
>
> | **Model** | **NELL23k ($h/t$)** | **WDsinger ($h/t$)** | **NELL23k (tail)** | **WDsinger (tail)** |
> |--|-|-|-|-|
> | | MRR / H@10 | MRR / H@10 | MRR / H@10 | MRR / H@10 |
> | Flock (Ours) | 0.233 / 0.398 | 0.410 / 0.528 | 0.286 / 0.498 | 0.492 / 0.620 |
> | ULTRA | 0.239 / 0.408 | 0.382 / 0.498 | 0.299 / 0.511 | 0.470 / 0.587 |
> | TRIX | 0.290 / 0.497 | 0.511 / 0.609 | 0.286 / 0.492 | 0.481 / 0.608 |
>
> ---
>
> *[1] Zhou et al. A multi-task perspective for link prediction with new relation types and nodes. arXiv preprint arXiv:2307.06046, 2023.*
>
> *[2] Alon et al. Non-backtracking random walks mix faster. arXiv preprint arXiv:math/0610550, 2006.*
>
> *[3] Arun et al: A semantic aware knowledge graph foundation model, EMNLP, 2025.*
>
> *[4] Zuckerman, On the time to traverse all edges of a graph, Information Processing Letters 1991.*
>
> *[5] Wang & Cho, Non-convolutional graph neural networks, NeurIPS 2024.*
>
> *[6] Kim et al., Revisiting random walks for learning on graphs, ICLR 2025.*
>
> *[7] Lv et al., Dynamic anticipation and completion for multi-hop reasoning over sparse knowledge graph, arXiv 2020.*
>
> *[8] Galkin et al., Towards foundation models for knowledge graph reasoning, ICLR 2024.*

---

> ### Comment · Reviewer_sNAr · 2025-11-26
> **update after rebuttal**
>
> Thanks for the authors' rebuttal. The author addressed most of my concerns, so I decided to raise my rating to “6: marginally above the acceptance threshold. But would not mind if paper is rejected”.

---

> > ### Author Response · Authors · 2025-11-26
> >
> > We thank the reviewer for going through our rebuttal and for raising their score. If there are any remaining issues, we would be grateful to hear them since this will give us a chance to further clarify them.

---

> > ### Author Response · Authors · 2025-11-28
> >
> > We thank the reviewer for raising their score from 4 to 6 following our rebuttal.

---

### Official Review · Reviewer_w7Ej · 2025-10-31

**Soundness:** 2
**Presentation:** 3
**Contribution:** 3
**Rating:** 6
**Confidence:** 4

**Summary:**

This work argues that existing Knowledge Graph Foundation Models (KGFMs) are limited by deterministic node-relation equivariance, preventing them from distinguishing structurally isomorphic yet semantically distinct relations. They propose probabilistic node-relation equivariance as a relaxation and introduce FLOCK, a KGFM based on sampling and encoding anonymized random walks using sequence models, without using traditional message-passing. FLOCK is claimed to respect probabilistic equivariance and be a universal approximator for link-invariant functions. They also design a diagnostic dataset (PETALS) to showcase the limitations of prior work, and showcase empirical results suggesting state-of-the-art performance on the standard benchmark of the domain with 54 KGs. Overall, I think this work makes a valuable contribution, and I would be happy to increase my score if my concerns are addressed.

**Strengths:**

1. **Problem:** Identifies and demonstrates a clear limitation (expressivity with respect to structurally isomorphic relations) of KGFMs based on strict equivariance using a new synthetic dataset (PETALS).
2. **Elegant solution:** Proposes probabilistic node-relation equivariance as a potentially more expressive alternative inductive bias and introduces FLOCK, a novel non-message-passing KGFM architecture based on random walks and sequence models.
3. **Theoretical backing:** Provides theoretical analysis regarding universality and probabilistic invariance (though practical relevance is debatable).
4. **Empirical Backing:** Shows strong empirical performance on a wide range of KGs, particularly in relation prediction and on the PETALS set.

**Weaknesses:**

1. **Scalability:** The reliance on random walks raises concerns for large graphs, both in terms of sampling time and ensuring adequate graph coverage. The proposed test-time adaptation is heuristic. Efficiency comparisons (Table 6) show FLOCK can be much slower than baselines.
2. **Practicality:** The theoretical benefits (universality, distinguishing specific isomorphic cases) might not translate into significant practical advantages across all common KG structures and tasks, especially given the variance introduced by stochasticity. Although the results across the standard benchmark suggested improvement, prior work [1, 2] has shown that this benchmark has potential issues. FLOCK's pipeline (sampling, recording, sequence processing, consensus) is also more complex than standard message passing approaches, potentially hindering adoption.
3. **Ablation Gaps:** More detailed ablation (like Figure 4b) on the random walk strategy (like length, different samplers) and sequence model architecture would be beneficial.

---

*[1] Harry Shomer, Jay Revolinsky, & Jiliang Tang (2025). Towards Better Benchmark Datasets for Inductive Knowledge Graph Completion. In Proceedings of the 31st ACM SIGKDD Conference on Knowledge Discovery and Data Mining (KDD).*

*[2] Arvindh Arun, Sumit Kumar, Mojtaba Nayyeri, Bo Xiong, Ponnurangam Kumaraguru, Antonio Vergari, & Steffen Staab. (2025). SEMMA: A Semantic Aware Knowledge Graph Foundation Model. In The Thirtieth Conference on Empirical Methods in Natural Language Processing (EMNLP).*

**Questions:**

1. Beyond PETALS, can you provide examples from the benchmark KGs where the ability to distinguish structurally isomorphic relations proved crucial for FLOCK's performance advantage? How common is this structural ambiguity in real KGs?
2. The universality proof sketch relies on walks covering all edges. How does the approximation quality degrade in practice when walks inevitably provide only partial coverage on large graphs? Does this partial coverage limit the types of functions FLOCK can effectively learn?
3. Given the computational overhead of sampling and ensembling, under what specific conditions (graph size, structure, task requirements) does FLOCK offer a clear practical advantage over potentially faster, deterministic KGFMs like TRIX or ULTRA, especially if these were to be augmented with techniques to increase expressivity (subgraph features or textual semantics like [2])? SEMMA [2] approaches the same problem of “losing the ability to distinguish between two entities with opposite relationships”, but by directly encoding the textual features into ULTRA. This at least warrants a discussion.
4. How was the sequence model architecture (biGRU + RMSNorm + SwiGLU) chosen? Did you experiment with other paradigms (like Transformers or SSMs), and how did they perform in terms of accuracy and efficiency?
5. Could the proposed probabilistic equivariance be achieved through methods other than random walks (like stochastic message passing, randomized node features)? How does the choice of random walks specifically contribute? Appendix F is great, but more than naive noise injection would make it stronger.

---

> ### Author Response · Authors · 2025-11-23
>
> >“W1. Scalability: The reliance on random walks raises concerns for large graphs, both in terms of sampling time and ensuring adequate graph coverage. The proposed test-time adaptation is heuristic. Efficiency comparisons (Table 6) show FLOCK can be much slower than baselines.”
>
> Thank you for the comment. We acknowledged the challenge of scaling random walks to large KGs in **Section 6**. Currently, Flock’s cost scales proportionally to the walk counts and lengths, instead of the full KG sizes, and we can leverage this to induce a cost-performance trade-off that helps handle large KGs. For example, we clamp the walk counts to 512 at maximum, which allows us to scale to the largest transductive KGs such as YAGO310 (**Tables 15, 18, 19**). While this approach, and more broadly our proposed test-time adaptation, is a heuristic as the reviewer pointed out, we believe it is a useful one in practice.
>
> At the same time, we believe this scalability and efficiency challenge has avenues for mitigation in the future. For example, the compared message passing baselines (ULTRA and TRIX) leverage highly optimized fused sparse kernels [1,2], and we believe Flock could benefit from similar dedicated optimizations, such as efficient approximations of random walks that have been historically of significant interest in the distributed and parallel computing literature ([3]). We view this as a promising direction for future work.
>
>
> >“W2. Practicality: The theoretical benefits (universality, distinguishing specific isomorphic cases) might not translate into significant practical advantages across all common KG structures and tasks, especially given the variance introduced by stochasticity.”
>
> While we acknowledge that theoretical properties may not translate to empirical gains in **all** datasets, we do observe consistent empirical improvements on the standard benchmarks, with particularly strong gains on datasets where structural reasoning is crucial (e.g. Metafam and Petals).
>
> Regarding the variance introduced by stochasticity, we ran an additional test by repeating zero-shot entity prediction on inductive KGs three times. The results are in **Table A**. We observe that the standard deviations over repeated trials are small, indicating a consistent prediction performance. This is partially due to the fact that we are already ensembling 16 randomized predictions of Flock at test-time, which has a variance reduction effect in addition to performance improvement. While we could not run repeated experiments for all tables due to the resource constraints, we will do our best to include error bars as much as possible in the final version to account for this concern.
>
> ---
>
> **Table A.** Performances (MRR/Hits@10) of Flock over three repeated tests. TRIX included for comparison.
>
> | **Model** | **Inductive $e,r$ (23 KGs)** | **Inductive $e$ (18 KGs)** |
> |-----------------------|-------------------------------------|-----------------------------------|
> | | MRR / H@10 | MRR / H@10 |
> | TRIX | 0.368 / 0.540 | 0.455 / 0.592 |
> | Flock trial 1 | 0.369 / 0.554 | 0.456 / 0.604 |
> | Flock trial 2 | 0.369 / 0.555 | 0.456 / 0.604 |
> | Flock trial 3 | 0.370 / 0.550 | 0.456 / 0.603 |
> | Flock average and std | **0.369 ± 0.000 / 0.553 ± 0.003** | **0.456 ± 0.000 / 0.604 ± 0.000** |
>
> ---

---

> ### Author Response · Authors · 2025-11-23
>
> > “Although the results across the standard benchmark suggested improvement, prior work [1, 2] has shown that this benchmark has potential issues.”
>
> While the 54 KGs used in the paper is a standard benchmark setup which we believe is useful for comparing against previous methods, we ran additional tests involving 31 KGs based on the references [4, 5] given by the reviewer. The results in **Tables B and C** support that Flock performs robustly in these more challenging setups, supporting that it is capable of genuine generalization.
>
> - **22 KGs without pretraining-test leakage.** [5] has observed overlaps between triples in the pretraining and test KGs in the standard benchmark setup that uses 54 KGs, and identified a challenging subset of 22 KGs free of the overlaps. We tested ULTRA, TRIX, and Flock on these KGs and observed that Flock outperforms previous methods often by a significant margin, with only one exception of TRIX in the partially inductive setting (inductive e). The result implies that Flock is capable of generalizing to genuinely novel domains not in the pretraining KGs.
>
> ---
>
> **Table B**. Zero-shot MRR/Hits@10 over 22 KGs from [5].
> | **Model**        | **Inductive** $e,r$ (8 graphs) | **Inductive** $e$ (7 graphs) | **Transductive** (7 graphs) | **Total average (22 graphs)** |
> |--------------|--------------------------|-------------------------|--------------------------|----------------------------|
> | ULTRA        | 0.389 / 0.589           | 0.348 / 0.468          | 0.288 / 0.424           | 0.344 / 0.498             |
> | SEMMA        | 0.399 / 0.589           | 0.357 / 0.482          | 0.295 / 0.441           | 0.353 / 0.508             |
> | SEMMA-H      | 0.406 / 0.600           | 0.357 / 0.484          | 0.298 / 0.445           | 0.356 / 0.514             |
> | TRIX         | 0.411 / 0.620           | **0.382** / 0.504          | 0.321 / 0.487           | 0.373 / 0.541             |
> | Flock (Ours) | **0.414** /**0.649**           | 0.370 / **0.512**          | **0.342** / **0.517**           | **0.377** / **0.564**             |
>
> ---
>
> - **9 KGs without prediction shortcuts.** [5] has proposed 9 new KGs for inductive link prediction with a removal of potential shortcuts that can be exploited for prediction while ignoring relational information. We ran a preliminary test on these KGs and observed that Flock outperforms both ULTRA and TRIX in both MRR and Hits@10. The result supports that Flock is capable of performing genuine KG reasoning without exploiting shortcuts.
>
> ---
>
> **Table C.** Zero-shot MRR and Hits@10 over 9 KGs from [5].
> | Model        | MRR   | Hits@10 |
> |--------------|-------|---------|
> | ULTRA        | 0.494 | 0.658   |
> | TRIX         | 0.503 | 0.669   |
> | Flock (Ours) | **0.510** | **0.674**   |
>
> ---
>
> We appreciate the reviewer’s comment and will incorporate the additional results into the next revision of the paper.
>
> > “FLOCK's pipeline (sampling, recording, sequence processing, consensus) is also more complex than standard message passing approaches, potentially hindering adoption.”
>
> While Flock may look more complicated on the surface, it essentially consists of two components: a random-walk extractor and a neural sequence processor, which is aligned with previous literature on random-walk-based graph representation learning, like DeepWalk, Node2Vec, and CraWl. Our experiments show that relatively simple choices of these two components are sufficient to achieve a strong performance. Unlike prior methods, Flock requires neither auxiliary relation graphs nor advanced message-passing techniques (which arguably are not elementary themselves). Its pipeline: sample random walks, anonymize, process as sequences, and predict, relies on standard, well-understood algorithms and may even be more approachable for an external reader.

---

> ### Author Response · Authors · 2025-11-23
>
> >“W3. Ablation Gaps: More detailed ablation (like Figure 4b) on the random walk strategy (like length, different samplers) and sequence model architecture would be beneficial.”
>
> Thank you for the constructive comment. We agree that these detailed ablation studies would be beneficial. Following the suggestion, we conducted an additional, detailed ablation study spanning the following axes: (1) random walk length and sampler (non-backtracking), (2) consensus protocol, and (3) sequence processor.
>
> The results are provided in **Table D**. The results on random walk length and sampler are interesting and warrant a meaningful discussion. In general, the choices of random walk sampling induce a trade-off between the coverage of KGs and computational/reasoning efficiency for the sequence models. For example, for random walks that are too localized (e.g., allowed to backtrack, or with short lengths), we expect the performances to degrade on large KGs (e.g., ones found in transductive setups) due to the coverage issues; on the other hand, for too long random walks, we expect computational inefficiency and learnability challenges with sequence neural networks (we remark that our GRU has 64 hidden dimensions). Hence, the best length needs to be balanced between these two factors. Indeed, the results in **Table D** show that backtracking or shorter walks are easier to learn for the sequence network but lead to degraded performances for large transductive KGs; on the other hand, longer walks pose efficiency and learnability issues. The results show that our choice of 128-length walks provides a good balance, keeping in mind the predictive power and scalability. In addition, ablations of the consensus protocol show that it is indeed contributing significantly to the performance of Flock.
>
> We have also conducted additional experiments using the RMSNorm-SwiGLU transformer with a similar parameter count.  This alternative, although it is relatively faster in both training and evaluation due to parallelization, does not deliver good results, which is explained by the restrictions on model scales that are enforced to scale to large KGs. Flock benefits from the reasoning efficiency of GRU in a limited parameter regime, gaining good performance and scalability together.
>
> We appreciate the comment and have added the result to the next revision of the paper.
>
> ---
>
> **Table D.** Additional ablation study results (MRR/Hits@10).
>
> | **Model** | **Inductive** $(e,r)$ | **Inductive** $e$ | **Transductive\*** | **Average** |
> |-----------------------|-----------------------|-------------------|--------------------|-------------------|
> | | MRR / H@10 | MRR / H@10 | MRR / H@10 | MRR / H@10 |
> | Flock (Ours) | 0.369 / 0.554 | 0.456 / 0.604 | 0.360 / 0.542 | **0.395 / 0.567** |
> | w/o non-backtracking | 0.370 / 0.549 | 0.456 / 0.605 | 0.334 / 0.499 | 0.386 / 0.551 |
> | walk length 0.5x | 0.372 / 0.556 | 0.459 / 0.606 | 0.351 / 0.534 | 0.394 / 0.565 |
> | walk length 2x | 0.360 / 0.548 | 0.458 / 0.605 | 0.338 / 0.508 | 0.385 / 0.553 |
> | w/o consensus | 0.351 / 0.526 | 0.448 / 0.593 | 0.361 / 0.515 | 0.387 / 0.545 |
> | transformer $f_\theta$ | 0.356 / 0.542 | 0.410 / 0.591 | 0.312 / 0.477 | 0.359 / 0.537 |
>
> \* NELL995, NELL23k, WDsinger, ConceptNet100k, YAGO310 tested due to resource limit.
>
> ---

---

> ### Author Response · Authors · 2025-11-23
>
> >“Q1. Beyond PETALS, can you provide examples from the benchmark KGs where the ability to distinguish structurally isomorphic relations proved crucial for FLOCK's performance advantage? How common is this structural ambiguity in real KGs?”
>
> Thank you for the comment. One good example from the benchmark KGs is Metafam [4]. It is built from a fixed family-relations ontology, and one is asked to answer link prediction queries with relations {mother\_of, father\_of, daughter\_of, son\_of}. Up to symmetries of genders, this means two predictive patterns (parent\_of and child\_of), that are semantically distinct (conflicting), are present [4]. This, in fact, is a challenging setup, as the relations in Metafam are known to exhibit a nontrivially high degree of structural symmetries (Figure 1 of [5]), making it challenging to distinguish and reason about them in a zero-shot manner. For example, it is known that the relations father\_of, son\_of, sister\_of, niece\_of, grandson\_of, … are not clearly separated in the representation space of ULTRA, although separating son\_of and father\_of is especially critical for correct predictions due to their conflicting predictive patterns.
>
> Here, Flock’s high expressive power, offered by probabilistic equivariance, can offer a natural inductive bias to leverage more clearly distinguished relation representations. To verify this, we have included the visualization of relation embeddings in **Figure 6** and discussed them in **Appendix G**. We observe that thanks to the expressivity of Flock, it is able to distinguish between father\_of, and son\_of in the Metafam datasets, unlike ULTRA, where they are assigned to almost identical embeddings. This precisely suggests that the ability to distinguish structurally similar (approximately isomorphic) relations in a zero-shot manner underlies Flock’s performance advantage on this dataset.
>
> While we have provided an illustrative example in **Figure 1**, in real KGs, we believe there are indeed practical cases of structural ambiguities. We can think of semantically distinct relations that connect (structurally) similar sets of entities in similar ways. An example is CEO\_of, CFO\_of, and founder\_of. They are structurally similar because they one-to-one connect a person to a company, and the connected sets of entities would be similar, as the sets of CEOs, CFOs, and founders have a meaningful overlap due to involvement in major corporate events. However, they are semantically distinct, as founder\_of is a static historical fact, and CEO\_of (overall leadership) and CFO\_of (financial management) are current roles with distinct responsibilities. Another example is CEO\_of (a company) and president\_of (a university). Here, the sets of entities are different but have meaningfully shared structural features, as they are both large organizations with locations, departments, and members. We can think of other examples as well, such as {headquartered\_in, founded\_in} that connect a company to a location, {located\_in, jurisdiction\_of} that connect an entity to geographical or political body, and {succeeded\_by, caused\_by} that connect two temporally ordered events. With our theoretical results and empirical analysis in **Appendix G**, we can expect that Flock would have advantages in handling these relations in a generalizable manner, thanks to its high expressive power offered by probabilistic equivariance.
>
> We appreciate the reviewer’s comment and will add the above discussion to the next revision of the paper.
>
> >“Q2.The universality proof sketch relies on walks covering all edges. How does the approximation quality degrade in practice when walks inevitably provide only partial coverage on large graphs? Does this partial coverage limit the types of functions FLOCK can effectively learn?”
>
> As the reviewer pointed out, our universality proof assumes sufficient computational resources such that the walks cover all edges, akin to the standard universality results for neural networks that require sufficient width or depth. We remark that deterministic equivariant networks are often limited in expressivity even with unlimited computational resources (e.g., see [7]).
>
> That said, yes, restricting the lengths of random walks will limit the family of approximable functions. Note that sampling a random walk is equivalent to sampling a subgraph consisting of the visited edges, establishing a link between Flock and subgraph GNNs. Previous work [6] has suggested an expressivity hierarchy based on $k$-reconstrucability, the ability to distinguish graphs based on their *multisets* of subgraphs of size $k$. When sampling walks of length $k$, Flock returns a distribution over $k$-subgraphs, not their multiset, which makes it nontrivial to compare the expressive power of Flock to that of $k$-reconstruction GNNs. We leave the expressivity studies of these restricted cases as an important future work.

---

> ### Author Response · Authors · 2025-11-23
>
> >“Q3.Given the computational overhead of sampling and ensembling, under what specific conditions (graph size, structure, task requirements) does FLOCK offer a clear practical advantage over potentially faster, deterministic KGFMs like TRIX or ULTRA, especially if these were to be augmented with techniques to increase expressivity (subgraph features or textual semantics like [2])? SEMMA [2] approaches the same problem of “losing the ability to distinguish between two entities with opposite relationships”, but by directly encoding the textual features into ULTRA. This at least warrants a discussion.”
>
> Thank you for the comment. We address the question in parts.
>
> > 1. Under what conditions does Flock perform best?
>
> Flock performs best when **structural disambiguation** is crucial, particularly in KGs where many relations are structurally similar but semantically distinct (as seen in Petals or Metafam). Still, we remark that Flock performs well in general KGs, not necessarily restricted to these specific KGs, as evidenced by our experimental results in **Section 5**.
>
> > 2. Will Flock be advantageous over deterministic KGFMs like TRIX and ULTRA augmented with techniques to increase expressivity (e.g., subgraph features or textual semantics [5])?
>
> We note that even with additional expressivity enhancements, e.g., with subgraph features or more parameters, the standard models would remain less expressive than Flock if they are still bounded by deterministic node-relation equivariance. In these cases, we expect Flock to be advantageous. As an empirical datapoint, we refer to **Section 6** of the ULTRA paper [5], where the authors discuss that the performance did not significantly improve with scaling the parameter count.
>
> In addition, we note that our supplementary experiment in **Appendix F** explicitly tests TRIX augmented with noise injection, which is specifically a technique that improves expressivity. The results provide additional support that Flock remains advantageous over current deterministic KGFMs augmented with techniques to increase expressivity.
>
> Regarding SEMMA [5], we would like to clarify that the proposed approach therein is **orthogonal** to ours. In principle, SEMMA-style textual semantics could also be combined with Flock’s random-walk structural encoding to enhance the performance, but this lies outside the scope of our current work.
>
> We appreciate the comment and will add the above discussion to the next revision of the paper.
>
> >“Q4.How was the sequence model architecture (biGRU + RMSNorm + SwiGLU) chosen? Did you experiment with other paradigms (like Transformers or SSMs), and how did they perform in terms of accuracy and efficiency?”
>
> Please see the response in **W3**, which shows that the SwiGLU-RMSNorm transformer, often used for language processing, with a similar parameter count, does not deliver satisfactory accuracy (although we have observed it is a bit faster due to parallelizability). In fact, Flock’s architecture (biGRU + RMSNorm + SwiGLU) has been chosen by first testing this transformer, and then changing the self-attention part to biGRU due to this accuracy issue. We also discussed the potential reason for this in the response in **W3**.

---

> ### Author Response · Authors · 2025-11-23
>
> >“Q5.Could the proposed probabilistic equivariance be achieved through methods other than random walks (like stochastic message passing, randomized node features)? How does the choice of random walks specifically contribute? Appendix F is great, but more than naive noise injection would make it stronger.”
>
> In principle, probabilistic equivariance can be achieved through other methods. Randomized node features via noise injection is a way to achieve it [11], which is the baseline we have tested in **Appendix F**. We note that the baseline is strong, as it uses as its backbone the prior state-of-the-art TRIX, and we additionally strengthen it with test-time ensembling with 16 samples for a fair comparison, which implements the idea of prior work on randomized node features (see **Eq. (1)** and **Appendix E.1.1** of [11]).
>
> More generally, probabilistic equivariance could be achieved with either structure-uninformed or structure-informed stochasticity. Simple uses of randomness, such as randomized node features via noise injection, would fall into the first category. Flock is in the second category, as it **introduces stochasticity through graph-aware random walk sampling**. The former does not depend on the underlying graph structure, whereas Flock’s sampling inherently reflects the topology and local connectivity patterns of each KG. Empirically, this difference matters, as we have shown in **Appendix F**, supporting our claim that structure-aware stochasticity is more effective in generalization.
>
> Regarding the choice of random walks, we would like to point to the additional ablation studies provided in the response to **W3**. In summary, the results show that localizing behavior of random walks (e.g., backtracking or short walks) makes learning easier but degrades performance on large KGs due to coverage issues, and in contrast, too long walks pose scalability and learning challenges. As a result, the experiments show that our choice of random walk algorithm provides a good balance considering predictive power and scalability.
>
>
> ---
>
>
> *[1] Zhu et al., Neural Bellman-Ford networks: A general graph neural network framework for link prediction, NeurIPS 2021.*
>
> *[2] Galkin et al., Towards foundation models for knowledge graph reasoning, ICLR 2024.*
>
> *[3] Lacki et al., Walking randomly, massively, and efficiently, arXiv 2020.*
>
> *[4] Zhou et al., A multi-task perspective for link prediction with new relation types and nodes, arXiv 2023.*
>
> *[5] Arun et al., SEMMA: A Semantic Aware Knowledge Graph Foundation Model, EMNLP 2025.*
>
> *[6] Cotta et al., Reconstruction for Powerful Graph Representations, NeurIPS 2021.*
>
> *[7] Xu et al., How powerful are graph neural networks? ICLR 2019.*
>
> *[8] Galkin et al., Towards foundation models for knowledge graph reasoning, ICLR 2024.*
>
> *[9] Beck et al., xLSTM: Extended long short-term memory, arXiv 2024.*
>
> *[10] Kaplan et al., Scaling laws for neural language models, arXiv 2020.*
>
> *[11] Gao, Jianfei, et al. Double equivariance for inductive link prediction for both new nodes and new relation types. arXiv 2023.*

---

> > ### Comment · Reviewer_w7Ej · 2025-11-27
> >
> > All of my concerns were addressed, which were mostly about more complete ablations and the practical applicability of the proposed method. The authors provided real dataset examples in Appendix G and added a detailed ablation study in Table 5.
> >
> > Provided the authors include the leftover promised writing changes (I'm not sure why it hasn't been included already and left over to the next revision), I think this is a valuable addition to the KGFM community, and hence I'm increasing my score to 8.

---

> > > ### Author Response · Authors · 2025-11-27
> > >
> > > We thank the reviewer for responding to our rebuttal and increasing their score. We are glad that the additional ablations and real dataset examples addressed their concerns. We will incorporate the remaining suggested changes in the next revision after consolidating feedback from all the reviewers.

---

> > > ### Author Response · Authors · 2025-11-28
> > >
> > > We thank the reviewer for raising their score from 6 to 8 following our rebuttal.

---

### Official Review · Reviewer_qE2N · 2025-11-01

**Soundness:** 3
**Presentation:** 2
**Contribution:** 2
**Rating:** 4
**Confidence:** 3

**Summary:**

The work proposes a novel knowledge graph foundation model (KGFM) architecture, FLOCK, with stochastic equivariant node and relation representations. The stochastic equivariance property endows the model with better expressivity without sacrificing zero-shot generalizability to novel KGs with unseen node and relation types. The authors achieve this via an innovative adoption of random walks that anonymize the identities of nodes and relation types, thus keeping the model informed of solely the structural roles that the visited nodes and relation types play in the random walks. In the experimental part, the paper then demonstrates the performance of FLOCK on a wide range of both synthetic and real-world KGs, showing superior performance to the baselines and the capability to distinguish isomorphic relation types for the link prediction tasks on the synthetic PETALS dataset.

**Strengths:**

1. The paper correctly identifies that most of the existing KGFMs employs deterministic node-relation equivariance, which creates an expressivity bottleneck. The paper then follows the well-established, theoretically sound notion of stochastic/probabilistic equivariance. The introduction of random-walk based methodology to tackle the KG representation is novel, and the proposed model design is both intuitive and well-motivated.

2. The paper creates the new PETALS synthetic dataset to cleverly demonstrate the unique advantage of FLOCK at distinguishing isomorphic relation types. This group of experiments is well-designed, providing targeted empirical validation. The rest of the experiments are also diverse and comprehensive, with FLOCK showing consistently superior performance.

**Weaknesses:**

W1. **My major concern is the paper's novelty claim of "the first stochastic KGFM"**. As identified in one of the previous work (Gao et al. 2023), the InGram model (Lee et al., 2023) can be viewed as a stochastic node-relation equivariant KG model, because it injects random initial node and relation type embeddings, thus creating node and relation embeddings that are equivariant in distribution. Based on this observation, Gao et al. then proposed an improvement, named DEq-InGram, that further stabilizes the stochastic equivariant embeddings of the original InGram, showing significant improvement in performance. Thus, I do not believe that this work may claim that FLOCK is the "first stochastic KGFM." Could the author then clarify this point? If DEq-InGram was indeed a predecessor to FLOCK on the line of stochastic KGFMs, could the author further elaborate how FLOCK could be uniquely advantageous compared to DEq-InGram?


W2. **Missing error bars on the empirical results**. In all experiment tables in the main text, only the average numbers are reported. The error bars (e.g. standard deviation numbers) are missing. It would be informative to see the comparison of variance of model performance, particularly for stochastic equivariant models, because the stochasticity nature of the model usually results in higher variance of model prediction accuracy.

W3. **Clarity of writing and lack of critical technical details in methodology**. The writing on the random walks section (line 219 - 229) is a bit confusing to read and could be improved. For instance, the paragraph on line 226 seems to repeat the same message with those discussed in the preceding paragraph. Furthermore, in this preceding paragraph that starts on line 219, it is not clearly explained why $3n$ walks are needed. Could the authors clarify why are there $3n$ walks?

Later in the sequence processor section (line 257 - 269), it is critical for the readers to understand how does the "structural indices" such as 1, 2, 3, for nodes and $\alpha$, $\beta$ from equation (5) and (6) translates to which neural embeddings in the GRU model from equation (7). This part is critical because later, the analysis of the equivariance-ness of the entire pipeline hinges on it. It would be great if more writing effort is devoted here to clarify and elaborate how the structural indices interacts with the GRU model.


W4. **Ablation on the Consensus Protocol**. In the consensus protocol section, the authors made the following claim regarding the approach of taking averages of the proposals:

> The drawback is that uninformative proposals from e.g. dangling regions of walks are not directly suppressed, and can affect the state updates.

This claim seems to be a conjecture at this stage, without theoretical analysis or empirical evidence. Would it be possible to provide further evidence to support this claim, such as some sort of ablation studies?


W5. **Lack of proof insight or proof sketch**. Although this is not necessarily a critical concern or weakness, it would be great if the theoretical section 4.2 have some discussions on the insights or ideas or proof sketch to illustrate intuitively, why does the claims and propositions are mathematically correct. Currently, it reads like the section 4.2 is simply an information dump of related theoretical results without much insights into their correctness.


W6. **Notations clarity**: On line 154

> Let $\omega$ be a function assigning to each KG $G = (V, E, R) \in \mathbb{K}_{n, m}$ ...

The notation $\mathbb{K}_{n, m}$ is first used before being properly defined and introduced (it was later introduced in Line 158).


References:

[1] Gao, Jianfei, Yangze Zhou, Jincheng Zhou, and Bruno Ribeiro. "Double equivariance for inductive link prediction for both new nodes and new relation types." arXiv preprint arXiv:2302.01313 (2023).

[2] Lee, Jaejun, Chanyoung Chung, and Joyce Jiyoung Whang. "InGram: Inductive knowledge graph embedding via relation graphs." In International conference on machine learning, pp. 18796-18809. PMLR, 2023.

**Questions:**

Please see the Weaknesses section for my questions and concerns.

---

> ### Author Response · Authors · 2025-11-23
>
> > “W1. My major concern is the paper's novelty claim of "the first stochastic KGFM". As identified in one of the previous work (Gao et al. 2023), the InGram model (Lee et al., 2023) can be viewed as a stochastic node-relation equivariant KG model, because it injects random initial node and relation type embeddings, thus creating node and relation embeddings that are equivariant in distribution. Based on this observation, Gao et al. then proposed an improvement, named DEq-InGram, that further stabilizes the stochastic equivariant embeddings of the original InGram, showing significant improvement in performance. Thus, I do not believe that this work may claim that FLOCK is the "first stochastic KGFM." Could the author then clarify this point? If DEq-InGram was indeed a predecessor to FLOCK on the line of stochastic KGFMs, could the author further elaborate how FLOCK could be uniquely advantageous compared to DEq-InGram?”
>
> We thank the reviewer for raising this point. InGram [1] and DEq-InGram [2] indeed introduced stochasticity into KGFMs through random node and relation initialization, and we now explicitly acknowledge these works in the paper (**Section 2 and Appendix F**). Accordingly, we have revised our phrasing to describe Flock as “a stochastic KGFM”, rather than “the first stochastic KGFM” (Line 114).
>
> While these prior methods incorporate randomness, we would like to highlight a fundamental conceptual distinction. **Flock introduces stochasticity through graph-aware random walk sampling**, whereas InGram and DEq-InGram rely on **uniform noise injection** over nodes and relations in the initialization process. The latter does not depend on the underlying graph structure, whereas Flock’s sampling inherently reflects the topology and local connectivity patterns of each KG. Empirically, this difference matters: Flock consistently outperforms a strong uniform noise-injection baseline (TRIX + noise with 16-samples ensembling) in **Table A**, (and also in **Appendix F, Table 8**), supporting our claim that structure-aware stochasticity is more effective in generalization.
>
> We remark that DEq-InGram is essentially a trained InGram with test-time ensembling with 10 samples (see Eq. (1) and Appendix E.1.1 of [2]). Since our noise injection baseline uses test-time ensembling with 16 samples, and its backbone is the prior state-of-the-art model TRIX, we believe it can be used as a strong baseline implementing the idea of DEq-InGram.
>
> ---
>
> **Table A.** Comparison of TRIX, noise-injected TRIX, and Flock across three tasks.
>
> | **Task**  | **Metric**  | **TRIX** | **TRIX + noise** | **Flock** |
> |----|---|---|---|-|
> | **Zero-shot entity prediction** | MRR         | 0.366    | 0.385            | **0.391** |
> | **Zero-shot relation prediction** | MRR       | 0.792    | 0.739            | **0.881** |
> | **Petals**             | Accuracy    | 50%      | 52%              | **100%**  |
>
> ---
>
>
> > “W2. Missing error bars on the empirical results. In all experiment tables in the main text, only the average numbers are reported. The error bars (e.g. standard deviation numbers) are missing. It would be informative to see the comparison of variance of model performance, particularly for stochastic equivariant models, because the stochasticity nature of the model usually results in higher variance of model prediction accuracy.”
>
> We thank the reviewer for the comment. To test for the variance of Flock’s performance, we repeated zero-shot entity prediction on inductive KGs three times. The results are in **Table B**. We observe that the standard deviations over repeated trials are small, indicating a consistent prediction performance. This is partially due to the fact that we are already ensembling 16 randomized predictions of Flock at test-time, which has a variance reduction effect in addition to performance improvement. While we could not run repeated experiments for all tables due to the resource constraints, we will do our best to include error bars as much as possible in the final version.
>
> ---
>
> **Table B.** Performances of Flock over three repeated tests. TRIX included for comparison (MRR/Hits@10).
>
> | **Model** | **Inductive $(e,r)$ (23 KGs)** | **Inductive $e$ (18 KGs)** |
> |-----------------------|------------------------------------|-----------------------------------|
> | | MRR / H@10 | MRR / H@10 |
> | TRIX | 0.368 / 0.540 | 0.455 / 0.592 |
> | Flock trial 1 | 0.369 / 0.554 | 0.456 / 0.604 |
> | Flock trial 2 | 0.369 / 0.555 | 0.456 / 0.604 |
> | Flock trial 3 | 0.370 / 0.550 | 0.456 / 0.603 |
> | Flock average and std | **0.369 ± 0.000 / 0.553 ± 0.003** | **0.456 ± 0.000 / 0.604 ± 0.000** |
>
> ---

---

> ### Author Response · Authors · 2025-11-23
>
> > “W3. Clarity of writing and lack of critical technical details in methodology. The writing on the random walks section (line 219 - 229) is a bit confusing to read and could be improved. For instance, the paragraph on line 226 seems to repeat the same message with those discussed in the preceding paragraph. Furthermore, in this preceding paragraph that starts on line 219, it is not clearly explained why 3n walks are needed. Could the authors clarify why there are 3n walks?”
>
> We thank the reviewer for the constructive comment. Following the suggestion, we have improved the descriptions of the random walk sampling in **Section 4.1**.
>
> Regarding the $3n$ walks, we need $3n$ walks to diversify the starting locations such that **local context** around the query $q = (h, r, ?)$, and **globally broad coverage** of the nodes and relations in the KG, are both well-captured in the random walks. The first $n$ walks start at $h$, capturing local context; the second $n$ walks start at edges of random relations, broadly capturing relations including $r$; the last $n$ walks start at random nodes, broadly capturing various regions of the KG. **Table C** compares this against all $3n$ walks starting at the query node. As expected, this causes degradations on all splits, showing the benefit of using both local and global information.
>
> ---
>
> **Table C.** Additional ablation study results (MRR/Hits@10).
>
> | **Model** | **Inductive $(e,r)$** | **Inductive $e$** | **Transductive\*** | **Average** |
> |---------------------|------------------------|-------------------|--------------------|--------------------|
> | | MRR / H@10 | MRR / H@10 | MRR / H@10 | MRR / H@10 |
> | Flock (Ours) | 0.369 / 0.554 | 0.456 / 0.604 | 0.360 / 0.542 | **0.395 / 0.567** |
> | w/o diverse starts | 0.360 / 0.539 | 0.448 / 0.596 | 0.319 / 0.488 | 0.385 / 0.553 |
>
> \* NELL995, NELL23k, WDsinger, ConceptNet100k, YAGO310 tested due to resource limit.
>
> ---
>
> > “Later in the sequence processor section (line 257 - 269), it is critical for the readers to understand how does the "structural indices" such as 1, 2, 3, for nodes and $\alpha$, $\beta$ from equation (5) and (6) translates to which neural embeddings in the GRU model from equation (7). This part is critical because later, the analysis of the equivariance-ness of the entire pipeline hinges on it. It would be great if more writing effort is devoted here to clarify and elaborate how the structural indices interact with the GRU model.”
>
> We thank the reviewer for the comment. We use standard categorical embedding tables (i.e., nn.Embedding in PyTorch) to translate the structural indices (anonymizations) into feature vectors. This gives a feature vector per anonymization on the sequence, which is then processed by the GRU in the usual way. Our proof in **Appendix B.2** reflects this. We have revised the description of the sequence processor (**Line 275**) to clarify this.
>
> > “W4. Ablation on the Consensus Protocol. In the consensus protocol section, the authors made the following claim regarding the approach of taking averages of the proposals: ‘The drawback is that uninformative proposals from e.g. dangling regions of walks are not directly suppressed, and can affect the state updates.’ This claim seems to be a conjecture at this stage, without theoretical analysis or empirical evidence. Would it be possible to provide further evidence to support this claim, such as some sort of ablation studies?”
>
> We thank the reviewer for the constructive comment. We agree that having an ablation study would add evidence to our intuition for the consensus protocol. Following the suggestion, we ran an additional experiment where we replace the consensus protocol with unweighted averages of the state update proposals, as discussed in **Lines 290-294**. The results are in **Table D**, and show that the consensus protocol is indeed contributing significantly to the performance of Flock.
>
> ---
>
> **Table D**. Additional ablation study for the consensus protocol (MRR/Hits@10).
>
> | **Model** | **Inductive $(e,r)$** | **Inductive $e$** | **Transductive\*** | **Average** |
> |------------------|-----------------------|-------------------|--------------------|--------------------|
> | | MRR / H@10 | MRR / H@10 | MRR / H@10 | MRR / H@10 |
> | Flock (Ours) | 0.369 / 0.554 | 0.456 / 0.604 | 0.360 / 0.542 | **0.395 / 0.567** |
> | w/o consensus | 0.351 / 0.526 | 0.448 / 0.593 | 0.361 / 0.515 | 0.387 / 0.545 |
>
> \* NELL995, NELL23k, WDsinger, ConceptNet100k, YAGO310 tested due to resource limit.

---

> ### Author Response · Authors · 2025-11-23
>
> > “W5. Lack of proof insight or proof sketch. Although this is not necessarily a critical concern or weakness, it would be great if the theoretical section 4.2 have some discussions on the insights or ideas or proof sketch to illustrate intuitively, why does the claims and propositions are mathematically correct. Currently, it reads like the section 4.2 is simply an information dump of related theoretical results without much insights into their correctness.”
>
> We thank the reviewer for this suggestion. While we have included a brief proof sketch in **Appendix B.1**, we updated the manuscript to have short outlines below **Propositions 4.1** and **4.2**. We briefly repeat the discussion here.
>
> **Expressivity**: If random walks in Flock are sufficiently long, they eventually cover all edges of the graph, and the anonymized recording assigns unique positional identifiers to every visited node and relation. Thus, with a sufficiently expressive sequence processor, Flock can therefore distinguish any nodes or relations and approximate any link-invariant function.
>
> **Invariance**: The random walk is probabilistically invariant, the recording protocol removes all graph-specific identifiers, and the consensus protocol aggregates only by structural roles. Since each of these components is invariant (in probability), and the invariance of an individual component is preserved under composition, we have that Flock is invariant.
>
> > “W6. Notations clarity: On line 154, “Let $\omega$ be a function assigning to each KG $G = (V,E,R) \in K_{n,m}$.The notation $K_{n,m}$ is first used before being properly defined and introduced (it was later introduced in Line 158).”
>
> We thank the reviewer for pointing this out. We have moved this definition to the initial “Knowledge graphs” paragraph.
>
> ---
>
> *[1] Lee, Jaejun, Chanyoung Chung, and Joyce Jiyoung Whang. "InGram: Inductive knowledge graph embedding via relation graphs." International conference on machine learning. PMLR, 2023.*
>
> *[2] Gao, Jianfei, et al. "Double equivariance for inductive link prediction for both new nodes and new relation types." arXiv preprint arXiv:2302.01313 (2023).*

---

### Official Review · Reviewer_LNUH · 2025-11-01

**Soundness:** 4
**Presentation:** 4
**Contribution:** 4
**Rating:** 8
**Confidence:** 3

**Summary:**

An architecture for zero-shot entity and relation prediction. FLOCK adopts a probabilistic approach to the equivariance assumption, as opposed to the conventional deterministic interpretation. The method relies on random walks sampling and direct sequence encoding.

**Strengths:**

- Research problem/limitation of SoTA KGFMs is well explained and clearly justified.
- Novelty: FLOCK is the first to adopt a stochastic take on node-relation equivariance. Extending the notion of probabilistic invariance to KGs is more than reasonable.
- Comprehensive evaluation campaign. Up-to-date baselines choice from literature. Protocol shows fair comparison w.r.t. to baselines.
- Relation prediction results clearly stronger than conventional KGFMs.
- Writing is clear and narrative flows well and is coherent.
- Related work is comprehensive and up to date.

**Weaknesses:**

- Scalability of random walks on large KGs (the authors are well aware and they have discussed this in the conclusions)
- I could not find an overall training time and inference time comparison against conventional KGFMs such as ULTRA, TRIX, FLOCK. It is not entirely clear to which extent the adoption of random walks is slower than message passing? (apologies if I have missed)
- Results: entity prediction results only marginally better than prior art.
- (minor) Figure 2 could help a better caption, to clarify color coding and actions taken at each step.
- Some examples would help clarify, e.g. "They reveal nodes $v_s$ and relations $r_s$ specific to each KG which obstructs transferability to unseen KGs." (line 239)

**Questions:**

- What is the recommended interval of $l$ values for random walks, keeping in mind impact on predictive power and training time?
- Have you considered using a transformer architecture for the sequence processor?
- line 236: Can you clarify what do you mean by "to keep GPU memory usage in a range"?

---

> ### Author Response · Authors · 2025-11-23
>
> > “W1. Scalability of random walks on large KGs (the authors are well aware and they have discussed this in the conclusions)”
>
> We thank the reviewer for the comment. As mentioned, we acknowledged the challenge of scaling random walks to large KGs in **Section 6**. Currently, Flock’s cost scales proportionally to the walk counts and lengths, instead of the full KG sizes, and we can leverage this to induce a cost-performance trade-off that helps handle large KGs. For example, we clamp the walk counts to 512 at maximum, which allows us to scale to the largest transductive KGs such as YAGO310 (**Tables 15, 18, 19**). At the same time, we believe this scalability challenge can be mitigated in the future by developing efficient approximations of random walks, which has been historically of significant interest in the distributed and parallel computing literature (e.g., [1]).
>
>
>
> > “W2. I could not find an overall training time and inference time comparison against conventional KGFMs such as ULTRA, TRIX, FLOCK. It is not entirely clear to which extent the adoption of random walks is slower than message passing? (apologies if I have missed)”
>
> We have included the overall training and inference time analysis with other KGFMs in **Appendix E**. While Flock is, in general, slower than message passing due to the dominating cost of random walk sampling, we believe it has avenues for significant acceleration in future work. For example, the compared message passing baselines (ULTRA and TRIX) leverage highly optimized fused sparse kernels ([2]), and we believe similar dedicated optimizations could benefit Flock. We additionally remark that these additional costs yield substantially higher expressivity and thus better performances overall in the downstream tasks, while it is known that ULTRA does not significantly benefit from scaling e.g., model size (see Section 6 of [3]).
>
> > “W3. Results: entity prediction results only marginally better than prior art.
> (minor) Figure 2 could help a better caption, to clarify color coding and actions taken at each step.”
>
> We thank the reviewer for the comment. We agree that there is room for improvement in entity prediction results, but surpassing the state-of-the-art results with a completely different approach than prior works (e.g., no message passing, no relation graph) is a fundamental contribution, which can enable further improvements in future work. We would also like to remark that our improvements are the most pronounced in relation prediction and Petals, validating the structural reasoning of our method.
>
> For the minor point, we have improved the caption of **Figure 2**. We thank the reviewer for the constructive suggestion.
>
> > “W4. Some examples would help clarify, e.g. "They reveal nodes $v_s$ and relations $r_s$ specific to each KG which obstructs transferability to unseen KGs." (line 239)”
>
> Thank you for the constructive suggestion. We will add a short illustrative example in the final version.

---

> ### Author Response · Authors · 2025-11-23
>
> > “Q1. What is the recommended interval of $\ell$ values for random walks, keeping in mind impact on predictive power and training time?”
>
> In our experiments, we fix $\ell=128$ after empirical validation of the lowest length reliably visiting the target node and relation type. We found that $\ell = 128$ yields nearly 100% entity and relation coverage on all benchmark KGs, while remaining efficient for the sequence processor.
>
> In general, the length of a random walk induces a trade-off between coverage of KG and computational/reasoning efficiency. With too short walks, we expect the performance to degrade on large KGs (e.g., ones found in transductive setups) due to the coverage issues; with too long walks, we expect computational inefficiency and learnability challenges with sequence neural networks. Hence, the best length needs to be balanced between these two factors.
>
> To illustrate this, we ran an ablation study using 0.5x (64) and 2x (256) lengths of random walks. The results are in **Table A**. As expected, shorter walks are easier to learn for the sequence neural network but lead to degraded performances for large transductive KGs; longer walks pose efficiency and learnability issues. Overall, we find that our choice of $\ell=128$ provides a good balance, keeping in mind the predictive power and scalability.
>
> ---
>
> **Table A.** Controlling random walk lengths (MRR/Hits@10).
>
> | **Model** | **Inductive** $(e,r)$ | **Inductive** $e$ | **Transductive\*** | **Average** |
> |--------------------|-----------------------|-------------------|--------------------|----------------|
> | | MRR / H@10 | MRR / H@10 | MRR / H@10 | MRR / H@10 |
> | Flock (Ours) | 0.369 / 0.554 | 0.456 / 0.604 | 0.360 / 0.542 | 0.395 / 0.567 |
> | walk length 0.5x | 0.372 / 0.556 | 0.459 / 0.606 | 0.351 / 0.534 | 0.394 / 0.565 |
> | walk length 2x | 0.360 / 0.548 | 0.458 / 0.605 | 0.338 / 0.508 | 0.385 / 0.553 |
>
> \* NELL995, NELL23k, WDsinger, ConceptNet100k, YAGO310 tested due to resource limit.
>
> ---
>
> > “Q2. Have you considered using a transformer architecture for the sequence processor?”
>
> Following the reviewer’s advice, we have conducted additional experiments using the RMSNorm-SwiGLU transformer with a similar parameter count and shown the results in **Table B**.  This alternative does not deliver good results, which is explained by the restrictions on model scales that are enforced to scale to large KGs. Flock benefits from the reasoning efficiency of GRU in a limited parameter regime, gaining good performance and scalability together.
>
> ---
>
> **Table B**. Additional ablation study results (MRR/Hits@10).
>
> | **Model** | **Inductive** $(e,r)$ | **Inductive** $e$ | **Transductive\*** | **Average** |
> |----------------------|-----------------------|-------------------|--------------------|--------------------|
> | | MRR / H@10 | MRR / H@10 | MRR / H@10 | MRR / H@10 |
> | Flock (Ours) | 0.369 / 0.554 | 0.456 / 0.604 | 0.360 / 0.542 | **0.395 / 0.567** |
> | transformer $f_\theta$ | 0.356 / 0.542 | 0.410 / 0.591 | 0.312 / 0.477 | 0.359 / 0.537 |
>
> \* NELL995, NELL23k, WDsinger, ConceptNet100k, YAGO310 tested due to resource limit.
>
> ---
>
> > “Q3. line 236: Can you clarify what do you mean by "to keep GPU memory usage in a range"?”
>
> Apologies for the confusion. We mean that we limit the choice of $n$ to avoid Out-Of-Memory errors on GPUs, e.g., a maximum of 512 for H100 GPUs. We have revised the text to clarify this.
>
> ---
>
> *[1] Lacki et al., Walking randomly, massively, and efficiently, STOC 2020.*
>
> *[2] Zhu et al., Neural Bellman-Ford networks: A general graph neural network framework for link prediction, NeurIPS 2021.*
>
> *[3] Galkin et al., Towards foundation models for knowledge graph reasoning, ICLR 2024.*
>
> *[4] Kaplan et al., Scaling laws for neural language models, arXiv 2020.*

---

### Author Response · Authors · 2025-11-23
**Response to all reviewers**

We thank the reviewers for their insightful and constructive feedback that has helped us significantly improve the paper.

We summarize the changes we have made to improve the presentation and clarity of the paper, as well as to supplement the experiments (leveraging the additional page), following feedback from all reviewers:

- Revised introduction and related work sections to clarify novelty and contribution (*Reviewer qE2N*).
- Added details to the caption of **Figure 2** to improve the overview description of Flock (*Reviewer LNUH*).
Improved methods section to more clearly communicate the design rationale and mechanism of each component of Flock (*Reviewer qE2N*).
- Added proof sketches to the theoretical results section to offer insights (*Reviewer qE2N*).
- Added experiments to explain the behavior of Flock and the role of each of its design choices:
  - (**Ablation studies, Table 5**). A comprehensive ablation study of random walks' lengths (*Reviewers LNUH, w7Ej, sNAr*), random walks starting point distribution (*Reviewer qE2N*), consensus protocol (*Reviewer qE2N*), and sequence network (*Reviewer LNUH, Reviewer w7Ej*), showing the importance of our design choices.
  - (**Case study on Metafam, Appendix G**). A case study on Metafam, verifying our key insight that Flock distinguishes structurally similar but semantically distinct relations, while ULTRA fails to do so (*Reviewer w7Ej, sNAr*).
  - (**Sparsity Analysis, Appendix H**). An analysis of the remaining 53 KGs shows that the performance of Flock is correlated with the sparsity of KGs, agreeing with our theory and existing work on edge coverages of random walks (*Reviewer LNUH, w7Ej*).
- Extended the conclusions section with additional directions on future work.

All revisions are highlighted in blue in the updated manuscript.

---

### Author Response · Authors · 2025-11-30
**Summary of changes during the rebuttal**

Dear AC,

We appreciate your efforts in this challenging situation.

To assist with assessing our work, we would like to provide an overview of changes during the rebuttal:

- **Reviewer w7Ej (6 → 8):** Clarified scalability via cost-performance tradeoff and low-variance via ensembling. Added evaluations on 31 KGs that suggest robust generalization. Added detailed ablations in Table 5 (walk length & sampler, sequence model, consensus). Added evidence of theoretical benefits turning into practical gains in Appendix G.
- **Reviewer LNUH (8):** Clarified scalability via cost-performance tradeoff. Improved caption of Figure 2. Added detailed ablations in Table 5 (walk length and sequence model).
- **Reviewer qE2N (4):** Revised “first stochastic KGFM” to “a stochastic KGFM” and acknowledged prior work on noise injection, while clarifying our novelty of using structure-aware stochasticity (random walks) in KGFMs and its empirical gain over noise injection (Appendix F). Clarified low-variance via ensembling. Improved writing of methods and added proof sketches. Added detailed ablations in Table 5 (walk start locations, consensus).
- **Reviewer sNAr (4 → 6):** Explained zero-shot vs. fine-tuning differences in Metafam. Added evidence of theoretical benefits turning into practical gains in Metafam (Appendix G). Added sparsity-performance analysis across 53 KGs with a theoretical grounding (Appendix H). Added detailed ablations in Table 5 (walk sampler). Analyzed failure cases in transductive split with evidence of benchmark artifacts.

This resulted in the **updated score of 8,8,6,4 (Average 6.5)**. All reviewers recognized the novelty, clear motivations, comprehensive evaluations, strong empirical results, and clear writing.

We hope this summary is helpful for your assessment.

Kind regards,

Authors of submission 19217

---

### Meta-Review · Area_Chair_9Ssn · 2026-01-07

**Summary:**

All reviewers recognized the strong motivation, theoretical grounding, comprehensive evaluation, and clear writing. Initial concerns included scalability of random walks on large graphs, novelty claims, missing error bars, lack of detailed ablations, and questions about practical advantage over deterministic KGFMs. The authors responded thoroughly: revising novelty claims, adding extensive ablations, providing variance analysis, clarifying methodology, reporting training/inference costs, and demonstrating robustness on challenging KG subsets. Two reviewers explicitly upgraded their score post-rebuttal. The opinions of the reviewers who gave 4 points have also been fully addressed.

**Reviewer Concerns:**

The authors addressed virtually all technical concerns raised.

**Reviewer Scores:**

Reviewer w7Ej (initial 6, later updated to 8 in summary) explicitly stated after rebuttal. Their final score is 8.
Reviewer LNUH (initial 8) did not respond post-rebuttal but had already rated the paper above threshold.
Reviewer qE2N (initial 4) received comprehensive responses but gave no indication of changing their assessment; their score is maintained at 4.
Reviewer sNAr (initial 4, later updated to 6 in summary) did not comment after rebutta but modified score as 6 based on their acknowledgment of addressed concerns.

---

### Decision · Program_Chairs · 2026-01-26

Accept (Poster)